# GRAPH DIFFUSION TRANSFORMERS ARE IN-CONTEXT MOLECULAR DESIGNERS

**Gang Liu**[1], **Jie Chen**[2], **Yihan Zhu**[1], **Michael Sun**[3]
**Tengfei Luo**[1], **Nitesh V. Chawla**[1], **Meng Jiang**[1]
[1]University of Notre Dame    [2]MIT-IBM Watson AI Lab, IBM Research    [3]MIT CSAIL
`{gliu7, mjiang2}@nd.edu`

## ABSTRACT

In-context learning lets large models adapt to new tasks from a few demonstrations, but it has shown limited success in molecular design, where labeled data are scarce and properties span millions of biological assays and material measurements. We introduce demonstration-conditioned diffusion models (DemoDiff), which define task contexts through molecule–score examples instead of texts. These demonstrations guide a denoising Transformer to generate molecules aligned with target properties. For scalable pretraining, we develop a new molecular tokenizer with Node Pair Encoding that represents molecules at the motif level, requiring $5.5\times$ fewer nodes. We pretrain a 0.7B parameter model on datasets covering drugs and materials. Across 33 design tasks in six categories, DemoDiff matches or surpasses language models $100–1000\times$ larger and achieves an average rank of 4.10 compared to 6.56–17.95 for 19 baselines. These results position DemoDiff as a molecular foundation model for in-context molecular design.

## 1 INTRODUCTION

In-context learning (ICL) is the emergent capability of large models to infer task-specific concepts from a few demonstrations (Xie et al., 2021). ICL has been studied in large language models (LLMs), but was found less effective for molecular design than specialized methods (Liu et al., 2024b). These specialized models often depend on extensive Oracle calls (Gao et al., 2022) or large labeled datasets beyond what context examples provide. Molecular tasks, however, involve millions of types, many with only a few labeled examples (Zdrazil et al., 2024). Such examples are enough to form task contexts but insufficient to train a new model. This trade-off motivates our *in-context molecular design*, which combines the flexibility of ICL with the efficiency of molecular domain knowledge.

Molecular structures and properties are discrete graphs and numbers with varying scales and units. Directly adapting the autoregressive framework from LLMs is infeasible (Brown et al., 2020) for in-context molecular designs, as the input and output of language data are text in sequential order. Diffusion models show promise for molecular structures (Vignac et al., 2022), and Graph Diffusion Transformers (Graph DiTs) are effective for modeling their joint distribution with properties (Liu et al., 2024c). However, Graph DiTs have been studied with at most five properties represented in a single vector. In practice, molecular properties span millions of assays in biology, including functions, binding, ADME, and toxicity, as well as material properties such as gas permeability, thermal conductivity, and glass transition temperature (Figure 1). Representing all properties in one-hot vector with millions of dimensions is inefficient, produces sparse pretraining data since many assays have fewer than ten labels, and limits generalization to unseen properties in downstream tasks.

Instead of a property vector with a large embedding table, we use demonstrations to define the task context for molecular design. As shown in Figure 1, the demonstrations consist of a set of molecules with scores in $[0, 1]$ and molecular design is framed as a query for the target score of 1. Molecules in the context do not follow a strict order, and their scores serve as *relative* positions to the target, functioning as a replacement for position IDs in Transformers. (Xie et al., 2021) described ICL in LLMs as implicit Bayesian inference over latent *concepts* expressed by examples in the prompt. Similarly, each task in Figure 1 shares the concepts defined by the joint distribution of molecules and their scores. The denoising Transformer in the Graph DiT attends to the context, implicitly

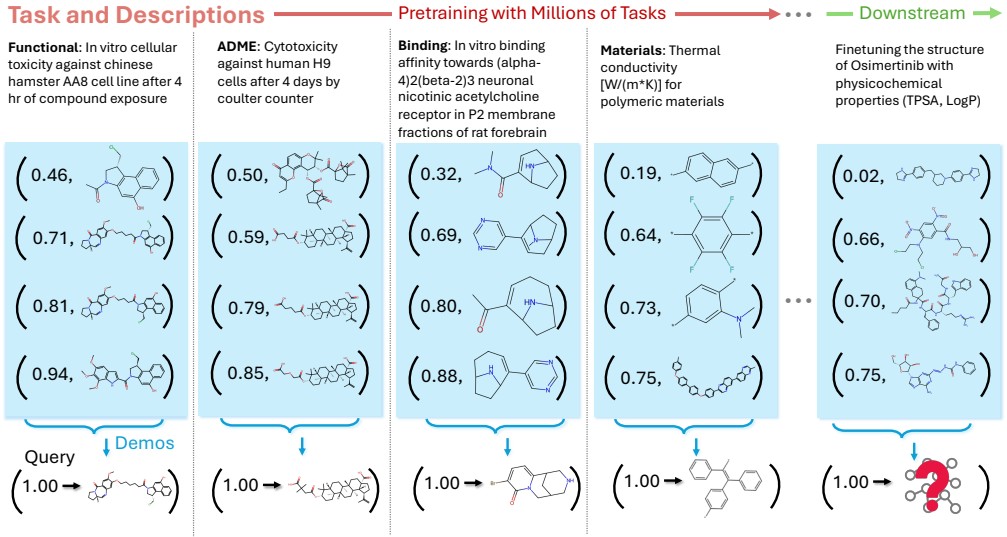

Figure 1: In-context molecular design with DemoDiff. Each demo is defined as a score–molecule pair, and a set of them forms the task context as conditions. After pretraining on large and diverse tasks, DemoDiff serves as a foundation model for designing molecules in new task contexts. Scores represent relative distances to the target and are converted from raw labels, as shown in Section 3.3.

extracts concepts, and uses them to guide the reverse process to refine the structure. An example generation trajectory is shown in Figure 7. A simple way to represent the task concept is to use positive demonstrations, such as molecules close to the target or active in the assay. However, positive examples alone may be insufficient, as they can overlap across tasks due to factors such as task relatedness (e.g., activity in non-small-cell lung cancer but across different cell lines) or due to sampling bias when the set of positives is extremely sparse (e.g., only one positive example shared by two tasks). To address this, we form the task context using not only positive but also medium and negative examples, providing a more complete representation of the task concepts.

With these task contexts, we propose **demo**nstration-conditioned **diff**usion models (DemoDiff) and pretrain a 0.7B model with a Graph DiT as the backbone, using over 140 H100 GPU days. To support efficient pretraining, we introduce a molecular tokenizer trained with Node Pair Encoding (NPE) for motif-level representation. On average, it reduces the number of nodes by 5.5× compared to atom-level representations (Figures 9 and 10). The tokenizer iteratively merges neighboring nodes and selects frequent motifs to construct vocabularies. Motifs are connected by directed edges that preserve bond types and attachment rules, ensuring lossless reconstruction. Graph DiTs naturally use this motif-level representation and attend to motif semantics for denoising. For pretraining, we construct a dataset of over 1.6 million tasks from 155K unique properties and one million molecules. It combines ChEMBL for drugs (Zdrazil et al., 2024) and multiple polymer data sources for materials (Otsuka et al., 2011; Thornton et al., 2012; Kuenneth et al., 2021). For ICL, we propose a consistency score as a confidence measure of whether a generation aligns more closely with higher-scoring molecules in demonstrations, effectively filtering out false positives in generation (Section 4.3).

We evaluate DemoDiff on 33 design tasks across six categories. With 0.7B parameters, it matches or surpasses LLMs 100–1000× larger in generating diverse, high-scoring molecules. Compared to 19 baseline models (average ranks 6.56–17.95), DemoDiff ranks 4.10, demonstrating its strength as a molecular foundation model. The new molecular tokenizer further improves representation efficiency.

## 2 PRELIMINARIES

### 2.1 IN-CONTEXT LEARNING WITH DEMONSTRATIONS

In ICL, the context is a set of demonstrations $\mathcal{C} = \{e_i\}_{i=1}^{L}$, where each $e_i$ is an input–label pair. Following (Xie et al., 2021), we assume $\mathcal{C}$ reflects a latent concept $\theta \in \Theta$ from a family of concepts

$\Theta$. For example, for a paragraph about Albert Einstein, the latent concept may be biography. Given a query $Q$, a foundation model with ICL generates an output $X$ by marginalizing over $\theta$:

$$p(X \mid \mathcal{C}, Q) = \int_\theta p(X \mid \theta, \mathcal{C}, Q)\, p(\theta \mid \mathcal{C}, Q)\, d\theta. \tag{1}$$

Here $(\mathcal{C}, Q)$ form the prompt. If $p(\theta|\mathcal{C}, Q)$ concentrates on the prompt concept with more demonstrations, then the model identifies and applies that concept through marginalization. ICL can thus be implicit Bayesian inference. All context, query, and outcomes are texts in language modeling. In inverse molecular design, we have molecule-score pairs as demonstrations. They capture the latent task concept, with semantics like the task descriptions in Figure 1. $Q$ is the target score, and $X$ is the molecule to be designed. We focus on a new molecular foundation model for ICL.

## 2.2 MOLECULAR DESIGN WITH GRAPH DIFFUSION TRANSFORMERS

Molecules are discrete graphs $X = (A, B)$, where $A$ denotes the set of atoms and $B$ the set of bonds. These structures are commonly modeled using discrete diffusion processes (Vignac et al., 2022; Liu et al., 2024c). Graph DiTs concatenate atom and bond features to $X$ into the input format of standard Transformers. Given $X$, for each atom $a_i \in A$ with $d_i$ neighbors, Graph DiTs define a token as $x = \{a_i, \{b_{ij}\}_{j=1}^{d_i}\}$, where $b_{ij} \in B$ encodes the bond type (single, double, triple, or none). Each token is represented by a feature vector $\mathbf{x} \in \mathbb{R}^F$, formed by concatenating the one-hot encoding of the atom type and the connection types to all other atoms (either a bond type or a null type indicating no connection). Discrete diffusion has a transition matrix $\mathbf{Q}$, initialized based on the frequency of atoms and bonds in the training set. At step $t$, $[\mathbf{Q}^t]_{ij} = q(\mathbf{x}_j^t \mid \mathbf{x}_i^{t-1})$ for $i, j \in [1, F]$.

The forward diffusion with $\mathbf{Q}$ is: $q(\mathbf{x}^t \mid \mathbf{x}^{t-1}) = \mathrm{Cat}(\mathbf{x}_t; \mathbf{p} = \mathbf{x}^{t-1}\mathbf{Q}^t)$, where $\mathrm{Cat}(\mathbf{x}; \mathbf{p})$ denotes two separate categorical sampling for atoms and bonds with probabilities from $\mathbf{p}$. Starting from the original data $\mathbf{x} = \mathbf{x}^0$, we have $q(\mathbf{x}^t \mid \mathbf{x}^0) = \mathrm{Cat}\left(\mathbf{x}^t; \mathbf{p} = \mathbf{x}^0\bar{\mathbf{Q}}^t\right)$, where $\bar{\mathbf{Q}}^t = \prod_{i \leq t} \mathbf{Q}^i$. The forward diffusion gradually corrupts data points. When the total timestep $T$ is large enough, $q(\mathbf{x}^T)$ converges to a stationary distribution. The reverse process samples from $q(\mathbf{x}^T)$ and gradually removes noise. The posterior distribution $q(\mathbf{x}^{t-1} \mid \mathbf{x}^t)$ is calculated as $q(\mathbf{x}^{t-1}|\mathbf{x}^t, \mathbf{x}^0) \propto \mathbf{x}^t(\mathbf{Q}^t)^\top \odot \mathbf{x}^0\bar{\mathbf{Q}}^{t-1}$. Given multiple properties $\{c_i\}_{i=1}^n$, the denoising model approximates $p_\phi(\mathbf{x}^{t-1} \mid \mathbf{x}^t, \mathbf{x}^0, \{c_i\}_{i=1}^n)$ under property conditions. This model is trained by minimizing the negative log-likelihood for $\mathbf{x}^0$:

$$\mathcal{L}_{\mathrm{DM}} = \mathbb{E}_{q(\mathbf{x}^0)}\mathbb{E}_{q(\mathbf{x}^t|\mathbf{x}^0)} \left[ -\log p_\phi\left(\mathbf{x}^0 \mid \mathbf{x}^t, c_1, c_2, \ldots, c_n\right) \right], \tag{2}$$

In Graph DiTs (Liu et al., 2024c), the constraints $\{c_i\}_{i=1}^n$ are numerical or categorical property values. In this work, we explore them as demonstrations for in-context learning.

## 3 LEARNING DIFFUSION MODEL WITH DEMONSTRATIONS

Figure 2 shows the generation process of DemoDiff, combining motif-based representation (Section 3.1) with graph diffusion transformers for in-context molecular generation (Section 3.2).

### 3.1 MOLECULAR GRAPH TOKENIZATION WITH NODE PAIR ENCODING

More demonstrations help capture the latent task concept and are empirically useful (Bertsch et al., 2024). However, prior work (Liu et al., 2024c) uses atom-level molecular representations, similar to modeling text at the character level, which fundamentally limits the number of examples in context. For efficient representation, we merge frequent sub-molecular patterns as *motif* $m = (\tilde{A}, \tilde{B}) \subseteq X$, where $\tilde{A} \subseteq A$ and $\tilde{B} \subseteq B$ define a connected substructure. A molecule becomes a collection of disjoint motifs $M = \{m_i\}_{i=1}^n$ such that: (1) $\tilde{A}_i \cap \tilde{A}_j = \emptyset$ for all $i \neq j$; (2) $\bigcup_{i=1}^n \tilde{A}_i = A$. Then, each edge $e_{ij} \in E$ is directed from a source motif $m_i$ to a target motif $m_j$, with two associated attributes: (1) bond type, and (2) attachment index, indicating the atom within $m_i$ from which the bond originates. This abstraction induces a tokenizer with two functions. They are $\mathrm{tokenizer.encode} : X = (A, B) \longmapsto \hat{X} = (M, E)$ that compresses the atom-level graph into motif-level form and $\mathrm{tokenizer.decode} : \hat{X} = (M, E) \longmapsto X = (A, B)$ for reconstruction. The tokenizer uses two vocabularies: $\mathcal{M}$ for motif types and $\mathcal{E}$ for edge types. It starts with the 118 atom

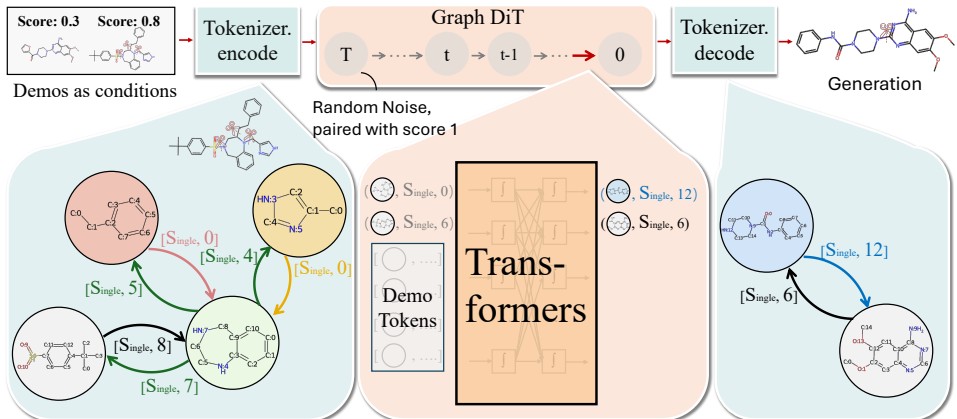

Figure 2: Demonstration-conditioned diffusion generation. In the reverse process, DemoDiff starts from random noise and denoises molecules conditioned on molecule–score demonstration pairs. A tokenizer encodes each molecule into a motif-level representation. Motifs are connected by edges with two attributes shown in square brackets: (1) bond type and (2) attachment index. The input to the transformers consists of tokens from the denoised molecules and the context of demo tokens.

types from the periodic table and one "*" for the polymerization point, which form the initial motifs in $\mathcal{M}$. This guarantees that, in the worst case, a new molecule can still be represented at the atom level. In each iteration, the tokenizer merges the most frequent neighbors until no further merge is found in $\mathcal{M}$, then proceeds with the corresponding connections between motifs.

To construct $\mathcal{M}$, existing methods, such as BRICS (Degen et al., 2008) or molecular grammars (Sun et al., 2025; 2024), rely on domain-specific heuristics based on chemical reactions or expert knowledge. The resulting vocabularies are independent of the pretraining data, often missing frequent motifs. To address this limitation, we propose *Node Pair Encoding* (NPE), a frequency-based algorithm for molecular graphs inspired by BPE, as outlined in Algorithm 1. We initialize $\mathcal{M}$ with elements from the periodic table and polymerization points "*". NPE iteratively performs three steps: (1) **Neighborhood merge**: For each molecule $X \in \mathcal{D}$ in the dataset $\mathcal{D}$ and current $\mathcal{M}$, we identify candidate motifs by merging adjacent substructures that appear in $\mathcal{M}$; (2) **Frequency selection**: The most frequent candidate motif is selected and added to $\mathcal{M}$; (3) **Graph update**: Each $X \in \mathcal{D}$ is updated by replacing instances of merged motif pairs with the new motif.

**Constrained NPE**: The standard NPE may produce multiple directed edges from a motif $m_i$ to another $m_j$ when decomposing ring structures (e.g., aromatic rings). It leads to ambiguity during decoding since $e_{ij}$ does not uniquely determine the attachment index within $m_j$. To address this, we introduce constraints such as rings into NPE at two stages. During initialization, we traverse each molecule to identify its set of maximal connected rings, denoted $\mathcal{R}$, compute their frequencies, and include the top-$K_{\text{ring}}$ most frequent rings in the initial vocabulary $\mathcal{M}$. During motif merging, any $m \in \mathcal{M}$ is merged with a ring $r \in \mathcal{R} \setminus \mathcal{M}$ as a complete unit, rather than merging individual atoms within $r$. This strategy integrates frequent rings into the vocabulary while preserving atom-level representations for rare rings, avoiding reconstruction ambiguity. We compare NPE with fingerprints (Rogers & Hahn, 2010) and virtual nodes (Hwang et al., 2022) in appendix A.

Using NPE, a molecule is represented as $n$ tokens $\{x_i\}_{i=1}^n$, where $x_i = \{m_i, \{e_{ij}\}_{j=1}^{d_i}\}$, with $m_i$ denoting a motif and $e_{ij}$ the associated edges. We set the motif vocabulary size to $K = 3000$ ($K_{\text{ring}} = 300$), with details and analysis provided in appendix C.2. As shown in Figure 2, an example input with 38 atoms can be compactly expressed using four motifs. An empirical comparison of atom- and motif-level representations over 1 million pretraining molecules is given in Figure 3b, with an average compression ratio of $5.446 \pm 2.569$, reducing the median count from 30 atoms to 5 motifs.

### 3.2 IN-CONTEXT LEARNING WITH GRAPH DIFFUSION TRANSFORMERS

We construct the dataset $\mathcal{D} = \{(\mathcal{C}_i, Q_i, X_i)\}_{i=1}^{N_{\text{pretrain}}}$ for pretraining, where each task consists of a context of molecule–score pairs $\mathcal{C}_i$, a query score $Q_i$, and a target molecule $X_i$. To pretrain DemoDiff

for in-context inverse molecular design, we replace the property conditions in Eq. (2) with $\mathcal{C}$ and $Q$:

$$\mathcal{L}_{\text{pretrain}} = \mathbb{E}_{q(\mathbf{x}^0)}\mathbb{E}_{q(\mathbf{x}^t|\mathbf{x}^0)}\left[-\log p_\theta\left(\mathbf{x}^0 \mid \mathbf{x}^t, \mathcal{C}, Q\right)\right]. \qquad (3)$$

With large and diverse pretraining data and scalable Transformers (Peebles & Xie, 2023), DemoDiff learns to infer the latent task concept to generate the target molecule and can serve as a foundation model for ICL. In a task, it performs implicit Bayesian inference over diffusion trajectories.

**ICL with Context Consistency (appendix B.4):** Given a query $Y$, demonstrations $\mathcal{C}_i$ are divided into positive, medium, and negative groups. For a generated molecule $X$, we compare its fingerprint-based similarity with these groups to assess whether it follows the relation $\text{pos} > \text{med} > \text{neg}$. This yields a consistency score that measures how well the generated molecule aligns with the relative relations in the context. In experiments (Section 4.1), we use this score to select high consistent generations before conducting the final evaluation with Oracles.

### 3.3 MODEL DESIGN AND PRETRAINING

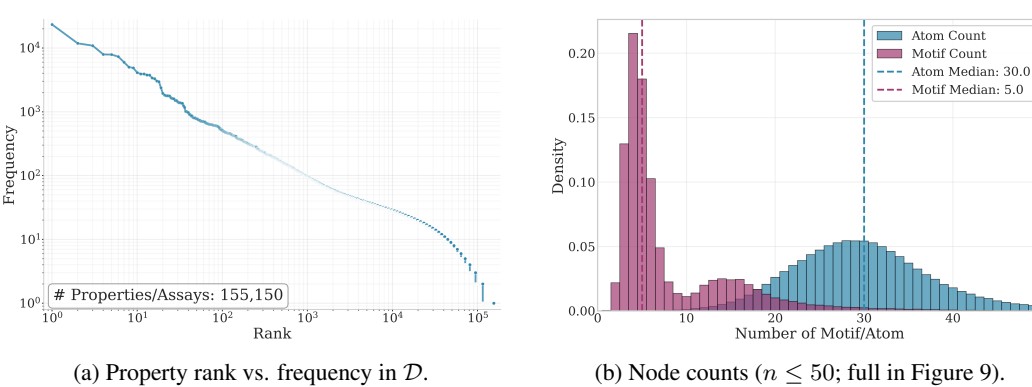

(a) Property rank vs. frequency in $\mathcal{D}$.  (b) Node counts ($n \le 50$; full in Figure 9).

Figure 3: Pretraining data statistics for property rank-frequency and node count density.

**Model Designs:** Figure 2 illustrates the model architecture. For the $i$-th task in the pretraining set $(\mathcal{C}_i, X_i, Y_i) \in \mathcal{D}$, the property score $Y_i$ is scaled within $[0, 1]$, providing positional signals for both demonstration molecules and the target. These scalar values are encoded using Rotary Position Embedding (RoPE) (Su et al., 2024). DemoDiff uses a tokenizer to process the atom-level representation and defines a maximum context length for the number of motif tokens. The context includes the target along with as many demonstration tokens as fit within the target length. Since molecules in the context are structurally disjoint, edge connectivity implicitly delineates context boundaries, removing the need for explicit delimiter tokens. Details are in appendix B.

**Pretraining:** To construct the pretraining dataset as illustrated in Section 3.2, we use the ChEMBL database (Zdrazil et al., 2024), the largest collection of biological assays, containing over 2.5 million molecules and 1.7 million assay records. To increase chemical diversity for materials discovery, we augment ChEMBL with polymer datasets from multiple sources (Liu et al., 2024b; Kuenneth et al., 2021), including properties such as thermal conductivity, free volume fraction, and glass transition temperature. For biological assays, we generate tasks by selecting a molecule-activity pair as the target and treating other molecule-activity pairs as context. The target is assigned a score of 1, and context scores are computed by normalizing differences in $p$ChEMBL values (negative log of bioactivity measures such as IC50 and potency) to the interval $[0, 1]$. We restrict targets to bioactive molecules with $p$ChEMBL $> 6$. For polymers, we apply the same strategy: each polymer is used as a target, and its property value is normalized against those of other polymers to form context-target pairs. We partition context examples into three groups by normalized scores: positive $[0.75, 1]$, medium $[0.5, 0.75)$, and negative $[0, 0.5)$, with up to 15 demonstrations from each. The final dataset comprises around 1 million molecules with 155K unique assays or properties, yielding 1.6 million tasks. As shown in Figure 3a, the frequency distribution of assays and properties follows Zipf's law, $P(Y_{\text{rank}}) \propto \text{rank}^{-1.13}$, consistent with patterns in language corpora. These 1 million molecules are used to initialize the motif vocabulary via NPE, which provides a more compact representation by reducing node counts (Figure 3b). We further extract edge connections to construct an edge

Table 1: We compute the oracle and diversity scores from the Top-10 generations and report the harmonic mean of these two values. We group 33 tasks into six categories and report the mean ± std within each category. The best results in each column are **bolded**. Task-specific results and additional metrics are provided in appendix D.2.

| Task | Drug Rediscovery | Drug MPO | Structure Constrained | Drug Design | Target Based | Material Design | Avg Rank | Total Sum |
|---|---|---|---|---|---|---|---|---|
| # Tasks | 7 | 7 | 5 | 4 | 5 | 5 | 33 | 33 |
| Molecular Optimization Methods with 100 Oracle Calls | | | | | | | | |
| GraphGA | 0.36±0.07 | 0.52±0.19 | 0.43±0.21 | 0.41±0.32 | 0.76±0.04 | 0.58±0.11 | 6.56 | 16.65 |
| REINVENT | 0.37±0.08 | 0.52±0.17 | 0.43±0.21 | 0.42±0.32 | 0.76±0.03 | 0.00±0.00 | 8.44 | 13.84 |
| GPBO | 0.37±0.07 | 0.51±0.18 | 0.42±0.22 | 0.39±0.33 | 0.76±0.03 | 0.60±0.21 | 6.89 | 16.65 |
| STONED | 0.36±0.07 | 0.52±0.19 | 0.39±0.24 | 0.37±0.36 | 0.76±0.04 | No pSMILES | 8.18 | 13.42 |
| Genetic GFN | 0.36±0.10 | 0.51±0.22 | 0.38±0.29 | 0.33±0.37 | 0.76±0.03 | No pSMILES | 9.28 | 13.04 |
| GenMol | 0.42±0.08 | 0.51±0.17 | 0.42±0.25 | 0.55±0.21 | 0.69±0.03 | 0.62±0.08 | 7.98 | 17.36 |
| Molecular Optimization Methods with 10000 Predictor Calls | | | | | | | | |
| GraphGA | 0.37±0.09 | 0.50±0.18 | 0.45±0.29 | 0.49±0.26 | 0.64±0.06 | 0.55±0.14 | 9.30 | 16.30 |
| REINVENT | 0.30±0.13 | 0.23±0.23 | 0.25±0.24 | 0.38±0.26 | 0.17±0.13 | 0.51±0.16 | 13.60 | 9.82 |
| GPBO | 0.33±0.10 | 0.45±0.22 | 0.43±0.29 | 0.49±0.15 | 0.74±0.03 | 0.42±0.24 | 9.87 | 15.35 |
| STONED | 0.33±0.10 | 0.40±0.20 | 0.50±0.28 | 0.27±0.10 | 0.20±0.27 | No pSMILES | 12.01 | 9.66 |
| Genetic GFN | 0.14±0.08 | 0.08±0.16 | 0.08±0.11 | 0.12±0.11 | 0.15±0.15 | No pSMILES | 17.95 | 3.14 |
| GenMol | 0.34±0.07 | 0.28±0.23 | 0.21±0.29 | 0.39±0.19 | 0.52±0.05 | 0.21±0.23 | 12.63 | 10.37 |
| Conditional Generation Models | | | | | | | | |
| LSTM | 0.39±0.25 | 0.16±0.07 | 0.55±0.32 | 0.33±0.35 | 0.72±0.04 | 0.16±0.11 | 11.77 | 12.30 |
| Graph-DiT | 0.43±0.21 | 0.50±0.18 | **0.58±0.34** | 0.48±0.37 | 0.71±0.04 | 0.55±0.17 | 8.53 | 17.64 |
| Learning from In-Context Demonstrations | | | | | | | | |
| DeepSeek-V3 | 0.45±0.18 | 0.51±0.20 | 0.49±0.24 | 0.65±0.18 | 0.64±0.06 | 0.39±0.24 | 8.08 | 16.90 |
| GPT-4o | **0.47±0.21** | 0.53±0.20 | 0.52±0.30 | 0.48±0.40 | 0.73±0.05 | 0.43±0.16 | 7.89 | 17.25 |
| Qwen-Max | 0.15±0.21 | 0.17±0.15 | 0.32±0.32 | 0.29±0.29 | 0.19±0.26 | 0.10±0.18 | 15.39 | 6.46 |
| Llama3.1-8B-FT | 0.21±0.13 | 0.24±0.23 | 0.24±0.34 | 0.31±0.31 | 0.02±0.05 | 0.29±0.40 | 14.91 | 6.04 |
| Qwen3-8B-FT | 0.37±0.19 | 0.27±0.16 | 0.26±0.34 | 0.46±0.22 | 0.67±0.07 | 0.44±0.39 | 10.96 | 12.23 |
| DemoDiff (Ours) | 0.44±0.21 | **0.54±0.23** | 0.56±0.33 | **0.79±0.11** | **0.78±0.05** | **0.67±0.11** | **4.10** | **20.10** |

vocabulary capturing motif-to-motif connectivity. Finally, we pretrain a DemoDiff model with 0.7B parameter on Eq. (3), using 146 H100 GPU days. Details are provided in appendix C.

## 4 EXPERIMENT

**Setups:** We curate 33 downstream tasks (see Table 1 and appendix D.1) across six categories to evaluate DemoDiff against 19 baselines. These tasks are primarily curated by domain experts and are distinct from pretraining. We include eight molecular optimization methods and two conditional generation models (LSTMs and Graph DiT (Liu et al., 2024c)), and LLMs (DeepSeek-V3, GPT-4o, and Qwen-Max). We select the top four molecular optimization algorithms from the PMO benchmark (Gao et al., 2022) (out of 25 evaluated methods) and two recent methods, Genetic GFN (Kim et al., 2024) and GenMol (Lee et al., 2025), under two settings: 100 oracle calls and 10,000 predictor calls. For evaluation, we generate 10 valid, unique, and novel molecules per task and score them with Oracles. We report the harmonic mean over two dimensions: (a) averaged oracle scores and (b) the diversity score Eq. (8). Each task has up to 450 molecule–score pairs, evenly divided into positive $[0.75, 1]$, medium $[0.5, 0.75)$, and negative $[0, 0.5)$ groups. Each task has an Oracle for evaluation. We use all molecules to train the task-specific predictor for predictor calls or to train conditional generation models directly. For LLMs with ICL, we include three closed-sourced models. We finetune Llama3.1-8B-FT and Qwen3-8B-FT on the DemoDiff pretraining set. Their demonstrations are randomly sampled with a similar budget for context. For the baselines, we follow their standard and original implementations to process the molecules.

### 4.1 PERFORMANCE ON DIVERSE MOLECULAR DESIGNS TASKS

**ICL achieves competitive performance with minimal supervision.** Table 1 compares the harmonic mean of the top-10 generated molecules based on both task scores and structural diversity, while Table 5 (appendix) reports the top-1 scoring molecule per task. Under limited data and Oracle budgets, ICL methods perform comparably to, or better than, fully trained conditional generators and molecular optimization baselines. Among ICL-based methods, DeepSeek-V3, GPT-4o, and DemoDiff consistently attain top-tier average ranks. These ICL methods rely on tens of demonstrations per task, significantly fewer than the training data or Oracle calls required by other models or algorithms.

**DemoDiff designs molecules with accurate scores and high diversity.** Across six task categories, it performs best on property-driven tasks, including drug design with bioactivity targets, protein binding affinity, and material design for polymer gas separation. DemoDiff achieves the lowest average rank of 4.10, outperforming the best baseline, GraphGA (rank 6.56). ICL methods with LLMs produce high-scoring top designs (Table 5) but often generate structurally similar molecules. These do not necessarily align better with the target score while reducing diversity. In contrast, DemoDiff designs molecules with scores closer to the query and better structural diversity.

**DemoDiff performs better on property-driven tasks than on structure-constrained ones.** It scores 0.67–0.79 on drug and material design, but around 0.44–0.56 for rediscovery and structure-constrained tasks, where Oracle scoring is tied to the presence of specific structures. While DemoDiff still ranks highly in structure-constrained tasks, its stronger results on property-driven tasks highlight its advantage in exploring chemical spaces with broader solution ranges.

## 4.2 ABLATION STUDIES AND PERFORMANCE ANALYSIS

Table 2: Performance across model sizes using harmonic mean scores from Top-10 generations

| DemoDiff | Drug Rediscovery | Drug MPO | Structure Constrained | Drug Design | Target Based | Material Design |
|---|---|---|---|---|---|---|
| 78M | 0.39 ± 0.17 | 0.46 ± 0.24 | 0.47 ± 0.27 | 0.57 ± 0.31 | 0.73 ± 0.03 | 0.62 ± 0.13 |
| 311M | 0.40 ± 0.17 | 0.46 ± 0.23 | 0.50 ± 0.28 | 0.53 ± 0.27 | 0.75 ± 0.04 | 0.62 ± 0.14 |
| 739M | 0.44 ± 0.21 | 0.54 ± 0.23 | 0.56 ± 0.33 | 0.79 ± 0.11 | 0.78 ± 0.05 | 0.67 ± 0.11 |

**Model Parameters:** We pretrain DemoDiff with varying sizes: small (78.7M), medium (311M), and large (739M) parameters. Table 2 reports performance using the top-10 harmonic means of task score and diversity. We present averages with deviations across six categories. DemoDiff achieves reasonable scores even at small scale. At the medium scale, performance improves in most tasks except drug design, while the benefits of parameter scaling become more evident (in five out of six task categories) at larger scales

**ICL with Demonstrations:** Figure 4 studies two factors in demonstrations: (1) context length and (2) ratio of positive examples. In Figure 4a, longer context includes more molecular examples and supports better ICL performance. This aligns with the rationale of motif-level tokenization, which captures more examples within a fixed context. Figure 4b shows that diverse demonstrations are important for ICL to represent the task accurately, while only positive examples are insufficient. This is because positive, medium, and negative examples together provide a holistic view of the task context, and DemoDiff pretrained on such contexts is better able to infer latent concepts from diverse examples. In Figure 4b, we also observe that fewer positive examples may still yield reasonable results. We investigate this further in Section 4.3 to assess whether DemoDiff can infer positive examples (score > 0.5) using only negative examples with scores below 0.5.

**ICL with Consistency Scores:** We ablate consistency scores and analyze their correlation with target scores in Figure 5. Using the consistency score as a confidence filter improves performance across task categories, with gains from 0.8% to 27.5%. The second figure shows the correlation between the consistency and target scores. Moderate correlation appears in tasks with explicit structural constraints, such as drug rediscovery and structure-constrained design. For property-driven tasks (drug MPO and materials design), high fingerprint-based consistency with positive examples does not always correlate with high target scores. In these cases, latent concepts may rely on subtle substructures (e.g., methyl groups (Liu et al., 2022)) that standard fingerprints fail to capture. Interestingly, context consistency still improves performance in these tasks. A possible reason is that the score helps filter out false positive generations.

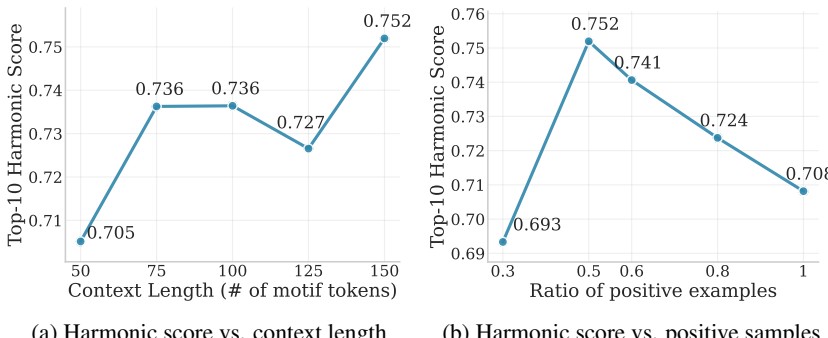

(a) Harmonic score vs. context length  (b) Harmonic score vs. positive samples

Figure 4: Ablation studies on Albuterol drug rediscovery.

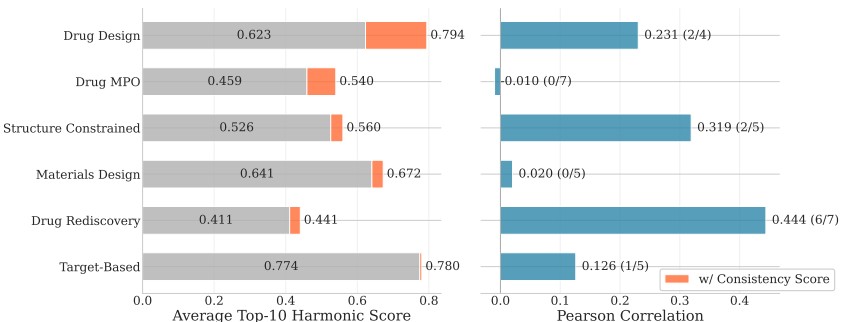

Figure 5: Ablation studies on context consistency scores: (1) left shows improvements; (2) right shows the relationship between consistency score and oracle scores.

## 4.3 CASE STUDIES

Figures 6 and 7 present two studies for DemoDiff. In extreme cases of inverse molecular design, demonstration sets may contain only negative examples, i.e., all scores $< 0.5$. In Figure 6, we study whether DemoDiff can still generate positive candidates when prompted solely with negative examples. Figure 6 presents the results for (a) structure-constrained design, (b) drug multi-objective optimization (MPO), and (c) target-based design. These findings suggest two insights: (1) negative demonstrations convey informative signals about the task concept, and (2) after pretraining, the posterior over the concept-to-structure mapping allows DemoDiff to generate desirable candidates that are aligned with the concept yet structurally distinct from the negative examples. Figure 7 is the generation trajectory from diffusion models. The task score, measured as structural similarity to Albuterol, rises from 0.22 at initial sampling to 0.74. This shows that the diffusion model refines the molecule toward the desired structure step by step with demonstrations.

## 5 RELATED WORK

**Inverse Molecular Design:** Molecular optimization uses diverse approaches, including genetic algorithms, Monte Carlo Tree Search (Jensen, 2019), and Bayesian optimization (Shahriari et al., 2015), applied to representations such as fingerprints, SMILES, graphs, and synthetic pathways (Gao et al., 2021). Gao et al. (2022) benchmarked 25 optimization methods and found that older models, such as genetic algorithms, remain competitive. However, existing benchmarks require on the order of 10,000 oracle calls, which is costly and limits applicability when single calls are expensive. Deep learning models offer an alternative by modeling the joint distribution of atoms and bonds without Oracle calls. GDSS applies noise and denoising in continuous space for graphs (Jo et al., 2022). DiGress (Vignac et al., 2022) introduces discrete noise through transition matrices based on marginal atom and bond distributions. Graph DiTs (Liu et al., 2024c) extend scalable diffusion transformers (Peebles & Xie, 2023) to discrete graphs. Yet, training diffusion models still requires hundreds of labeled molecules and is limited to specific tasks. Recent efforts explore chemical

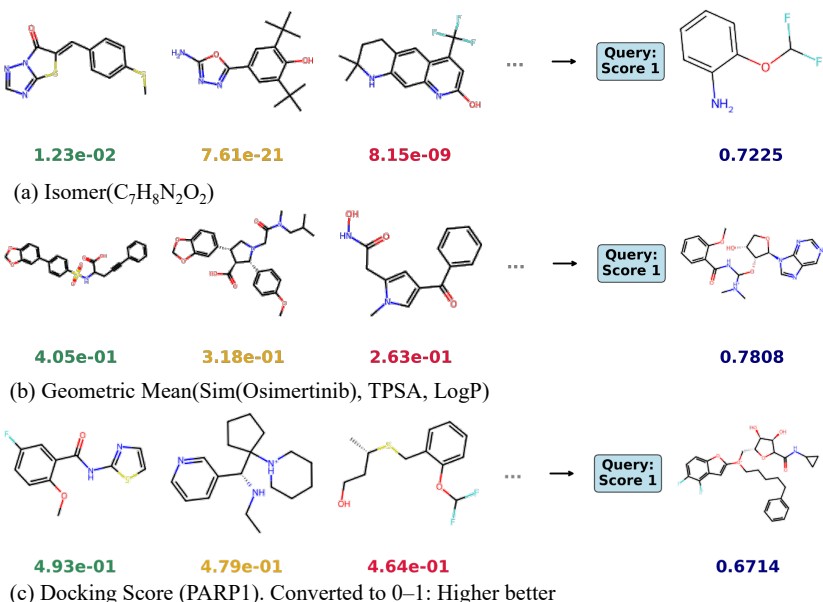

1.23e-02     7.61e-21     8.15e-09        0.7225

(a) Isomer($C_7H_8N_2O_2$)

4.05e-01     3.18e-01     2.63e-01        0.7808

(b) Geometric Mean(Sim(Osimertinib), TPSA, LogP)

4.93e-01     4.79e-01     4.64e-01        0.6714

(c) Docking Score (PARP1). Converted to 0–1: Higher better

Figure 6: Learning from negative demonstrations (score $< 0.5$) to infer a target with score 1. All demonstrations are shown in Figures 14 to 16, with only three displayed here.

Step1: 0.22    Step 100: 0.20    Step 200: 0.32    Step 300: 0.41    Step 400: 0.59    Step 500: 0.74

Figure 7: Diffusion trajectory for Albuterol drug rediscovery: we sample five intermediate diffusion steps and score them with the Albuterol Oracle, which computes similarity to the ground truth.

foundation models based on LLMs (Yu et al., 2024; Liu et al., 2024b), but their applications are either diverted to other molecular tasks, such as property prediction, or rely on fine-tuning within a restricted scope of design tasks.

**In-Context Learning:** ICL is an emergent ability observed in LLMs (Brown et al., 2020; Chan et al., 2022). Empirical and theoretical studies investigate this phenomenon from three perspectives: models, data, and learning mechanisms (Xie et al., 2021; Min et al., 2022). For the learning mechanism, ICL can be interpreted as implicit Bayesian inference (Xie et al., 2021), where pretraining data are generated from latent concepts and the posterior distribution marginalizes over them for inference. On the model side, Garg et al. (2022) trained Transformers from scratch on prompt-style input–label pairs of simple functions and found performance comparable to task-specific algorithms. Bhattamishra et al. (2023) compared Transformers with attention-free models and showed they do not match Transformer performance across tasks. On the data side, Chan et al. (2022) found that Transformers outperform recurrent models (e.g., LSTMs) on data with distributional properties resembling natural language, such as burstiness (words appearing in clusters) and query tasks with many rare classes. Singh et al. (2025) analyzed the strategy competition between ICL and in-weight learning, showing that the asymptotic strategy depends on in-weight information but is also context-constrained. This aligns with (Chan et al., 2022), suggesting that a foundation model should support both capacities. A skewed Zipfian distribution over tasks (e.g., Figure 3a) balances learning by storing common task information in weights while developing ICL ability from the long tail of rare tasks. Besides LLMs, ICL is also studied on graphs (Huang et al., 2023), but mainly for node and edge classification tasks.

## 6 CONCLUSION

We presented DemoDiff, a demonstration-conditioned diffusion Transformer model for in-context molecular design. We constructed a large-scale pretraining dataset with over one million molecules and 155K unique biological assays and material properties, yielding millions of demonstration–target pairs. Using this dataset, we pretrained a 0.7B-parameter model and showed that it matches or outperforms much larger LLMs and ranks higher than domain- and task-specific methods. To support scalable pretraining, we introduce Node Pair Encoding, a motif-level graph tokenizer that efficiently represents molecules with fewer nodes while preserving reconstruction. Experiments demonstrate that DemoDiff is a promising molecular foundation model, highlighting its potential to scale further with larger models, broader datasets, and greater compute.

## ACKNOWLEDGMENTS

This work was partially supported by NSF IIS-2142827, IIS-2146761, IIS-2234058, and CBET-2332270. We also appreciate the support from the Foundation Models and Applications Lab of Lucy Institute, ND-IBM Tech Ethics Lab, and OpenAI Researcher Access Program.

## REPRODUCIBILITY STATEMENT

We provide pretraining and inference code in the supplementary materials. The appendix describes the method and model settings to ensure reproducibility.

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

## A    DISCUSSION ON NODE PAIR ENCODINGS AND RELATED WORK ON FINGERPRINTS AND VIRTUAL NODES

The node pair encodings (NPE) differ from fingerprints (Rogers & Hahn, 2010) and virtual nodes (Hwang et al., 2022) in three aspects: (1) representation and invertibility, (2) computation process, and (3) use cases. Specifically: (1) NPE builds a motif vocabulary from data and encodes a molecule as a graph of motifs with typed edges and attachment rules, allowing the atom-level structure to be reconstructed. In contrast, Morgan fingerprints hash local atom neighborhoods into a fixed-length bit vector, which is lossy and not invertible. Virtual nodes are additional nodes inserted into graphs to provide shortcuts for information exchange between nodes rather than forming a new molecular representation. If we use only virtual nodes to re-represent the molecule, the structure is not invertible. (2) NPE identifies frequent substructures from a dataset to build a discrete motif vocabulary. In contrast, Morgan fingerprints are computed directly from atom invariants with a chosen radius and bit size. Virtual nodes can be added randomly or through generic graph partitioning algorithms, which do not reflect how frequently a partition or cluster appears across the molecular dataset. (3) NPE produces a compact motif-level graph suited for generative tasks. In contrast, Morgan fingerprints produce a vector suited for predictive tasks. Virtual nodes are used to create shortcuts for sharing information between graph nodes.

## B    DETAILS ON DEMODIFF

### B.1    ICL WITH DEMODIFF

We construct the pretraining dataset as $\mathcal{D} = \{(\mathcal{C}_i, Q_i, X_i)\}_{i=1}^{N_{\text{pretrain}}}$. Each task contains a context of molecule–score pairs $\mathcal{C}_i$, a query score $Q_i$, and a target molecule $X_i$. For each task, we first sample a latent concept (e.g., the thermal conductivity of polymeric materials from Figure 1) from a prior $\theta \sim p(\theta)$, followed by a set of molecule–property pairs associated with that concept. This induces the pretraining distribution:

$$p(\mathcal{C}, Q, X) = \int_\theta p(\theta)\, p(\mathcal{C} \mid \theta)\, p(Q \mid \mathcal{C}, \theta) \left[ \int_{\mathbf{x}^{1:T}} p(\mathbf{x}^T) \prod_{t=1}^{T} p_\theta(\mathbf{x}^{t-1} \mid \mathbf{x}^t, Q, \mathcal{C})\, d\mathbf{x}^{1:T} \right] d\theta, \quad (4)$$

where $\mathbf{x}^0$ represents the feature of the target token. With sufficiently large and diverse pretraining data and scalable diffusion transformers (Peebles & Xie, 2023), DemoDiff can a foundation model for ICL. For a downstream task, we use $(\mathcal{C}, Q)$ as the prompt, and $p(\theta \mid \mathcal{C}, Q)$ form the prompt concept. Then DemoDiff marginalizes the concept over the diffusion trajectories $\mathbf{x}^{1:T}$:

$$p(X \mid \mathcal{C}, Q) = \int_{\theta \in \Theta} \int_{\mathbf{x}^{1:T}} \left[ p(\mathbf{x}^T) \prod_{t=1}^{T} p_\theta(\mathbf{x}^{t-1} \mid \mathbf{x}^t, \mathcal{C}, Q) \right] p(\theta \mid \mathcal{C}, Q)\, d\mathbf{x}^{1:T}\, d\theta. \quad (5)$$

### B.2    GRAPH-LEVEL TOKENS AND PRETRAINING LOSS

Graph DiTs define a graph-level token that concatenates the node feature with all related edge features. In molecular generation, we use a special type of edge, the null edge, to represent that there is no edge between two nodes. Thus, The feature vector $\mathbf{x}$ (or $\mathbf{x}^0$) of a graph-level token $x = \{M, \{e_j\}_{j=1}^d\}$ consists of three components: $F_{\text{motif}}$ motif types, $F_{\text{bond}}$ bond types, and $F_{\text{attach}}$ attachment specifications. Here, $F_{\text{motif}}$ is the size of the motif vocabulary $\mathcal{M}$, $F_{\text{bond}} = 4$ represents null, single, double, and triple bonds, and $F_{\text{attach}} = \arg\max_{M \in \mathcal{M}} |M|$ is the maximum number of

atoms in $\mathcal{M}$. Eq. (3) can be decomposed as $\mathcal{L}_{\text{pretrain}} = \mathcal{L}_{\text{motif}} + \mathcal{L}_{\text{bond}} + \mathcal{L}_{\text{attach}}$. Specifically,

$$\mathcal{L}_{\text{pretrain}} = \mathbb{E}_{q(\mathbf{x})} \mathbb{E}_{q(\mathbf{x}^t|\mathbf{x})} \left[ \underbrace{- \log p_\theta^{\text{motif}}(\mathbf{m} \mid \mathbf{x}^t, \mathcal{C}, Y)}_{\mathcal{L}_{\text{motif}}} \right.$$

$$\underbrace{- \sum_{j=1}^{d} \log p_\theta^{\text{bond}}(\mathbf{b}_j \mid \mathbf{x}^t, \mathcal{C}, Y)}_{\mathcal{L}_{\text{bond}}}$$

$$\left. \underbrace{- \sum_{j=1}^{d} \log p_\theta^{\text{attach}}(\mathbf{a}_j \mid \mathbf{x}^t, \mathcal{C}, Y)}_{\mathcal{L}_{\text{attach}}} \right]. \tag{6}$$

To align feature dimensions across tokens, we use the dense edge representation by treating all non-connections as null bonds. The resulting feature dimension is $F = F_{\text{motif}} + n \times F_{\text{bond}} + n \times F_{\text{attach}}$, where $n$ is the maximum number of nodes in the motif-represented dataset. For optimization with Eq. (6), we include the null bond type in $\mathcal{L}_{\text{bond}}$ but exclude attachment specifications of null edges in $\mathcal{L}_{\text{attach}}$.

## B.3 TRANSITION MATRICES IN DIFFUSION MODELS

We define the transition matrix $\mathbf{Q}$ that perturbs molecules at the motif level to pretrain DemoDiff. We model the joint distribution of nodes and edges with the transition matrix. It is constructed from four submatrices $\mathbf{Q}_V, \mathbf{Q}_{EV}, \mathbf{Q}_E, \mathbf{Q}_{VE}$, denoting transitions node $\rightarrow$ node, edge $\rightarrow$ node, edge $\rightarrow$ edge, and node $\rightarrow$ edge, respectively:

$$\mathbf{Q}_G = \begin{bmatrix} \mathbf{Q}_V & \mathbf{1}'_N \otimes \mathbf{Q}_{VE} \\ \mathbf{1}_n \otimes \mathbf{Q}_{EV} & \mathbf{1}_{n \times n} \otimes \mathbf{Q}_E \end{bmatrix}, \tag{7}$$

where $\otimes$ denotes the Kronecker product, and $\mathbf{1}_N, \mathbf{1}'_n$, and $\mathbf{1}_{n \times n}$ are the column vector, row vector, and all-ones matrix, respectively. Here $n$ is the number of nodes. For edges, diffusion is applied to bond types only, while attachment attributes are optimized and predicted directly by the denoising model as in Eq. (6). For categorical sampling, we separate the unnormalized logits of node and edge from the model outputs, compute probabilities for each motif and bond individually before sampling.

To obtain the transition matrices, we use the prior from the pretraining data. The noisy distribution is defined as the marginal distributions of motif types $\mathbf{m}_V$ and bond types $\mathbf{m}_E$. The transition matrices are defined as $\mathbf{Q}_V = \bar{\alpha}^t \mathbf{I} + (1 - \bar{\alpha}^t) \mathbf{1} \mathbf{m}'_V$ and $\mathbf{Q}_E = \bar{\alpha}^t \mathbf{I} + (1 - \bar{\alpha}^t) \mathbf{1} \mathbf{m}'_E$, where $\mathbf{m}'$ denotes the transpose and $\mathbf{I}$ is the identity matrix. We compute co-occurrence frequencies of motif and bond types in training graphs to obtain the marginal distributions $\mathbf{m}_{EV}$ and $\mathbf{m}_{VE}$. Each row in $\mathbf{m}_{EV}$ gives the probability of co-occurring motifs for a bond type, and $\mathbf{m}_{VE}$ is its transpose. The transition matrices are then defined as $\mathbf{Q}_{EV} = \bar{\alpha}^t \mathbf{I} + (1 - \bar{\alpha}^t) \mathbf{1} \mathbf{m}'_{EV}$ and $\mathbf{Q}_{VE} = \bar{\alpha}^t \mathbf{I} + (1 - \bar{\alpha}^t) \mathbf{1} \mathbf{m}'_{VE}$, where $\bar{\alpha}^t$ is cumulative noise coefficient in diffusion. The cosine schedule is chosen as $\bar{\alpha}^t = \cos(0.5\pi(t/T + s)/(1 + s))^2$.

## B.4 DETAILS ON CONSISTENCY SCORE

Given a query $Y$, demonstrations $\mathcal{C}_i = \{(X_{ij}, Y_{ij})\}_{j=1}^{L}$ are divided into positive $\mathcal{C}^{\text{pos}}$, medium $\mathcal{C}^{\text{med}}$, and negative $\mathcal{C}^{\text{neg}}$ examples to guide ICL. For a generated molecule $X$, we use the Tanimoto similarity of fingerprints as the similarity measure. We compute the similarity between $X$ and all molecules in each group and average them to obtain group-wise similarity scores

$$\text{sim}^{\text{pos}}, \ \text{sim}^{\text{med}}, \ \text{sim}^{\text{neg}} \in [0, 1].$$

**Difference-based score.** We compute margin differences between groups:

$$d_{\text{pos,med}} = \max(\text{sim}^{\text{pos}} - \text{sim}^{\text{med}}, 0), \quad d_{\text{med,neg}} = \max(\text{sim}^{\text{med}} - \text{sim}^{\text{neg}}, 0),$$

$$d_{\text{pos,neg}} = \max(\text{sim}^{\text{pos}} - \text{sim}^{\text{neg}}, 0).$$

The normalized difference-based score is

$$s_{\text{diff}} = \min\left(\frac{d_{\text{pos,med}} + d_{\text{med,neg}} + d_{\text{pos,neg}}}{3}, 1\right).$$

In experiments, the consistency score can be computed efficiently before applying Oracle functions. For example, we generate 1000 molecules and select the top 100 with the highest consistency scores. These molecules better follow the order of structural similarity across positive, medium, and negative examples. This removes poor generations that conflict with the demonstration semantics and increases confidence that selected molecules align with the query scores. Table 3 reports empirical improvements across task categories, each containing 4–7 tasks (appendix D.1).

Table 3: Improvement with the consistency score (average Top-10 harmonic scores).

| Category | Without | With | Improvement |
|---|---|---|---|
| Drug Design | 0.6230 | 0.7943 | +27.5% |
| Drug MPO | 0.4592 | 0.5400 | +17.6% |
| Drug Rediscovery | 0.4110 | 0.4407 | +7.2% |
| Structure Constrained | 0.5258 | 0.5598 | +6.5% |
| Materials Design | 0.6407 | 0.6724 | +4.9% |
| Target-Based | 0.7745 | 0.7803 | +0.8% |

## C DETAILS ON PRETRAINING

The final pretraining dataset contains 1,084,566 molecules (polymers) and 155,150 unique assays or properties, yielding 1,639,515 tasks. These are constructed from ChEMBL (Zdrazil et al., 2024) and multiple polymer data sources (Otsuka et al., 2011; Thornton et al., 2012; Kuenneth et al., 2021). Each task has a query molecule–score pair with the query score fixed at 1. Up to 45 molecules are grouped into positive, medium, and negative demonstrations based on their scores. The query molecule is the target, while the query score and demonstrations serve as inputs to DemoDiff during pretraining on Eq. (3). For pretraining with a fixed maximum context window, we allocate half the window to positive demonstrations and one quarter each to medium and negative demonstrations, after excluding the target molecule.

### C.1 PRETRAINING DATASET

**ChEMBL dataset** We constructed molecular activity contexts from the ChEMBL database (version 35), which provides a large collection of bioactivity measurements across diverse assays. ChEMBL standardizes published activity types, values, and units into a unified variable, $p\text{ChEMBL} = -\log(\text{molar IC50, XC50, EC50, AC50}, K_i, K_d, \text{or Potency})$. This value places different measures of half-maximal response, potency, or affinity on a comparable negative logarithmic scale. For example, an IC50 of 1 nanomolar $(1 \times 10^{-9} \text{ M})$ corresponds to a $p\text{ChEMBL}$ value of 9. We extracted assay-level activity values ($p\text{ChEMBL}$). For each assay, molecules were grouped according to their recorded activities. Within each group, we selected anchor molecules with strong activity ($p\text{ChEMBL} > 6$) as targets for building demonstrations. Each anchor was compared against all other molecules in the same assay to compute normalized distances, defined as the relative difference between the anchor's $p\text{ChEMBL}$ value and that of the candidate molecule, converted to the range $[0, 1]$. Specifically, for an anchor with value $v_a$ and a candidate with value $v_c$, the normalized distance was given by $d = (v_1 - v_c)/10$. Based on this distance, we partitioned candidate molecules into three categories relative to the anchor. Molecules with distances in $[0, 0.25)$ correspond to candidates with activity between 75% and 100% of the anchor and were assigned to the positive context. Molecules with distances in $[0.25, 0.5)$ correspond to candidates with activity between 50% and 75% of the anchor and were assigned to the medium context. Molecules with distances $[0.5, 1.0]$ correspond to candidates with activity below 50% of the anchor and were assigned to the negative context. From each category, we sampled up to 15 molecules to balance neighborhood size. Thus, there are up to

45 demonstration molecules for each task. Not all of them are used during pretraining due to the constraint of maximum context length. This procedure produced triplets of anchor molecules and their associated positive, medium, and negative contexts.

**Polymeric materials** We have polymeric material datasets from different sources, including Poly-Info (Otsuka et al., 2011), MSA (Thornton et al., 2012), and from (Kuenneth et al., 2021). We considered a wide range of polymer properties spanning several categories, including thermal properties (e.g., heat capacity, glass transition temperature, melting temperature, and thermal conductivity), electronic properties (e.g., ionization energy, electron affinity, and band gap), structural properties (e.g., density, crystallinity, and radius of gyration), and transport properties (e.g., gas diffusion, solubility, and permeability coefficients). For each property, raw values were normalized to the unit interval using min–max scaling, with logarithmic transformation applied when dynamic ranges exceeded 1000 and non-negative shifts applied when necessary. Each polymer with valid property values was treated as an anchor, and pairwise distances in normalized property space were computed against all other polymers. Candidate molecules were partitioned into positive $[0, 0.25)$, medium $[0.25, 0.5)$, and negative $[0.5, 1]$ contexts, with up to 15 examples sampled per category based on smallest distances.

## C.2 TOKENIZER PREPARATION

---
**Algorithm 1** Node Pair Encoding (NPE) with Constraints

---
**Require:** molecule list $\mathcal{D}$, motif vocabulary $\mathcal{M} = \emptyset$, max size $K$, ring count threshold $N_{\text{ring}}$
**Ensure:** motif vocabulary $\mathcal{M}$
 1: Initialize each molecule $X \in \mathcal{D}$ with atom-level and ring-based motifs
 2: Count frequencies of ring-based motifs across $\mathcal{D}$
 3: Add all periodic-table elements, polymerization "*", and top-$N_{\text{ring}}$ frequent rings to $\mathcal{M}$
 4: **while** $|\mathcal{M}| < K$ **do**
 5:    **(1) Merge Neighbor:** Initialize empty multiset $\mathcal{S} \leftarrow \emptyset$
 6:    **for** each molecule $X \in \mathcal{D}$ **do**
 7:      **for** each motif $m$ from $X$ **do**
 8:        **for** each adjacent motif $m'$ in $X$ such that $m$ and $m'$ are mergeable under structural constraints (e.g., rings treated as units) **do**
 9:          form new motif $m \leftarrow m \cup m'$
10:          add $m$ to multiset $\mathcal{S}$ with frequency count
11:        **end for**
12:      **end for**
13:    **end for**
14:    **(2) Frequency Selection:** Find most frequent motif $m^* \in \mathcal{S}$
15:    **(3) Update Graph:**
16:    **for** each molecule $X \in \mathcal{D}$ **do**
17:      **for** each pair of adjacent motifs $(m, m')$ in $X$ **do**
18:        **if** their merged form equals $m^*$ **then**
19:          replace $m$ and $m'$ with $m^*$ in $X$
20:        **end if**
21:      **end for**
22:    **end for**
23:    Add $m^*$ to $\mathcal{M}$ if not already in it
24: **end while**
25: **return** $\mathcal{M}$

---

We present NPE in Algorithm 1, inspired by both the classic BPE and (Kong et al., 2022). We build the tokenizer with NPE on the pretraining data. To choose the motif vocabulary size, we analyze the number of nodes in motif-represented molecular graphs as the vocabulary size varies (Figure 8). We report mean, max, and median counts. We set $K_{\text{ring}} = K/10$, except for $K = 6000$, where $K_{\text{ring}} = 300$. When $K \geq 3000$, the mean and max node counts no longer significantly decrease, and the median remains unchanged. Therefore, we select $K = 3000$ with $K_{\text{ring}} = 300$ for pretraining.

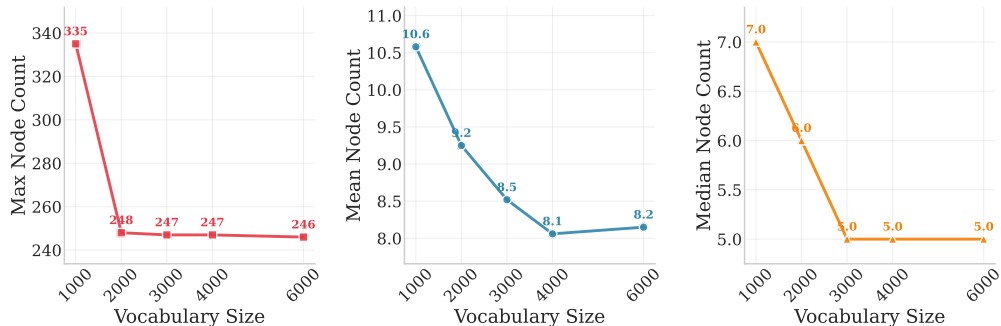

Figure 8: Change in node count with varying motif vocabulary size.

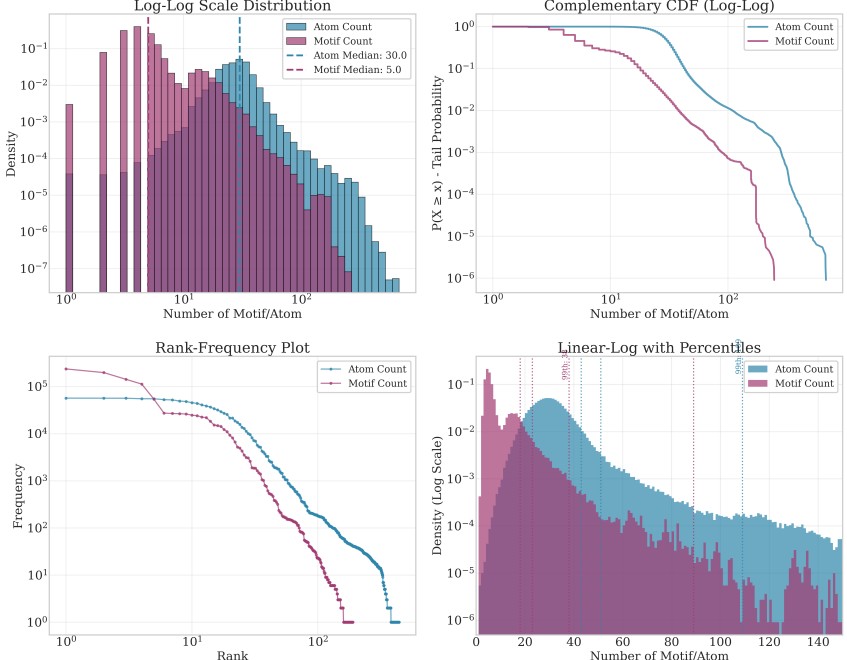

Figure 9: Comparison of the number of nodes in atom- and motif-based representations for 1 million molecules from the pretraining set.

Next, Figure 9 presents the tokenization results on the pretraining dataset. log-scale distributions of motif- and atom-level node counts. Both representations exhibit heavy-tailed behavior, as shown by the rank-frequency plots and complementary cumulative distribution functions (CCDFs).

Figure 10 shows the distribution of compression ratios, defined as $\frac{\text{Uncompressed Graph Size}}{\text{Compressed Graph Size}}$, in both linear and logarithmic scale. The ratio ranges from 1 to 40, with a median and mean around 5.5, indicating a consistent reduction in graph size by approximately a factor of five.

Figure 11 provides a detailed analysis of the relationship between atom-level and motif-level representations. We observe a mild positive correlation: larger molecules tend to yield higher compression ratios. Notably, molecules with 150 to 200 atoms are reduced by up to a factor of 15, demonstrating efficient compression at larger scales.

**Whether NPE captures popular functional groups and reaction fragments:** We evaluate the chemical validity of the motif vocabulary from the pretraining set. First, all motifs are chemically valid. Second, we compare the motifs in the vocabulary with 48 common functional groups. Forty-seven of them are present and one is missing. The groups are:

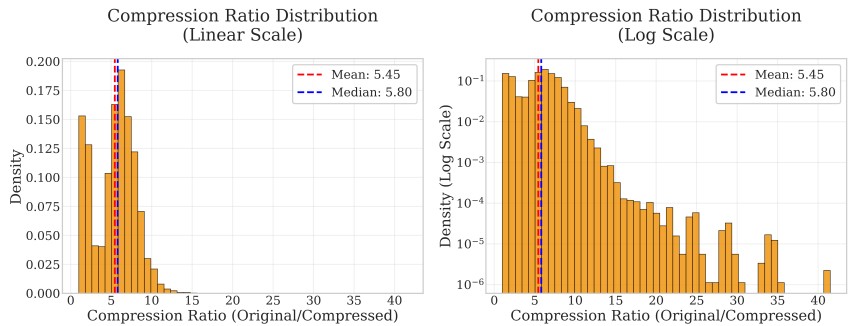

Figure 10: Analysis on the compression ratio.

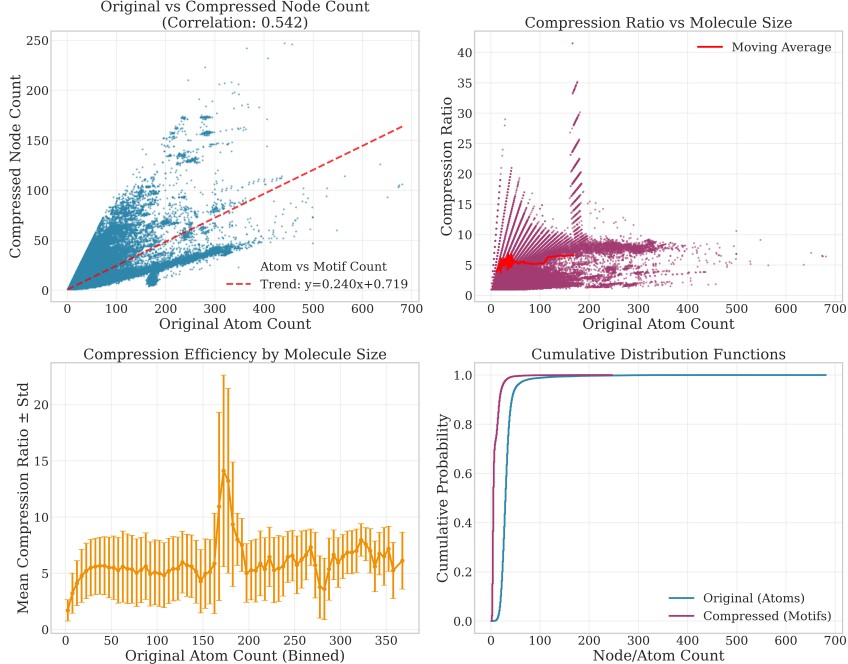

Figure 11: Analysis on the relationship between atom- and motif-level representations.

- **Included (47):** alcohol, aldehyde, ketone, carboxylic_acid, ester, ether, amine_primary, amine_secondary, amine_tertiary, amide, nitrile, nitro, sulfoxide, sulfone, thiol, disulfide, benzene, pyridine, pyrimidine, imidazole, thiophene, furan, pyrrole, methyl, ethyl, propyl, isopropyl, butyl, tert_butyl, phenyl, benzyl, trifluoromethyl, methoxy, ethoxy, acetyl, morpholine, piperidine, piperazine, pyrrolidine, tetrahydrofuran, oxazole, thiazole, phosphate, sulfonamide, carbamate, urea, guanidine.
- **Not included (1):** phosphonate.

Third, we compare the motifs in the vocabulary with fragments produced by breaking retrosynthetically interesting chemical substructures (BRICS) (Degen et al., 2008). BRICS produces more than 129,000 fragments. Among these fragments, 98.5% appear fewer than 100 times in more than one million pretraining molecules, and 63.2% appear only once. When we compare the top 100 BRICS fragments, 89% are present in the vocabulary constructed by NPE.

## C.3 MODEL PRETRAINING

We pretrain a 0.7B-parameter model (Transformer depth 24, hidden size 1280, 16 heads, MLP ratio 4) for 550 epochs, requiring 49 days on 2–4 H100 GPUs, or about 146 GPU days. We monitor

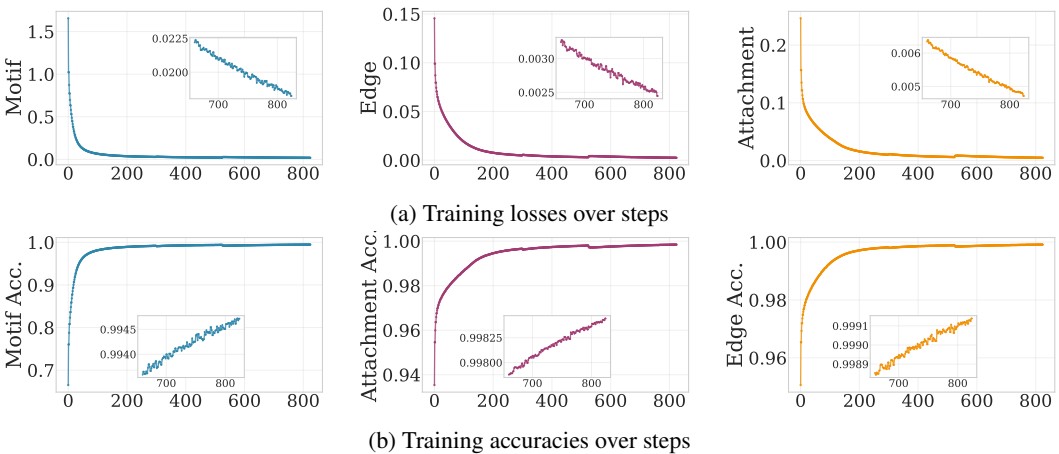

(a) Training losses over steps

(b) Training accuracies over steps

Figure 12: Training curves showing (a) losses and (b) accuracies.

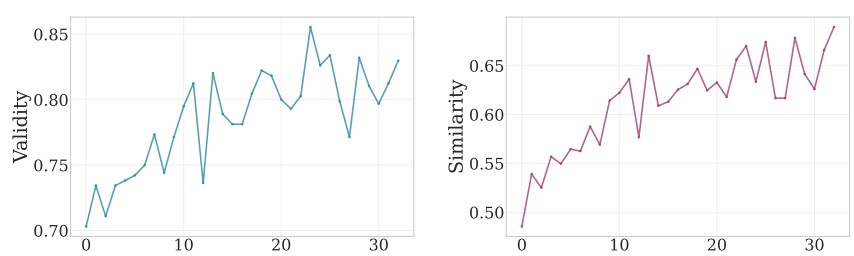

Figure 13: Generation validity and structure similarity to the target during pretraining.

training loss and reconstruction accuracy for each component in Eq. (6). As shown in Figure 12, loss decreases and accuracy increases throughout training. Near the end, values plateau but still show incremental gains. Pretraining was stopped once reconstruction accuracy exceeded 0.99 for all components due to resource limits. During training, we also generated 512 molecules at sampled steps using the validation set. In Figure 13, we report chemical validity and structural similarity to ground truth measured by MACCS fingerprints (Durant et al., 2002). Both metrics improve with training and reach about 0.83 validity and 0.69 similarity.

These trends in loss, accuracy, validity, and similarity indicate that larger models, more data, and additional compute could further improve DemoDiff as a molecular foundation model.

## D DETAILS ON EXPERIMENTS

### D.1 DETAILS ON EXPERIMENTAL SET-UPS

We curate 33 benchmark tasks across six categories to evaluate DemoDiff against 13 baselines, including eight molecular optimization methods, two conditional generation models, and three LLMs. Details are in Table 4. The benchmarks span seven drug rediscovery tasks, seven drug multi-objective optimization (MPO) tasks, five structure-constrained generation tasks, four drug design tasks, five target-based generation tasks, and five polymer property design tasks. Specifically, the benchmarks span the following tasks:

- **Drug rediscovery (7 tasks):** Celecoxib rediscovery, Mestranol similarity, Thiothixene rediscovery, Troglitazone rediscovery, Median 1 (median similarity between camphor and menthol), Median 2 (median similarity between tadalafil and sildenafil), and Albuterol similarity. These tasks use Oracle scoring functions based on the similarity between the drug and the target using extended connectivity fingerprint (Brown et al., 2019).

Table 4: Benchmark Task Statistics: Example Counts and Score Ranges [min, median, max]

| Task Name | Total | Pos | Med | Neg | All Scores | Pos Scores | Med Scores | Neg Scores |
|---|---|---|---|---|---|---|---|---|
| **Drug Rediscovery** | | | | | | | | |
| Albuterol Similarity | 450 | 150 | 150 | 150 | [0.075, 0.537, 1.000] | [0.752, 0.807, 1.000] | [0.502, 0.537, 0.744] | [0.075, 0.219, 0.476] |
| Celecoxib Rediscovery | 450 | 150 | 150 | 150 | [0.015, 0.548, 1.000] | [0.753, 0.786, 1.000] | [0.505, 0.548, 0.735] | [0.015, 0.153, 0.367] |
| Median 1 | 150 | 0 | 0 | 150 | [0.000, 0.057, 0.419] | | | [0.000, 0.057, 0.419] |
| Median 2 | 150 | 0 | 0 | 150 | [0.038, 0.122, 0.413] | | | [0.038, 0.122, 0.413] |
| Mestranol Similarity | 450 | 150 | 150 | 150 | [0.004, 0.538, 1.000] | [0.752, 0.824, 1.000] | [0.500, 0.538, 0.713] | [0.004, 0.150, 0.379] |
| Thiothixene Rediscovery | 336 | 36 | 150 | 150 | [0.019, 0.510, 1.000] | [0.753, 0.790, 1.000] | [0.505, 0.560, 0.736] | [0.019, 0.155, 0.354] |
| Troglitazone Rediscovery | 383 | 83 | 150 | 150 | [0.018, 0.526, 1.000] | [0.752, 0.792, 1.000] | [0.504, 0.553, 0.727] | [0.018, 0.143, 0.250] |
| **Drug MPO** | | | | | | | | |
| Amlodipine Mpo | 450 | 150 | 150 | 150 | [0.000, 0.517, 0.871] | [0.750, 0.784, 0.871] | [0.502, 0.517, 0.742] | [0.000, 0.145, 0.500] |
| Fexofenadine Mpo | 450 | 150 | 150 | 150 | [0.000, 0.570, 0.960] | [0.750, 0.772, 0.960] | [0.501, 0.570, 0.720] | [0.000, 0.125, 0.498] |
| Osimertinib Mpo | 450 | 150 | 150 | 150 | [0.000, 0.638, 0.908] | [0.750, 0.761, 0.908] | [0.501, 0.638, 0.747] | [0.000, 0.047, 0.493] |
| Perindopril Mpo | 306 | 6 | 150 | 150 | [0.000, 0.502, 0.790] | [0.766, 0.778, 0.790] | [0.502, 0.522, 0.680] | [0.000, 0.114, 0.415] |
| Ranolazine Mpo | 450 | 150 | 150 | 150 | [0.000, 0.575, 0.867] | [0.750, 0.764, 0.867] | [0.502, 0.575, 0.743] | [0.000, 0.052, 0.454] |
| Sitagliptin Mpo | 389 | 89 | 150 | 150 | [0.000, 0.610, 0.841] | [0.750, 0.768, 0.841] | [0.500, 0.641, 0.748] | [0.000, 0.000, 0.213] |
| Zaleplon Mpo | 300 | 0 | 150 | 150 | [0.000, 0.493, 0.637] | | [0.501, 0.528, 0.637] | [0.000, 0.003, 0.486] |
| **Structure Constrained Design** | | | | | | | | |
| Deco Hop | 450 | 150 | 150 | 150 | [0.252, 0.529, 0.953] | [0.793, 0.842, 0.953] | [0.502, 0.529, 0.677] | [0.252, 0.283, 0.500] |
| Isomers C7H8N2O2 | 450 | 150 | 150 | 150 | [0.000, 0.592, 1.000] | [0.799, 0.819, 1.000] | [0.535, 0.592, 0.741] | [0.000, 0.000, 0.449] |
| Isomers C9H10N2O2Pf2Cl | 450 | 150 | 150 | 150 | [0.000, 0.561, 0.882] | [0.767, 0.779, 0.882] | [0.503, 0.561, 0.720] | [0.000, 0.000, 0.386] |
| Scaffold Hop | 450 | 150 | 150 | 150 | [0.333, 0.510, 0.828] | [0.754, 0.782, 0.828] | [0.500, 0.510, 0.627] | [0.333, 0.380, 0.444] |
| Valsartan Smarts | 188 | 23 | 15 | 150 | [0.000, 0.000, 0.975] | [0.757, 0.806, 0.975] | [0.512, 0.686, 0.739] | [0.000, 0.000, 0.000] |
| **Drug Design** | | | | | | | | |
| DRD2 | 450 | 150 | 150 | 150 | [0.000, 0.623, 1.000] | [0.760, 0.961, 1.000] | [0.507, 0.623, 0.742] | [0.000, 0.004, 0.344] |
| GSK3B | 450 | 150 | 150 | 150 | [0.000, 0.615, 1.000] | [0.760, 0.880, 1.000] | [0.510, 0.615, 0.750] | [0.000, 0.030, 0.380] |
| JNK3 | 450 | 150 | 150 | 150 | [0.000, 0.570, 1.000] | [0.760, 0.890, 1.000] | [0.510, 0.570, 0.750] | [0.000, 0.010, 0.340] |
| QED | 450 | 150 | 150 | 150 | [0.010, 0.644, 0.947] | [0.751, 0.819, 0.947] | [0.501, 0.644, 0.749] | [0.010, 0.348, 0.499] |
| **Target Based Design** | | | | | | | | |
| Docking 5ht1b | 450 | 150 | 150 | 150 | [0.000, 0.607, 0.879] | [0.757, 0.771, 0.879] | [0.507, 0.607, 0.729] | [0.000, 0.450, 0.500] |
| Docking braf | 450 | 150 | 150 | 150 | [0.000, 0.600, 0.871] | [0.757, 0.771, 0.871] | [0.507, 0.600, 0.736] | [0.000, 0.464, 0.500] |
| Docking fa7 | 305 | 5 | 150 | 150 | [0.000, 0.507, 1.000] | [0.764, 0.800, 1.000] | [0.507, 0.536, 0.629] | [0.000, 0.450, 0.500] |
| Docking jak2 | 450 | 150 | 150 | 150 | [0.000, 0.586, 0.936] | [0.757, 0.771, 0.936] | [0.507, 0.586, 0.714] | [0.000, 0.464, 0.500] |
| Docking parp1 | 450 | 150 | 150 | 150 | [0.000, 0.614, 0.864] | [0.757, 0.771, 0.864] | [0.507, 0.614, 0.750] | [0.000, 0.471, 0.500] |
| **Material Design** | | | | | | | | |
| Polymer $CO_2$ $CH_4$ | 160 | 1 | 9 | 150 | [0.042, 0.203, 0.777] | [0.777, 0.777, 0.777] | [0.538, 0.603, 0.731] | [0.042, 0.198, 0.487] |
| Polymer $CO_2$ $N_2$ | 158 | 0 | 8 | 150 | [0.000, 0.110, 0.631] | | [0.506, 0.561, 0.631] | [0.000, 0.102, 0.498] |
| Polymer $H_2$ $CH_4$ | 177 | 13 | 14 | 150 | [0.000, 0.117, 1.000] | [0.752, 0.988, 1.000] | [0.509, 0.615, 0.739] | [0.000, 0.050, 0.487] |
| Polymer $H_2$ $N_2$ | 175 | 7 | 18 | 150 | [0.005, 0.178, 1.000] | [0.783, 0.974, 1.000] | [0.514, 0.619, 0.731] | [0.005, 0.151, 0.495] |
| Polymer $O_2$ $N_2$ | 173 | 0 | 23 | 150 | [0.309, 0.355, 0.726] | | [0.501, 0.579, 0.726] | [0.309, 0.346, 0.493] |

- **Drug MPO (7 tasks):** Perindopril MPO, Ranolazine MPO, Osimertinib MPO, Zaleplon MPO, Sitagliptin MPO, Amlodipine MPO, and Fexofenadine MPO. These tasks use Oracle scoring functions based on drug–target similarity with extended connectivity fingerprints, along with additional constraints such as logP, TPSA, and Bertz, computed using RD-Kit (Brown et al., 2019).

- **Structure-constrained design (5 tasks):** Isomers $C_7H_8N_2O_2$, Isomers $C_9H_{10}N_2O_2PF_2Cl$, Decoration hop, Scaffold hop, and Valsartan SMARTS. These tasks use Oracle scoring functions primarily based on SMARTS patterns that evaluate whether a particular structure is present or absent in the target, optionally combined with other computational constraints. Or whether the target is an isomer of the molecular formula.

- **Drug design (4 tasks):** DRD2, JNK3, GSK3$\beta$, and QED. These tasks use Oracle scoring functions based on ML models. QED is based on RDKit.

- **Target-based design (5 tasks):** Docking BRAF, Docking PARP1, Docking JAK2, Docking FA7, and Docking 5-HT1B. These tasks use Oracle scoring functions based on the docking program QuickVina 2 (Alhossary et al., 2015). Docking scores are negative, with smaller values indicating better binding. To map them into $[0, 1]$ (where larger values are better), we use $s = \text{clip}\left(-\frac{\text{docking score}}{14}, 0, 1\right)$, where $\text{clip}(x, a, b) = \min(\max(x, a), b)$.

- **Material design (5 tasks):** Polymer gas separation for different gas pairs: $CO_2/CH_4$, $CO_2/N_2$, $H_2/CH_4$, $H_2/N_2$, and $O_2/N_2$. Each task studies whether two gases can be separated based on the polymeric membrane materials. We evaluate their selectivity score for gas separation (Robeson, 2008), defined as the log-ratio of permeabilities relative to an empirical boundary, shifted and clipped into the range $[0, 1]$. Gas permeabilities are calculated using ML models trained on all available labeled data (a superset of the task-specific data), and the selectivity score is then computed based on gas permeabilities.

Each task contains up to 450 molecule–score pairs, evenly split into positive, medium, and negative groups. Some tasks may have fewer pairs due to insufficient positive examples, as shown in Table 4. For instance, the Median 1 and Median 2 tasks in drug rediscovery have no positive or medium examples. These pairs are used to train predictors for molecular optimization methods, conditional generators, or to provide demonstrations for ICL methods. Task scores lie within $[0, 1]$, with the objective of generating molecules with score 1. Each task also defines an oracle function, which is used only for evaluation, except by molecular optimization methods that actively query the oracle.

For baselines, we compare against four molecular optimization methods from the PMO benchmark (Gao et al., 2022). They are the top four methods selected from 25 candidates: Graph Genetic Algorithm (GraphGA), REINVENT (SMILES-based), Gaussian Process Bayesian Optimization (GPBO), and Superfast Traversal, Optimization, Novelty, Exploration, and Discovery (STONED, based on SELFIES). SELFIES is unavailable for polymers and STONED cannot be applied to material design tasks. We also include Genetic GFN (Kim et al., 2024) and GenMol (Lee et al., 2025). We have two evaluation settings: one with 100 oracle calls and one with 10,000 predictor calls. While PMO permits up to 10,000 Oracle calls, such budgets are impractical in real-world settings due to the cost and time associated with laboratory experiments, which may require days to months for a single call. To address this issue, we examine whether molecular optimization methods can be paired with predictor calls. Following prior work (Gao et al., 2022; Liu et al., 2024c), we use a random forest predictor trained on all 450 molecule–score pairs as the task-specific predictor.

We include conditional generation models such as LSTM and Graph DiT (Liu et al., 2024c). They are trained on all available training data for each task. For ICL, we compare DemoDiff (739M parameters) with recent large-scale LLMs, including DeepSeek-V3 (Liu et al., 2024a), GPT-4o (Achiam et al., 2023), and Qwen-Max (Yang et al., 2025), each with up to hundreds of billions of parameters. Besides, we finetune Llama3.1-8B-FT and Qwen3-8B-FT on the DemoDiff pretraining set. For LLMs, we sample 12 positive, 6 medium, and 6 negative as demonstrations.

For DemoDiff, we set the context size to 150 motif tokens. Excluding the target molecule, the context includes on average 23 demonstrations: half positive, one quarter medium, and one quarter negative. For evaluation, each method generates 10 valid, unique, and novel molecules per task, which are scored by oracle functions. We report the average of the top-10 oracle scores as the performance

Table 5: We compute the oracle of the top generation. We group 33 tasks into six categories and report the mean ± std within each category. Best results in each column are **bolded**.

| Task | Drug Rediscovery | Drug MPO | Structure Constrained | Drug Design | Target Based | Material Design | Avg Rank | Total Sum |
|---|---|---|---|---|---|---|---|---|
| Molecular Optimization Methods with 100 Oracle Calls | | | | | | | | |
| GraphGA | 0.28±0.06 | 0.49±0.18 | 0.46±0.14 | 0.45±0.33 | 0.74±0.04 | 0.72±0.23 | 9.14 | 16.73 |
| REINVENT | 0.31±0.07 | 0.47±0.16 | 0.45±0.14 | 0.56±0.38 | 0.75±0.06 | 0.00±0.00 | 10.74 | 13.68 |
| GPBO | 0.28±0.06 | 0.46±0.18 | 0.47±0.14 | 0.41±0.35 | 0.75±0.07 | 0.80±0.25 | 9.34 | 16.95 |
| STONED | 0.28±0.06 | 0.49±0.18 | 0.46±0.14 | 0.44±0.35 | 0.74±0.06 | No pSMILES | 12.23 | 13.07 |
| Genetic GFN | 0.28±0.11 | 0.51±0.19 | 0.37±0.23 | 0.36±0.39 | 0.75±0.04 | No pSMILES | 11.75 | 12.50 |
| GenMol | 0.38±0.09 | 0.55±0.18 | 0.51±0.31 | 0.61±0.32 | 0.77±0.07 | 0.87±0.14 | 6.94 | 19.63 |
| Molecular Optimization Methods with 10000 Predictor Calls | | | | | | | | |
| GraphGA | 0.33±0.14 | 0.56±0.21 | 0.54±0.38 | 0.60±0.41 | 0.83±0.12 | 0.70±0.21 | 8.10 | 19.03 |
| REINVENT | 0.36±0.29 | 0.38±0.30 | 0.42±0.43 | 0.77±0.12 | 0.59±0.37 | 0.87±0.17 | 9.11 | 17.67 |
| GPBO | 0.37±0.31 | 0.50±0.25 | 0.47±0.32 | 0.64±0.38 | **0.84±0.08** | 0.58±0.34 | 9.42 | 18.13 |
| STONED | 0.28±0.17 | 0.43±0.25 | 0.51±0.32 | 0.30±0.10 | 0.26±0.36 | No pSMILES | 14.36 | 10.07 |
| Genetic GFN | 0.16±0.07 | 0.19±0.33 | 0.19±0.27 | 0.28±0.28 | 0.39±0.37 | No pSMILES | 15.92 | 6.44 |
| GenMol | 0.44±0.28 | 0.35±0.34 | 0.27±0.38 | 0.62±0.37 | 0.70±0.11 | 0.43±0.34 | 9.74 | 15.04 |
| Conditional Generation Models | | | | | | | | |
| LSTM | 0.47±0.36 | 0.33±0.20 | **0.64±0.39** | 0.36±0.34 | 0.73±0.12 | 0.48±0.29 | 11.91 | 16.30 |
| Graph-DiT | 0.46±0.27 | 0.53±0.07 | 0.60±0.36 | 0.51±0.44 | 0.70±0.07 | 0.78±0.23 | 10.16 | 19.44 |
| Learning from In-Context Demonstrations | | | | | | | | |
| DeepSeek-V3 | **0.66±0.37** | **0.60±0.26** | 0.54±0.25 | 0.74±0.14 | 0.71±0.12 | 0.75±0.27 | 8.23 | 21.77 |
| GPT-4o | 0.53±0.31 | 0.56±0.20 | 0.52±0.32 | 0.54±0.41 | 0.69±0.09 | 0.77±0.22 | 9.56 | 19.69 |
| Qwen-Max | 0.18±0.28 | 0.19±0.17 | 0.53±0.34 | 0.51±0.45 | 0.21±0.29 | 0.25±0.40 | 14.19 | 9.60 |
| Llama3.1-8B-FT | 0.25±0.34 | 0.38±0.32 | 0.25±0.38 | 0.33±0.29 | 0.08±0.19 | 0.20±0.30 | 14.67 | 8.42 |
| Qwen3-8B-FT | 0.47±0.33 | 0.48±0.25 | 0.28±0.38 | 0.88±0.14 | 0.70±0.08 | 0.37±0.51 | 9.80 | 17.02 |
| DemoDiff (Ours) | 0.54±0.33 | 0.54±0.19 | 0.59±0.37 | **0.91±0.07** | 0.77±0.10 | **0.93±0.16** | **4.68** | **22.63** |

score and compute its harmonic mean with the diversity score. The diversity score is computed as

$$\text{IntDiv}(G) = 1 - \left( \frac{1}{|G|^2} \sum_{\substack{m_1,m_2 \in G \\ m_1 \neq m_2}} T(m_1, m_2)^2 \right)^{\frac{1}{2}}, \tag{8}$$

where $G$ denotes the generated set of molecules for evaluation. $T$ denotes the Tanimoto similarity. For DemoDiff, we first generate 1000 candidates and select the top 10 with the highest consistency scores, prioritizing alignment with the context order of positive, medium, and negative examples.

## D.2 Additional Discussion of Experimental Results

We include more results in Tables 8 to 11 and 14 to 17. We use an additional diverse hits metric from (Renz et al., 2024) and present the results in Table 7. We include detailed results from the top-10 generation on the oracle scores and diversity scores in Table 6. Beyond the discussion of ICL methods and DemoDiff in Section 4.1, we have additional observations:

**Oracle quality critically affects molecular optimization.** Comparing molecular optimization methods under varying numbers of function calls, we find that allowing more predictor queries does not consistently lead to better performance. This suggests that both the quantity and quality of function evaluations (oracle or predictor) are essential for guiding molecular optimization. While not the main focus of this study, this insight points to an important direction for future work. For instance, in the structure-constrained design task involving the Valsartan SMARTS pattern (CN(C=O)Cc1ccc(c2ccccc2)cc1), shown in Table 16, all molecular optimization methods receive a score of zero. This failure is due to a predictor trained on limited data, which cannot model the latent design constraints, such as satisfying multiple SMARTS patterns and physicochemical properties (e.g., logP, TPSA, and Bertz index (Brown et al., 2019)). In contrast, in target-based design tasks (e.g., Table 18), where training data are sufficient, more predictor calls improve performance by allowing finer structural optimization.

Table 6: We compute the oracle and diversity scores from the Top-10 generations. We group 33 tasks into six categories and report the mean ± std within each category. The best results in each column are **bolded**.

| Method | Drug Design | | Drug MPO | | Drug Rediscovery | | Material Design | | Structure Constrained | | Target Based | |
|---|---|---|---|---|---|---|---|---|---|---|---|---|
| | Oracle 4 | Diversity 4 | Oracle 7 | Diversity 7 | Oracle 7 | Diversity 7 | Oracle 5 | Diversity 5 | Oracle 5 | Diversity 5 | Oracle 5 | Diversity 5 |
| GraphGA | 0.34±0.38 | 0.88±0.02 | 0.40±0.19 | 0.86±0.01 | 0.23±0.06 | 0.87±0.02 | 0.57±0.19 | 0.61±0.08 | 0.30±0.19 | 0.89±0.01 | 0.68±0.06 | 0.87±0.01 |
| REINVENT | 0.35±0.39 | 0.88±0.01 | 0.39±0.17 | 0.86±0.02 | 0.24±0.06 | 0.86±0.02 | 0.00±0.00 | 0.89±0.01 | 0.31±0.19 | 0.88±0.02 | 0.67±0.05 | 0.86±0.01 |
| GPBO | 0.32±0.38 | 0.88±0.01 | 0.39±0.19 | 0.87±0.01 | 0.24±0.06 | 0.87±0.02 | 0.49±0.24 | 0.87±0.02 | 0.30±0.19 | 0.88±0.01 | 0.68±0.06 | 0.86±0.01 |
| STONED | 0.31±0.40 | 0.88±0.01 | 0.40±0.19 | 0.86±0.01 | 0.23±0.06 | 0.87±0.02 | No pSMILES | No pSMILES | 0.30±0.19 | 0.88±0.01 | 0.68±0.06 | 0.86±0.01 |
| Genetic GFN | 0.28±0.41 | 0.87±0.02 | 0.39±0.21 | 0.86±0.01 | 0.23±0.08 | 0.86±0.01 | No pSMILES | No pSMILES | 0.29±0.24 | 0.87±0.03 | 0.68±0.05 | 0.86±0.01 |
| GenMol | 0.54±0.28 | 0.60±0.11 | 0.48±0.22 | 0.65±0.08 | 0.33±0.08 | 0.59±0.15 | 0.61±0.12 | 0.64±0.07 | 0.35±0.22 | 0.73±0.08 | 0.71±0.07 | 0.67±0.02 |
| *Molecular Optimization Methods with 10000 Predictor Calls* | | | | | | | | | | | | |
| GraphGA | 0.57±0.42 | 0.57±0.12 | 0.48±0.26 | 0.65±0.09 | 0.28±0.11 | 0.65±0.13 | 0.60±0.26 | 0.54±0.09 | 0.47±0.29 | 0.58±0.15 | 0.78±0.11 | 0.55±0.03 |
| REINVENT | 0.72±0.14 | 0.28±0.23 | 0.35±0.29 | 0.28±0.31 | 0.32±0.25 | 0.50±0.32 | 0.40±0.18 | 0.78±0.05 | 0.30±0.28 | 0.35±0.17 | 0.50±0.32 | 0.16±0.09 |
| GPBO | 0.60±0.36 | 0.57±0.17 | 0.42±0.30 | 0.66±0.11 | 0.34±0.32 | 0.68±0.24 | 0.51±0.30 | 0.42±0.05 | 0.47±0.30 | 0.75±0.15 | 0.80±0.09 | 0.70±0.08 |
| STONED | 0.21±0.09 | 0.42±0.10 | 0.38±0.23 | 0.56±0.25 | 0.24±0.15 | 0.75±0.17 | No pSMILES | No pSMILES | 0.47±0.30 | 0.75±0.15 | 0.26±0.35 | 0.37±0.07 |
| Genetic GFN | 0.28±0.27 | 0.09±0.06 | 0.19±0.33 | 0.14±0.10 | 0.15±0.06 | 0.14±0.10 | No pSMILES | No pSMILES | 0.19±0.27 | 0.11±0.06 | 0.36±0.35 | 0.12±0.06 |
| GenMol | 0.60±0.38 | 0.39±0.10 | 0.34±0.33 | 0.41±0.13 | 0.39±0.21 | 0.35±0.05 | 0.29±0.26 | 0.31±0.19 | 0.26±0.38 | 0.46±0.06 | 0.65±0.10 | 0.44±0.06 |
| *Conditional Generation Models* | | | | | | | | | | | | |
| LSTM | 0.26±0.35 | 0.90±0.02 | 0.09±0.04 | 0.89±0.03 | 0.33±0.27 | 0.82±0.12 | 0.10±0.07 | 0.81±0.18 | 0.46±0.29 | 0.87±0.04 | 0.61±0.06 | 0.89±0.01 |
| Graph-DiT | 0.41±0.38 | 0.89±0.05 | 0.38±0.17 | 0.87±0.06 | 0.33±0.21 | 0.83±0.07 | 0.43±0.18 | 0.85±0.02 | 0.52±0.31 | 0.86±0.08 | 0.59±0.05 | 0.89±0.01 |
| *Learning from In-Context Demonstrations* | | | | | | | | | | | | |
| DeepSeek-V3 | 0.61±0.24 | 0.77±0.08 | 0.51±0.26 | 0.61±0.08 | 0.58±0.32 | 0.52±0.17 | 0.37±0.27 | 0.65±0.18 | 0.44±0.29 | 0.66±0.10 | 0.68±0.12 | 0.62±0.11 |
| GPT-4o | 0.43±0.42 | 0.86±0.01 | 0.43±0.21 | 0.81±0.06 | 0.41±0.26 | 0.77±0.11 | 0.30±0.16 | 0.88±0.02 | 0.44±0.27 | 0.84±0.05 | 0.64±0.07 | 0.85±0.01 |
| Qwen-Max | 0.29±0.39 | 0.64±0.16 | 0.14±0.14 | 0.31±0.17 | 0.14±0.21 | 0.18±0.23 | 0.09±0.16 | 0.30±0.24 | 0.40±0.29 | 0.44±0.28 | 0.17±0.24 | 0.25±0.36 |
| Llama3.1-8B-FT | 0.23±0.29 | 0.87±0.01 | 0.17±0.17 | 0.77±0.15 | 0.13±0.09 | 0.79±0.10 | 0.40±0.10 | 0.37±0.52 | 0.23±0.36 | 0.83±0.12 | 0.01±0.03 | 0.88±0.01 |
| Qwen3-8B-FT | 0.34±0.23 | 0.86±0.01 | 0.17±0.10 | 0.88±0.02 | 0.29±0.21 | 0.77±0.16 | 0.46±0.47 | 0.80±0.18 | 0.23±0.33 | 0.84±0.10 | 0.55±0.09 | 0.88±0.03 |
| DemoDiff (Ours) | 0.81±0.16 | 0.79±0.08 | 0.43±0.23 | 0.85±0.02 | 0.41±0.27 | 0.71±0.17 | 0.56±0.15 | 0.86±0.03 | 0.54±0.33 | 0.79±0.14 | 0.71±0.08 | 0.87±0.01 |

Table 7: Diverse hits across six task categories. Diverse hits from (Renz et al., 2024) are computed as the number of generated hits with oracle scores greater than 0.5 and pairwise Tanimoto distances greater than 0.7. Results are reported as mean ± standard deviation for each category. Best results in each column are **bolded**.

| Task | Drug Design | Drug MPO | Drug Rediscovery | Material Design | Structure Constrained | Target Based | Avg Rank | Total Sum |
|---|---|---|---|---|---|---|---|---|
| # Tasks | 4 | 7 | 7 | 5 | 5 | 5 | 33 | 33 |
| GraphGA | 11.25±22.50 | 12.86±21.96 | 0.00±0.00 | 12.20±12.54 | 9.00±20.12 | **45.00±0.00** | 4.67 | 90.31 |
| REINVENT | 11.25±22.50 | 10.43±18.47 | 0.00±0.00 | 0.00±0.00 | 9.00±20.12 | 44.80±0.45 | 8.00 | 75.48 |
| GPBO | 11.25±22.50 | 12.86±21.96 | 0.00±0.00 | 10.40±10.02 | 9.00±20.12 | 44.40±0.55 | 5.67 | 87.91 |
| STONED | 11.25±22.50 | 12.86±21.96 | 0.00±0.00 | No pSMILES | 9.00±20.12 | **45.00±0.00** | 7.17 | 78.11 |
| Genetic GFN | 11.00±22.00 | 12.71±21.72 | 0.00±0.00 | No pSMILES | 8.80±19.68 | **45.00±0.00** | 9.17 | 77.51 |
| GenMol | 14.75±17.15 | 9.00±12.78 | 0.14±0.38 | 7.60±6.11 | 5.00±10.63 | 27.80±2.17 | 8.33 | 64.29 |
| *Molecular Optimization Methods with 10000 Predictor Calls* | | | | | | | | |
| GraphGA | 8.25±16.50 | 5.00±6.51 | 0.29±0.76 | 5.20±9.55 | 10.40±14.40 | 8.60±6.02 | 9.50 | 37.74 |
| REINVENT | 1.00±2.00 | 1.00±2.65 | 0.00±0.00 | 4.20±6.22 | 0.00±0.00 | 0.00±0.00 | 13.50 | 6.20 |
| GPBO | 0.00±0.00 | 5.86±10.59 | 0.00±0.00 | 0.00±0.00 | 12.20±19.24 | 25.80±15.97 | 11.17 | 43.86 |
| STONED | 0.00±0.00 | 0.00±0.00 | 0.00±0.00 | No pSMILES | 15.80±21.48 | 0.20±0.45 | 13.33 | 16.00 |
| Genetic GFN | 0.00±0.00 | 0.00±0.00 | 0.00±0.00 | No pSMILES | 0.00±0.00 | 0.00±0.00 | 15.83 | 0.00 |
| GenMol | 0.00±0.00 | 0.00±0.00 | 0.00±0.00 | 0.20±0.45 | 0.40±0.89 | 0.40±0.89 | 14.33 | 1.00 |
| *Conditional Generation Models* | | | | | | | | |
| LSTM | 11.25±22.50 | 0.00±0.00 | **7.57±15.89** | 0.20±0.45 | 25.00±23.09 | 38.00±15.65 | 7.50 | 82.02 |
| Graph-DiT | 19.75±23.24 | 8.86±16.44 | 6.43±16.14 | 4.40±6.43 | **28.00±23.35** | 40.20±10.73 | 5.33 | 107.64 |
| *Learning from In-Context Demonstrations* | | | | | | | | |
| DeepSeek-V3 | 29.25±19.72 | 12.57±14.73 | 1.57±4.16 | 0.80±1.79 | 1.80±3.49 | 19.80±15.64 | 8.67 | 65.79 |
| GPT-4o | 22.25±25.70 | 12.29±13.49 | 7.14±11.95 | 3.80±6.38 | 19.60±22.65 | 44.80±0.45 | 4.67 | 109.88 |
| Qwen-Max | 3.50±6.35 | 0.00±0.00 | 0.00±0.00 | 0.00±0.00 | 4.60±9.21 | 0.00±0.00 | 14.00 | 8.10 |
| Llama3.1-8B-FT | 0.75±1.50 | 2.71±5.53 | 0.00±0.00 | 0.00±0.00 | 2.60±5.81 | 0.00±0.00 | 14.00 | 6.06 |
| Qwen3-8B-FT | 8.25±12.69 | 0.43±0.53 | 0.86±2.27 | 0.33±0.58 | 6.00±13.42 | 25.20±16.65 | 11.33 | 41.07 |
| DemoDiff (Ours) | **35.00±10.80** | **14.43±21.03** | 5.00±13.23 | **15.00±14.20** | 20.20±23.08 | 44.40±0.89 | **2.67** | **134.03** |

**Performance alignment across methods may indicate task difficulty.** It is hard to give a formal measure of task difficulty because it depends on the oracle setup and on the quality of the data. We find that tasks where molecular optimization reaches good results, such as target based tasks or material design tasks, are also tasks where ICL methods can infer the core concept with only a few demonstrations. This pattern suggests that task difficulty may be partly reflected in the consistency across methods. There are exceptions. As shown in Table 9, for DRD2 and JNK3, molecular optimization does not perform well under limited supervision, while ICL methods, including DeepSeek-V3, GPT-4o, and DemoDiff, reach strong results.

Table 8: We compute the oracle and diversity scores from the Top-10 generations and report the harmonic mean of these two values. The results are reported for the Drug Design task category. Best results in each column are **bolded**.

| Task | DRD2 | JNK3 | GSK3B | QED | Avg Rank | Total Sum |
|------|------|------|-------|-----|----------|-----------|
| # Tasks | | | | | 33 | 33 |
| Molecular Optimization Methods with 100 Oracle Calls | | | | | | |
| GraphGA | 0.22 | 0.23 | 0.31 | 0.90 | 8.50 | 1.65 |
| REINVENT | 0.31 | 0.21 | 0.26 | **0.90** | 9.50 | 1.68 |
| GPBO | 0.20 | 0.25 | 0.24 | 0.89 | 10.00 | 1.57 |
| STONED | 0.11 | 0.23 | 0.24 | 0.90 | 11.00 | 1.47 |
| Genetic GFN | 0.10 | 0.16 | 0.16 | 0.88 | 13.75 | 1.30 |
| GenMol | 0.68 | 0.29 | 0.48 | 0.77 | 6.75 | 2.22 |
| Molecular Optimization Methods with 10000 Predictor Calls | | | | | | |
| GraphGA | 0.60 | 0.18 | 0.40 | 0.79 | 9.25 | 1.97 |
| REINVENT | 0.31 | 0.09 | 0.40 | 0.71 | 12.50 | 1.51 |
| GPBO | 0.55 | 0.27 | 0.54 | 0.60 | 8.25 | 1.97 |
| STONED | 0.14 | 0.23 | 0.37 | 0.32 | 12.25 | 1.07 |
| Genetic GFN | 0.04 | 0.12 | 0.28 | 0.06 | 17.00 | 0.50 |
| GenMol | 0.56 | 0.39 | 0.49 | 0.12 | 8.50 | 1.57 |
| Conditional Generation Models | | | | | | |
| LSTM | 0.15 | 0.04 | 0.27 | 0.83 | 14.00 | 1.30 |
| Graph-DiT | 0.78 | 0.08 | 0.23 | 0.81 | 12.25 | 1.90 |
| Learning from In-Context Demonstrations | | | | | | |
| DeepSeek-V3 | 0.71 | **0.65** | 0.42 | 0.84 | 4.75 | 2.62 |
| GPT-4o | 0.80 | 0.13 | 0.13 | 0.84 | 11.25 | 1.90 |
| Qwen-Max | 0.00 | 0.14 | 0.36 | 0.68 | 15.00 | 1.18 |
| Llama3.1-8B-FT | 0.06 | 0.21 | 0.76 | 0.19 | 12.25 | 1.22 |
| Qwen3-8B-FT | 0.37 | 0.50 | 0.22 | 0.74 | 10.75 | 1.83 |
| DemoDiff (Ours) | **0.88** | 0.65 | **0.78** | 0.87 | **2.50** | **3.18** |

## D.3 DETAILS ON CASE STUDIES

Figures 14 to 16 present case studies using negative demonstrations to generate molecules with positive scores. The tasks include structure-constrained design of an isomer with 17 demonstrations, drug MPO of Osimertinib with 23 demonstrations, and protein target design of PARP1 with 26 demonstrations. These molecules contain 460, 631, and 481 atoms, respectively, far exceeding the 150-token context window under atom-level representations. The motif-level representation efficiently encodes all demonstrations within the same window.

Table 9: We compute the oracle of the top generation. The results are reported for the Drug Design task category. Best results in each column are **bolded**.

| Task | DRD2 | JNK3 | GSK3B | QED | Avg Rank | Total Sum |
|------|------|------|-------|-----|----------|-----------|
| Molecular Optimization Methods with 100 Oracle Calls | | | | | | |
| GraphGA | 0.23 | 0.23 | 0.41 | 0.94 | 10.25 | 1.82 |
| REINVENT | 0.83 | 0.23 | 0.24 | 0.94 | 9.50 | 2.25 |
| GPBO | 0.23 | 0.23 | 0.23 | 0.94 | 12.00 | 1.64 |
| STONED | 0.19 | 0.23 | 0.41 | 0.94 | 12.25 | 1.77 |
| Genetic GFN | 0.19 | 0.14 | 0.15 | **0.95** | 13.50 | 1.43 |
| GenMol | 0.85 | 0.24 | 0.45 | 0.90 | 9.25 | 2.44 |
| Molecular Optimization Methods with 10000 Predictor Calls | | | | | | |
| GraphGA | 0.99 | 0.14 | 0.38 | 0.90 | 11.75 | 2.41 |
| REINVENT | 0.76 | 0.61 | 0.80 | 0.91 | 7.00 | 3.08 |
| GPBO | 0.99 | 0.19 | 0.47 | 0.93 | 9.50 | 2.58 |
| STONED | 0.25 | 0.21 | 0.44 | 0.30 | 13.75 | 1.20 |
| Genetic GFN | 0.03 | 0.41 | 0.61 | 0.07 | 12.75 | 1.12 |
| GenMol | 1.00 | 0.63 | 0.75 | 0.11 | 7.25 | 2.49 |
| Conditional Generation Models | | | | | | |
| LSTM | 0.28 | 0.06 | 0.26 | 0.85 | 15.25 | 1.45 |
| Graph-DiT | 0.99 | 0.07 | 0.22 | 0.78 | 14.00 | 2.06 |
| Learning from In-Context Demonstrations | | | | | | |
| DeepSeek-V3 | 0.77 | 0.75 | 0.54 | 0.88 | 8.50 | 2.94 |
| GPT-4o | 0.94 | 0.21 | 0.17 | 0.85 | 13.75 | 2.17 |
| Qwen-Max | 0.00 | 0.26 | **0.89** | 0.89 | 10.50 | 2.04 |
| Llama3.1-8B-FT | 0.03 | 0.33 | 0.72 | 0.26 | 12.25 | 1.34 |
| Qwen3-8B-FT | 0.90 | **1.00** | 0.69 | 0.94 | 4.25 | 3.54 |
| DemoDiff (Ours) | **1.00** | 0.83 | **0.89** | 0.93 | **2.75** | **3.65** |

Table 10: We compute the oracle and diversity scores from the Top-10 generations and report the harmonic mean of these two values. The results are reported for the Drug MPO task category. Best results in each column are **bolded**.

| Task | Perindopril MPO | Ranolazine MPO | Osimertinib MPO | Zaleplon MPO | Sitagliptin MPO | Amlodipine MPO | Fexofenadine MPO | Avg Rank | Total Sum |
|------|------|------|------|------|------|------|------|------|------|
| # Tasks | | | | | | | | 33 | 33 |
| Molecular Optimization Methods with 100 Oracle Calls | | | | | | | | | |
| GraphGA | 0.52 | 0.35 | 0.77 | 0.47 | 0.24 | 0.60 | 0.70 | 5.14 | 3.64 |
| REINVENT | 0.50 | 0.39 | 0.75 | 0.48 | 0.24 | 0.59 | 0.66 | 6.43 | 3.62 |
| GPBO | 0.51 | 0.39 | 0.76 | 0.45 | 0.23 | 0.59 | 0.68 | 7.57 | 3.60 |
| STONED | 0.52 | 0.35 | 0.77 | 0.47 | 0.24 | 0.60 | 0.70 | 6.14 | 3.64 |
| Genetic GFN | 0.52 | 0.47 | 0.77 | 0.43 | 0.09 | 0.58 | 0.70 | 8.00 | 3.54 |
| GenMol | 0.49 | 0.56 | 0.70 | **0.52** | 0.17 | 0.54 | 0.62 | 7.86 | 3.59 |
| Molecular Optimization Methods with 10000 Predictor Calls | | | | | | | | | |
| GraphGA | 0.41 | 0.60 | 0.64 | 0.48 | 0.13 | 0.59 | 0.62 | 8.00 | 3.48 |
| REINVENT | 0.11 | 0.24 | 0.65 | 0.39 | 0.00 | 0.09 | 0.10 | 16.29 | 1.58 |
| GPBO | 0.23 | 0.36 | 0.68 | 0.45 | 0.12 | 0.62 | 0.67 | 9.29 | 3.12 |
| STONED | 0.53 | 0.26 | 0.47 | 0.47 | 0.01 | 0.60 | 0.45 | 10.14 | 2.79 |
| Genetic GFN | 0.00 | 0.01 | 0.43 | 0.00 | 0.00 | 0.00 | 0.12 | 18.71 | 0.56 |
| GenMol | 0.02 | 0.47 | 0.55 | 0.34 | 0.00 | 0.29 | 0.00 | 15.29 | 1.68 |
| Conditional Generation Models | | | | | | | | | |
| LSTM | 0.06 | 0.14 | 0.26 | 0.09 | 0.13 | 0.21 | 0.21 | 16.29 | 1.10 |
| Graph-DiT | 0.59 | 0.24 | 0.71 | 0.43 | **0.31** | 0.55 | 0.66 | 8.43 | 3.49 |
| Learning from In-Context Demonstrations | | | | | | | | | |
| DeepSeek-V3 | 0.49 | **0.63** | 0.70 | 0.42 | 0.10 | 0.63 | 0.58 | 8.29 | 3.55 |
| GPT-4o | **0.63** | 0.39 | 0.68 | 0.46 | 0.17 | **0.69** | 0.67 | 6.00 | 3.70 |
| Qwen-Max | 0.26 | 0.21 | 0.00 | 0.39 | 0.00 | 0.25 | 0.06 | 17.00 | 1.17 |
| Llama3.1-8B-FT | 0.05 | 0.39 | 0.26 | 0.00 | 0.00 | 0.42 | 0.55 | 16.00 | 1.67 |
| Qwen3-8B-FT | 0.27 | 0.39 | 0.36 | 0.03 | 0.06 | 0.40 | 0.38 | 14.14 | 1.88 |
| DemoDiff (Ours) | 0.52 | 0.58 | **0.80** | 0.43 | 0.09 | 0.62 | **0.73** | 5.00 | 3.78 |

Table 11: We compute the oracle of the top generation. The results are reported for the Drug MPO task category. Best results in each column are **bolded**.

| Task | Perindopril MPO | Ranolazine MPO | Osimertinib MPO | Zaleplon MPO | Sitagliptin MPO | Amlodipine MPO | Fexofenadine MPO | Avg Rank | Total Sum |
|---|---|---|---|---|---|---|---|---|---|
| Molecular Optimization Methods with 100 Oracle Calls | | | | | | | | | |
| GraphGA | 0.43 | 0.36 | 0.78 | 0.40 | 0.23 | 0.56 | 0.62 | 8.14 | 3.40 |
| REINVENT | 0.40 | 0.41 | 0.74 | 0.43 | 0.23 | 0.48 | 0.58 | 11.57 | 3.26 |
| GPBO | 0.38 | 0.36 | 0.78 | 0.37 | 0.23 | 0.48 | 0.60 | 13.00 | 3.20 |
| STONED | 0.43 | 0.36 | 0.78 | 0.40 | 0.23 | 0.56 | 0.62 | 9.57 | 3.40 |
| Genetic GFN | 0.42 | 0.61 | 0.75 | 0.46 | 0.16 | 0.49 | 0.66 | 9.29 | 3.54 |
| GenMol | 0.43 | 0.73 | 0.79 | 0.47 | 0.28 | 0.49 | 0.64 | 6.14 | 3.82 |
| Molecular Optimization Methods with 10000 Predictor Calls | | | | | | | | | |
| GraphGA | 0.40 | 0.67 | **0.83** | 0.41 | 0.28 | 0.53 | **0.81** | 5.43 | 3.93 |
| REINVENT | 0.17 | 0.34 | 0.82 | 0.40 | 0.00 | 0.22 | 0.72 | 12.29 | 2.68 |
| GPBO | 0.15 | 0.41 | 0.78 | 0.40 | 0.30 | 0.64 | 0.80 | 8.00 | 3.48 |
| STONED | 0.44 | 0.26 | 0.77 | 0.42 | 0.01 | 0.52 | 0.62 | 11.29 | 3.03 |
| Genetic GFN | 0.00 | 0.01 | 0.73 | 0.00 | 0.00 | 0.00 | 0.61 | 18.00 | 1.35 |
| GenMol | 0.01 | 0.68 | 0.81 | 0.47 | 0.00 | 0.47 | 0.00 | 11.43 | 2.44 |
| Conditional Generation Models | | | | | | | | | |
| LSTM | 0.08 | 0.11 | 0.59 | 0.22 | 0.44 | 0.42 | 0.46 | 15.71 | 2.31 |
| Graph-DiT | 0.58 | 0.51 | 0.61 | 0.46 | **0.46** | 0.48 | 0.61 | 9.00 | 3.71 |
| Learning from In-Context Demonstrations | | | | | | | | | |
| DeepSeek-V3 | **0.74** | **0.77** | 0.78 | 0.39 | 0.10 | **0.77** | 0.63 | 7.29 | **4.17** |
| GPT-4o | 0.74 | 0.47 | 0.71 | 0.43 | 0.19 | 0.71 | 0.65 | 8.14 | 3.90 |
| Qwen-Max | 0.30 | 0.21 | 0.00 | **0.49** | 0.00 | 0.22 | 0.12 | 15.14 | 1.33 |
| Llama3.1-8B-FT | 0.12 | 0.67 | 0.61 | 0.00 | 0.00 | 0.53 | 0.71 | 13.43 | 2.63 |
| Qwen3-8B-FT | 0.42 | 0.57 | 0.67 | 0.08 | 0.22 | 0.74 | 0.68 | 10.00 | 3.38 |
| DemoDiff (Ours) | 0.44 | 0.65 | 0.77 | 0.39 | 0.24 | 0.55 | 0.73 | 7.14 | 3.77 |

Table 12: We compute the oracle and diversity scores from the Top-10 generations and report the harmonic mean of these two values. The results are reported for the Drug Rediscovery task category. Best results in each column are **bolded**.

| Task
#Tasks | Celecoxib Rediscovery | Mestranol Similarity | Thiothixene Rediscovery | Troglitazone Rediscovery | Median 1 | Median 2 | Albuterol Similarity | Avg Rank
33 | Total Sum
33 |
|---|---|---|---|---|---|---|---|---|---|
| Molecular Optimization Methods with 100 Oracle Calls | | | | | | | | | |
| GraphGA | 0.39 | 0.39 | 0.38 | 0.34 | 0.26 | 0.29 | 0.49 | 9.14 | 2.54 |
| REINVENT | 0.39 | 0.44 | 0.37 | 0.34 | 0.30 | 0.28 | 0.50 | 8.29 | 2.62 |
| GPBO | 0.37 | 0.39 | 0.39 | 0.33 | 0.28 | 0.30 | 0.51 | 8.57 | 2.57 |
| STONED | 0.39 | 0.39 | 0.38 | 0.34 | 0.26 | 0.29 | 0.49 | 10.14 | 2.54 |
| Genetic GFN | 0.44 | 0.40 | 0.36 | 0.31 | 0.24 | 0.25 | 0.51 | 10.86 | 2.51 |
| GenMol | 0.48 | 0.45 | 0.38 | 0.29 | **0.41** | **0.37** | 0.55 | 6.86 | 2.93 |
| Molecular Optimization Methods with 10000 Predictor Calls | | | | | | | | | |
| GraphGA | 0.38 | 0.41 | 0.38 | 0.35 | 0.28 | 0.28 | 0.55 | 8.57 | 2.61 |
| REINVENT | 0.31 | 0.15 | 0.38 | 0.26 | 0.19 | 0.27 | 0.53 | 12.00 | 2.08 |
| GPBO | 0.34 | 0.43 | 0.37 | 0.26 | 0.19 | 0.27 | 0.46 | 12.71 | 2.32 |
| STONED | 0.39 | 0.49 | 0.38 | 0.32 | 0.19 | 0.27 | 0.26 | 10.86 | 2.30 |
| Genetic GFN | 0.27 | 0.07 | 0.21 | 0.12 | 0.08 | 0.09 | 0.11 | 18.86 | 0.96 |
| GenMol | 0.43 | 0.36 | 0.34 | 0.35 | 0.25 | 0.24 | 0.41 | 11.57 | 2.38 |
| Conditional Generation Models | | | | | | | | | |
| LSTM | **0.61** | **0.66** | 0.20 | 0.25 | 0.20 | 0.12 | 0.70 | 10.14 | 2.74 |
| Graph-DiT | 0.60 | 0.58 | 0.33 | 0.22 | 0.39 | 0.17 | 0.73 | 8.29 | 3.02 |
| Learning from In-Context Demonstrations | | | | | | | | | |
| DeepSeek-V3 | 0.54 | 0.46 | 0.47 | **0.57** | 0.18 | 0.25 | 0.70 | 5.86 | 3.18 |
| GPT-4o | 0.54 | 0.64 | **0.61** | 0.37 | 0.20 | 0.19 | 0.73 | **5.29** | **3.28** |
| Qwen-Max | 0.48 | 0.00 | 0.00 | 0.00 | 0.00 | 0.15 | 0.40 | 17.29 | 1.03 |
| Llama3.1-8B-FT | 0.16 | 0.35 | 0.24 | 0.00 | 0.05 | 0.10 | 0.35 | 18.43 | 1.25 |
| Qwen3-8B-FT | 0.49 | 0.46 | 0.32 | 0.40 | 0.09 | 0.18 | 0.65 | 10.00 | 2.58 |
| DemoDiff (Ours) | 0.50 | 0.57 | 0.42 | 0.49 | 0.12 | 0.23 | **0.75** | 6.29 | 3.09 |

Table 13: We compute the oracle of the top generation. The results are reported for the Drug Rediscovery task category. Best results in each column are **bolded**.

| Task | Celecoxib Rediscovery | Mestranol Similarity | Thiothixene Rediscovery | Troglitazone Rediscovery | Median 1 | Median 2 | Albuterol Similarity | Avg Rank | Total Sum |
|---|---|---|---|---|---|---|---|---|---|
| Molecular Optimization Methods with 100 Oracle Calls | | | | | | | | | |
| GraphGA | 0.27 | 0.28 | 0.28 | 0.23 | 0.23 | 0.23 | 0.40 | 10.86 | 1.93 |
| REINVENT | 0.34 | 0.39 | 0.32 | 0.25 | 0.23 | 0.23 | 0.41 | 8.57 | 2.18 |
| GPBO | 0.28 | 0.31 | 0.29 | 0.23 | 0.23 | 0.23 | 0.41 | 10.43 | 1.99 |
| STONED | 0.27 | 0.28 | 0.28 | 0.23 | 0.23 | 0.23 | 0.40 | 12.57 | 1.93 |
| Genetic GFN | 0.38 | 0.30 | 0.26 | 0.21 | 0.18 | 0.16 | 0.46 | 13.14 | 1.95 |
| GenMol | 0.39 | 0.37 | 0.31 | 0.27 | 0.36 | **0.38** | 0.56 | 6.86 | 2.65 |
| Molecular Optimization Methods with 10000 Predictor Calls | | | | | | | | | |
| GraphGA | 0.35 | 0.38 | 0.28 | 0.25 | 0.21 | 0.20 | 0.62 | 9.43 | 2.30 |
| REINVENT | 0.36 | 0.28 | 0.28 | 0.27 | 0.13 | 0.20 | **1.00** | 10.00 | 2.51 |
| GPBO | 0.25 | 0.57 | 0.26 | 0.20 | 0.13 | 0.20 | **1.00** | 12.00 | 2.61 |
| STONED | 0.27 | 0.64 | 0.28 | 0.23 | 0.13 | 0.20 | 0.22 | 12.71 | 1.96 |
| Genetic GFN | 0.28 | 0.13 | 0.17 | 0.15 | 0.06 | 0.10 | 0.19 | 18.00 | 1.09 |
| GenMol | **1.00** | 0.42 | 0.28 | 0.38 | 0.25 | 0.19 | 0.55 | 7.57 | 3.06 |
| Conditional Generation Models | | | | | | | | | |
| LSTM | **1.00** | 0.82 | 0.18 | 0.33 | 0.20 | 0.09 | 0.69 | 9.29 | 3.31 |
| Graph-DiT | 0.78 | 0.47 | 0.51 | 0.18 | **0.40** | 0.10 | 0.81 | 8.14 | 3.25 |
| Learning from In-Context Demonstrations | | | | | | | | | |
| DeepSeek-V3 | 0.77 | **1.00** | **0.84** | **0.75** | 0.12 | 0.17 | **1.00** | 6.00 | **4.64** |
| GPT-4o | 0.53 | 0.68 | 0.81 | 0.46 | 0.15 | 0.15 | 0.93 | 6.86 | 3.70 |
| Qwen-Max | 0.72 | 0.00 | 0.00 | 0.00 | 0.00 | 0.10 | 0.41 | 17.29 | 1.23 |
| Llama3.1-8B-FT | 0.13 | 0.31 | 0.18 | 0.00 | 0.07 | 0.09 | **1.00** | 15.57 | 1.78 |
| Qwen3-8B-FT | 0.77 | 0.58 | 0.27 | 0.43 | 0.10 | 0.15 | **1.00** | 9.14 | 3.30 |
| DemoDiff (Ours) | 0.84 | 0.64 | 0.34 | 0.65 | 0.14 | 0.16 | **1.00** | **5.57** | 3.77 |

Table 14: We compute the oracle and diversity scores from the Top-10 generations and report the harmonic mean of these two values. The results are reported for the Material Design task category. Best results in each column are **bolded**.

| Task
#  Tasks | Polymer CO2/CH4 | Polymer CO2/N2 | Polymer H2/CH4 | Polymer H2/N2 | Polymer O2/N2 | Avg Rank
33 | Total Sum
33 |
|---|---|---|---|---|---|---|---|
| Molecular Optimization Methods with 100 Oracle Calls | | | | | | | |
| GraphGA | 0.59 | 0.42 | 0.71 | 0.52 | 0.64 | 5.80 | 2.88 |
| REINVENT | 0.00 | 0.00 | 0.00 | 0.00 | 0.00 | 17.00 | 0.00 |
| GPBO | 0.46 | 0.31 | **0.76** | 0.79 | 0.68 | 3.80 | 3.00 |
| STONED | No pSMILES | No pSMILES | No pSMILES | No pSMILES | No pSMILES | 18.00 | 0.00 |
| Genetic GFN | No pSMILES | No pSMILES | No pSMILES | No pSMILES | No pSMILES | 14.40 | 0.00 |
| GenMol | 0.60 | 0.50 | 0.63 | 0.72 | 0.63 | 4.20 | 3.08 |
| Molecular Optimization Methods with 10000 Predictor Calls | | | | | | | |
| GraphGA | 0.60 | **0.55** | 0.75 | 0.40 | 0.44 | 5.20 | 2.75 |
| REINVENT | 0.32 | 0.37 | 0.56 | 0.69 | 0.61 | 7.20 | 2.55 |
| GPBO | 0.55 | 0.00 | 0.57 | 0.52 | 0.46 | 10.20 | 2.09 |
| STONED | No pSMILES | No pSMILES | No pSMILES | No pSMILES | No pSMILES | 19.20 | 0.00 |
| Genetic GFN | No pSMILES | No pSMILES | No pSMILES | No pSMILES | No pSMILES | 13.40 | 0.00 |
| GenMol | 0.07 | 0.31 | 0.57 | 0.12 | 0.00 | 11.60 | 1.07 |
| Conditional Generation Models | | | | | | | |
| LSTM | 0.00 | 0.29 | 0.16 | 0.12 | 0.23 | 13.60 | 0.80 |
| Graph-DiT | 0.27 | 0.55 | 0.74 | 0.57 | 0.63 | 5.40 | 2.76 |
| Learning from In-Context Demonstrations | | | | | | | |
| DeepSeek-V3 | 0.13 | 0.45 | 0.53 | 0.68 | 0.15 | 8.80 | 1.94 |
| GPT-4o | 0.37 | 0.30 | 0.44 | 0.70 | 0.33 | 8.60 | 2.14 |
| Qwen-Max | 0.00 | 0.09 | 0.00 | 0.00 | 0.42 | 16.20 | 0.51 |
| Llama3.1-8B-FT | 0.00 | 0.00 | 0.00 | 0.57 | 0.00 | 13.60 | 0.57 |
| Qwen3-8B-FT | 0.00 | 0.00 | 0.75 | 0.56 | 0.00 | 11.60 | 1.32 |
| DemoDiff (Ours) | **0.63** | 0.52 | 0.72 | **0.81** | **0.68** | **2.20** | **3.36** |

Table 15: We compute the oracle of the top generation. The results are reported for the Material Design task category. Best results in each column are **bolded**.

| Task | Polymer CO2/CH4 | Polymer CO2/N2 | Polymer H2/CH4 | Polymer H2/N2 | Polymer O2/N2 | Avg Rank | Total Sum |
|------|------|------|------|------|------|------|------|
| Molecular Optimization Methods with 100 Oracle Calls | | | | | | | |
| GraphGA | 0.79 | 0.41 | **1.00** | 0.57 | 0.84 | 7.00 | 3.60 |
| REINVENT | 0.00 | 0.00 | 0.00 | 0.00 | 0.00 | 17.20 | 0.00 |
| GPBO | 0.89 | 0.43 | **1.00** | **1.00** | 0.68 | 5.40 | 4.00 |
| STONED | No pSMILES | No pSMILES | No pSMILES | No pSMILES | No pSMILES | 18.20 | 0.00 |
| Genetic GFN | No pSMILES | No pSMILES | No pSMILES | No pSMILES | No pSMILES | 15.00 | 0.00 |
| GenMol | 0.76 | 0.70 | **1.00** | **1.00** | 0.88 | 3.20 | 4.34 |
| Molecular Optimization Methods with 10000 Predictor Calls | | | | | | | |
| GraphGA | 0.69 | 0.82 | **1.00** | 0.51 | 0.50 | 7.60 | 3.52 |
| REINVENT | 0.71 | 0.67 | **1.00** | **1.00** | 0.97 | 4.60 | 4.35 |
| GPBO | 0.67 | 0.00 | 0.80 | 0.85 | 0.57 | 11.20 | 2.90 |
| STONED | No pSMILES | No pSMILES | No pSMILES | No pSMILES | No pSMILES | 19.40 | 0.00 |
| Genetic GFN | No pSMILES | No pSMILES | No pSMILES | No pSMILES | No pSMILES | 14.00 | 0.00 |
| GenMol | 0.11 | 0.30 | **1.00** | 0.37 | 0.39 | 10.80 | 2.17 |
| Conditional Generation Models | | | | | | | |
| LSTM | 0.00 | 0.77 | 0.55 | 0.63 | 0.44 | 11.60 | 2.39 |
| Graph-DiT | 0.44 | **1.00** | **1.00** | 0.79 | 0.69 | 6.80 | 3.92 |
| Learning from In-Context Demonstrations | | | | | | | |
| DeepSeek-V3 | 0.35 | 0.60 | 0.95 | **1.00** | 0.86 | 7.60 | 3.76 |
| GPT-4o | 0.67 | 0.52 | **1.00** | **1.00** | 0.65 | 8.00 | 3.85 |
| Qwen-Max | 0.00 | 0.34 | 0.00 | 0.02 | 0.91 | 13.80 | 1.27 |
| Llama3.1-8B-FT | 0.00 | 0.32 | 0.00 | 0.67 | 0.00 | 13.60 | 0.99 |
| Qwen3-8B-FT | 0.00 | 0.00 | 0.98 | 0.89 | 0.00 | 13.00 | 1.87 |
| DemoDiff (Ours) | **1.00** | 0.64 | **1.00** | **1.00** | **1.00** | **2.00** | **4.64** |

Table 16: We compute the oracle of the top generation. The results are reported for the Structure Constrained Design task category. Best results in each column are **bolded**.

| Task | Isomers c7h8n2o2 | Isomers c9h10n2o2pf2cl | Deco Hop | Scaffold Hop | Valsartan Smarts | Avg Rank | Total Sum |
|------|------|------|------|------|------|------|------|
| Molecular Optimization Methods with 100 Oracle Calls | | | | | | | |
| GraphGA | 0.55 | 0.47 | 0.58 | 0.45 | **0.23** | 9.80 | 2.29 |
| REINVENT | 0.55 | 0.44 | 0.58 | 0.45 | **0.23** | 10.00 | 2.26 |
| GPBO | 0.55 | 0.50 | 0.59 | 0.47 | **0.23** | 8.20 | 2.34 |
| STONED | 0.55 | 0.47 | 0.58 | 0.45 | **0.23** | 11.60 | 2.29 |
| Genetic GFN | 0.29 | 0.54 | 0.58 | 0.45 | 0.00 | 12.00 | 1.86 |
| GenMol | 0.80 | 0.69 | 0.58 | 0.46 | 0.00 | 10.00 | 2.53 |
| Molecular Optimization Methods with 10000 Predictor Calls | | | | | | | |
| GraphGA | **1.00** | 0.78 | 0.54 | 0.39 | 0.00 | 11.40 | 2.72 |
| REINVENT | **1.00** | 0.00 | 0.62 | 0.47 | 0.00 | 9.20 | 2.09 |
| GPBO | 0.88 | 0.47 | 0.55 | 0.45 | 0.00 | 13.80 | 2.36 |
| STONED | 0.82 | 0.72 | 0.57 | 0.47 | 0.00 | 10.80 | 2.57 |
| Genetic GFN | 0.00 | 0.00 | 0.56 | 0.39 | 0.00 | 17.20 | 0.95 |
| GenMol | 0.00 | 0.02 | 0.84 | 0.47 | 0.03 | 10.00 | 1.36 |
| Conditional Generation Models | | | | | | | |
| LSTM | **1.00** | **0.87** | 0.54 | **0.78** | 0.00 | 8.60 | **3.18** |
| Graph-DiT | 0.88 | 0.82 | 0.53 | 0.77 | 0.00 | 10.20 | 3.01 |
| Learning from In-Context Demonstrations | | | | | | | |
| DeepSeek-V3 | 0.72 | 0.50 | 0.84 | 0.41 | 0.21 | 10.00 | 2.68 |
| GPT-4o | 0.74 | 0.82 | 0.59 | 0.46 | 0.00 | 7.80 | 2.61 |
| Qwen-Max | 0.90 | 0.73 | 0.56 | 0.45 | 0.01 | 9.40 | 2.67 |
| Llama3.1-8B-FT | 0.00 | 0.00 | 0.85 | 0.41 | 0.00 | 14.80 | 1.26 |
| Qwen3-8B-FT | 0.05 | 0.04 | **0.87** | 0.47 | 0.00 | 10.40 | 1.42 |
| DemoDiff (Ours) | 0.88 | 0.73 | 0.86 | 0.49 | 0.00 | **4.80** | 2.96 |

Table 17: We compute the oracle and diversity scores from the Top-10 generations and report the harmonic mean of these two values. The results are reported for the Structure Constrained Design task category. Best results in each column are **bolded**.

| Task
# Tasks | Isomers
c7h8n2o2 | Isomers
c9h10n2o2pf2cl | Deco Hop | Scaffold
Hop | Valsartan
Smarts | Avg
Rank
33 | Total
Sum
33 |
|---|---|---|---|---|---|---|---|
| Molecular Optimization Methods with 100 Oracle Calls | | | | | | | |
| GraphGA | 0.30 | 0.44 | 0.68 | 0.57 | **0.14** | 7.60 | 2.13 |
| REINVENT | 0.36 | 0.40 | 0.68 | 0.57 | 0.14 | 8.00 | 2.15 |
| GPBO | 0.27 | 0.45 | 0.68 | 0.57 | 0.14 | 7.00 | 2.11 |
| STONED | 0.18 | 0.39 | 0.68 | 0.57 | **0.14** | 9.00 | 1.97 |
| Genetic GFN | 0.15 | 0.50 | 0.68 | 0.57 | 0.00 | 9.60 | 1.89 |
| GenMol | 0.39 | 0.56 | 0.60 | 0.54 | 0.00 | 11.40 | 2.08 |
| Molecular Optimization Methods with 10000 Predictor Calls | | | | | | | |
| GraphGA | 0.71 | 0.69 | 0.42 | 0.45 | 0.00 | 11.60 | 2.27 |
| REINVENT | 0.49 | 0.00 | 0.46 | 0.30 | 0.00 | 16.00 | 1.24 |
| GPBO | 0.69 | 0.29 | 0.65 | 0.53 | 0.00 | 12.80 | 2.16 |
| STONED | 0.62 | 0.65 | 0.68 | 0.57 | 0.00 | 9.20 | 2.51 |
| Genetic GFN | 0.00 | 0.00 | 0.21 | 0.17 | 0.00 | 17.80 | 0.38 |
| GenMol | 0.00 | 0.01 | 0.63 | 0.42 | 0.01 | 14.60 | 1.06 |
| Conditional Generation Models | | | | | | | |
| LSTM | 0.80 | 0.75 | 0.65 | 0.55 | 0.00 | 9.20 | 2.75 |
| Graph-DiT | **0.85** | 0.74 | 0.66 | **0.66** | 0.00 | 7.40 | **2.91** |
| Learning from In-Context Demonstrations | | | | | | | |
| DeepSeek-V3 | 0.64 | 0.47 | 0.70 | 0.52 | 0.09 | 7.60 | 2.43 |
| GPT-4o | 0.61 | **0.78** | 0.67 | 0.54 | 0.00 | 8.00 | 2.60 |
| Qwen-Max | 0.70 | 0.55 | 0.00 | 0.36 | 0.01 | 11.60 | 1.62 |
| Llama3.1-8B-FT | 0.00 | 0.00 | **0.71** | 0.51 | 0.00 | 14.00 | 1.21 |
| Qwen3-8B-FT | 0.01 | 0.02 | 0.70 | 0.55 | 0.00 | 11.20 | 1.28 |
| DemoDiff (Ours) | 0.82 | 0.77 | 0.64 | 0.57 | 0.00 | **6.40** | 2.80 |

Table 18: We compute the oracle of the top generation. The results are reported for the Target Based Design task category. Best results in each column are **bolded**.

| Task | Docking Braf | Docking Parp1 | Docking Jak2 | Docking Fa7 | Docking 5HT1B | Avg Rank | Total Sum |
|---|---|---|---|---|---|---|---|
| Molecular Optimization Methods with 100 Oracle Calls | | | | | | | |
| GraphGA | 0.75 | 0.76 | 0.74 | 0.67 | 0.76 | 8.80 | 3.69 |
| REINVENT | 0.77 | 0.80 | 0.69 | 0.66 | 0.80 | 7.60 | 3.73 |
| GPBO | 0.79 | 0.83 | 0.72 | 0.64 | 0.78 | 7.00 | 3.76 |
| STONED | 0.74 | 0.77 | 0.77 | 0.64 | 0.76 | 9.20 | 3.69 |
| Genetic GFN | 0.79 | 0.79 | 0.71 | 0.70 | 0.74 | 7.60 | 3.73 |
| GenMol | 0.79 | 0.79 | 0.70 | 0.69 | 0.87 | 6.20 | 3.84 |
| Molecular Optimization Methods with 10000 Predictor Calls | | | | | | | |
| GraphGA | **0.84** | 0.91 | 0.79 | 0.66 | **0.95** | 3.00 | 4.15 |
| REINVENT | 0.68 | **1.00** | 0.59 | 0.69 | 0.00 | 11.60 | 2.96 |
| GPBO | 0.84 | 0.91 | **0.84** | **0.71** | 0.91 | **2.00** | **4.21** |
| STONED | 0.71 | 0.00 | 0.61 | 0.00 | 0.00 | 18.20 | 1.32 |
| Genetic GFN | 0.78 | 0.71 | 0.00 | 0.44 | 0.00 | 15.60 | 1.93 |
| GenMol | 0.74 | 0.76 | 0.72 | 0.51 | 0.79 | 11.40 | 3.51 |
| Conditional Generation Models | | | | | | | |
| LSTM | 0.74 | 0.76 | 0.63 | 0.61 | 0.92 | 11.00 | 3.66 |
| Graph-DiT | 0.72 | 0.77 | 0.66 | 0.59 | 0.74 | 12.80 | 3.49 |
| Learning from In-Context Demonstrations | | | | | | | |
| DeepSeek-V3 | 0.80 | 0.72 | 0.81 | 0.52 | 0.72 | 10.00 | 3.57 |
| GPT-4o | 0.77 | 0.72 | 0.64 | 0.57 | 0.76 | 12.80 | 3.46 |
| Qwen-Max | 0.54 | 0.00 | 0.00 | 0.51 | 0.00 | 19.00 | 1.06 |
| Llama3.1-8B-FT | 0.00 | 0.00 | 0.42 | 0.00 | 0.00 | 18.40 | 0.42 |
| Qwen3-8B-FT | 0.76 | 0.73 | 0.75 | 0.56 | 0.71 | 12.00 | 3.51 |
| DemoDiff (Ours) | 0.81 | 0.86 | 0.79 | 0.60 | 0.77 | 5.80 | 3.84 |

Table 19: We compute the oracle and diversity scores from the Top-10 generations and report the harmonic mean of these two values. The results are reported for the Target Based Design task category. Best results in each column are **bolded**.

| Task | Docking Braf | Docking Parp1 | Docking Jak2 | Docking Fa7 | Docking 5HT1B | Avg Rank | Total Sum |
|------|------|------|------|------|------|------|------|
| # Tasks | | | | | | 33 | 33 |
| Molecular Optimization Methods with 100 Oracle Calls | | | | | | | |
| GraphGA | 0.78 | 0.79 | 0.76 | 0.70 | 0.79 | 3.20 | 3.82 |
| REINVENT | 0.77 | 0.79 | 0.73 | 0.71 | 0.78 | 4.40 | 3.78 |
| GPBO | 0.77 | 0.79 | 0.76 | 0.70 | 0.78 | 4.40 | 3.79 |
| STONED | 0.77 | 0.78 | 0.77 | 0.70 | 0.78 | 4.60 | 3.80 |
| Genetic GFN | 0.78 | 0.79 | 0.75 | 0.70 | 0.77 | 4.20 | 3.79 |
| GenMol | 0.69 | 0.70 | 0.66 | 0.67 | 0.73 | 10.80 | 3.45 |
| Molecular Optimization Methods with 10000 Predictor Calls | | | | | | | |
| GraphGA | 0.66 | 0.70 | 0.60 | 0.57 | 0.70 | 13.20 | 3.22 |
| REINVENT | 0.19 | 0.27 | 0.31 | 0.09 | 0.00 | 17.60 | 0.86 |
| GPBO | 0.69 | 0.78 | 0.76 | **0.73** | 0.74 | 6.80 | 3.70 |
| STONED | 0.48 | 0.00 | 0.52 | 0.00 | 0.00 | 17.60 | 1.00 |
| Genetic GFN | 0.34 | 0.20 | 0.00 | 0.19 | 0.00 | 17.40 | 0.73 |
| GenMol | 0.47 | 0.55 | 0.52 | 0.48 | 0.59 | 15.80 | 2.60 |
| Conditional Generation Models | | | | | | | |
| LSTM | 0.73 | 0.73 | 0.71 | 0.66 | 0.77 | 8.40 | 3.61 |
| Graph-DiT | 0.71 | 0.74 | 0.70 | 0.66 | 0.75 | 9.40 | 3.55 |
| Learning from In-Context Demonstrations | | | | | | | |
| DeepSeek-V3 | 0.63 | 0.64 | 0.63 | 0.57 | 0.73 | 13.20 | 3.19 |
| GPT-4o | 0.77 | 0.74 | 0.70 | 0.66 | 0.76 | 8.20 | 3.63 |
| Qwen-Max | 0.48 | 0.00 | 0.00 | 0.49 | 0.00 | 18.20 | 0.96 |
| Llama3.1-8B-FT | 0.00 | 0.00 | 0.11 | 0.00 | 0.00 | 18.40 | 0.11 |
| Qwen3-8B-FT | 0.70 | 0.73 | 0.66 | 0.55 | 0.72 | 12.00 | 3.35 |
| DemoDiff (Ours) | **0.81** | **0.82** | **0.78** | 0.69 | **0.79** | **2.20** | **3.90** |

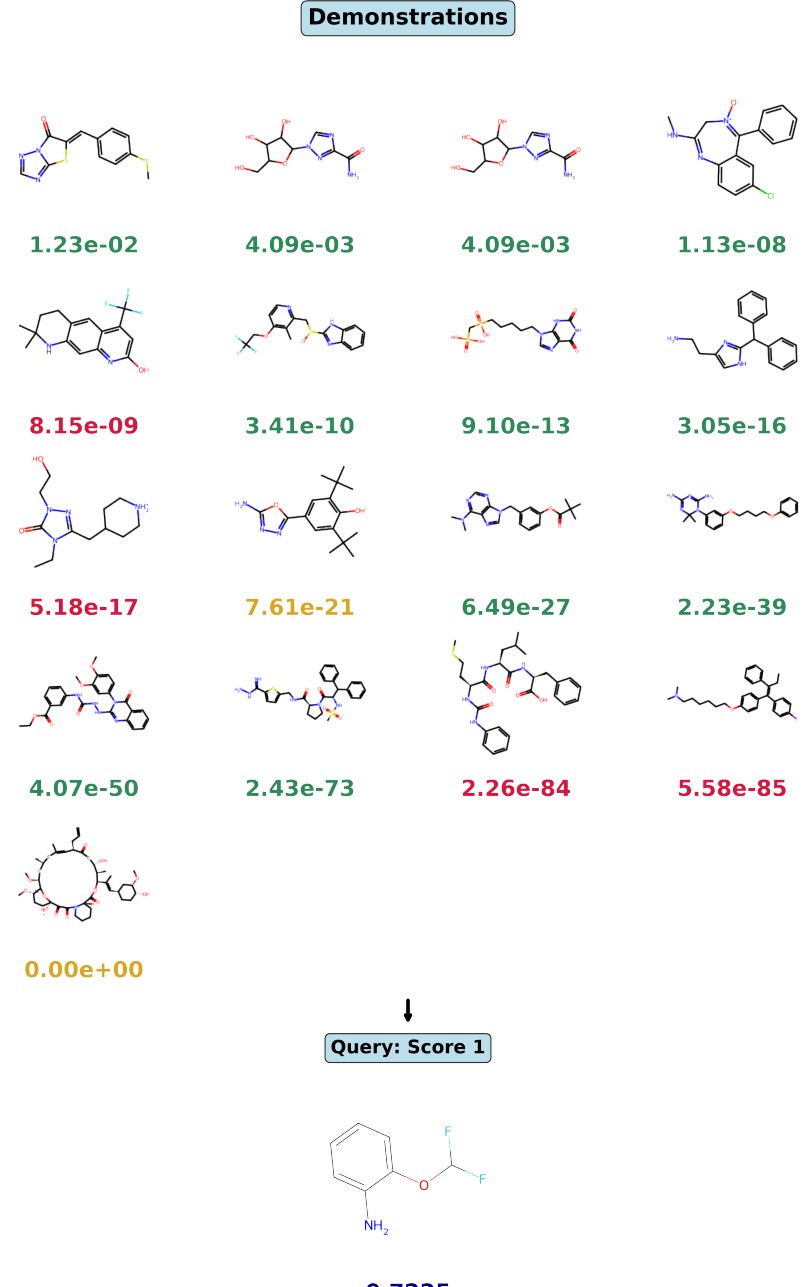

Figure 14: Structure constrained generation for Isomer with all negative demonstrations.

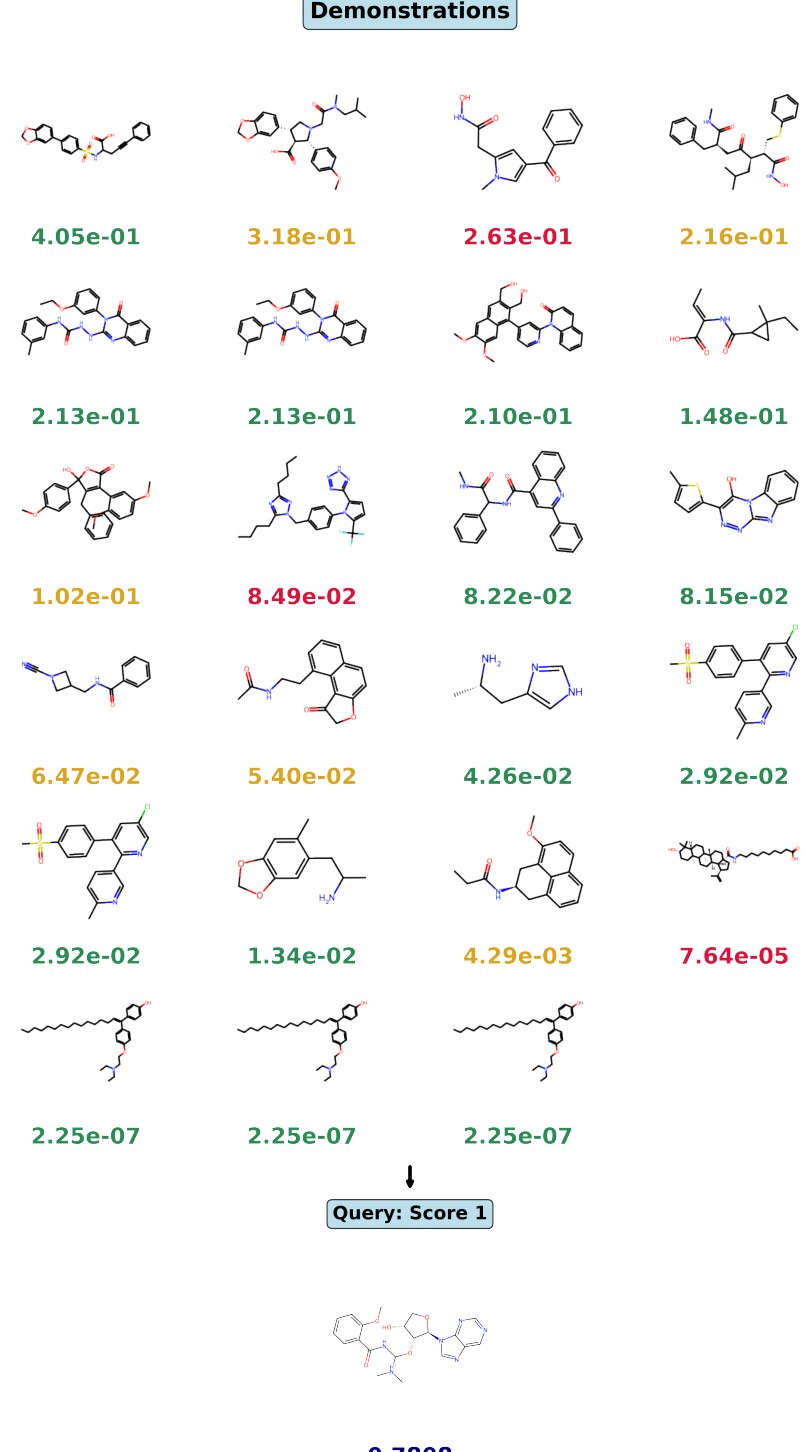

Figure 15: Drug MPO for Osimertinib with all negative demonstrations.

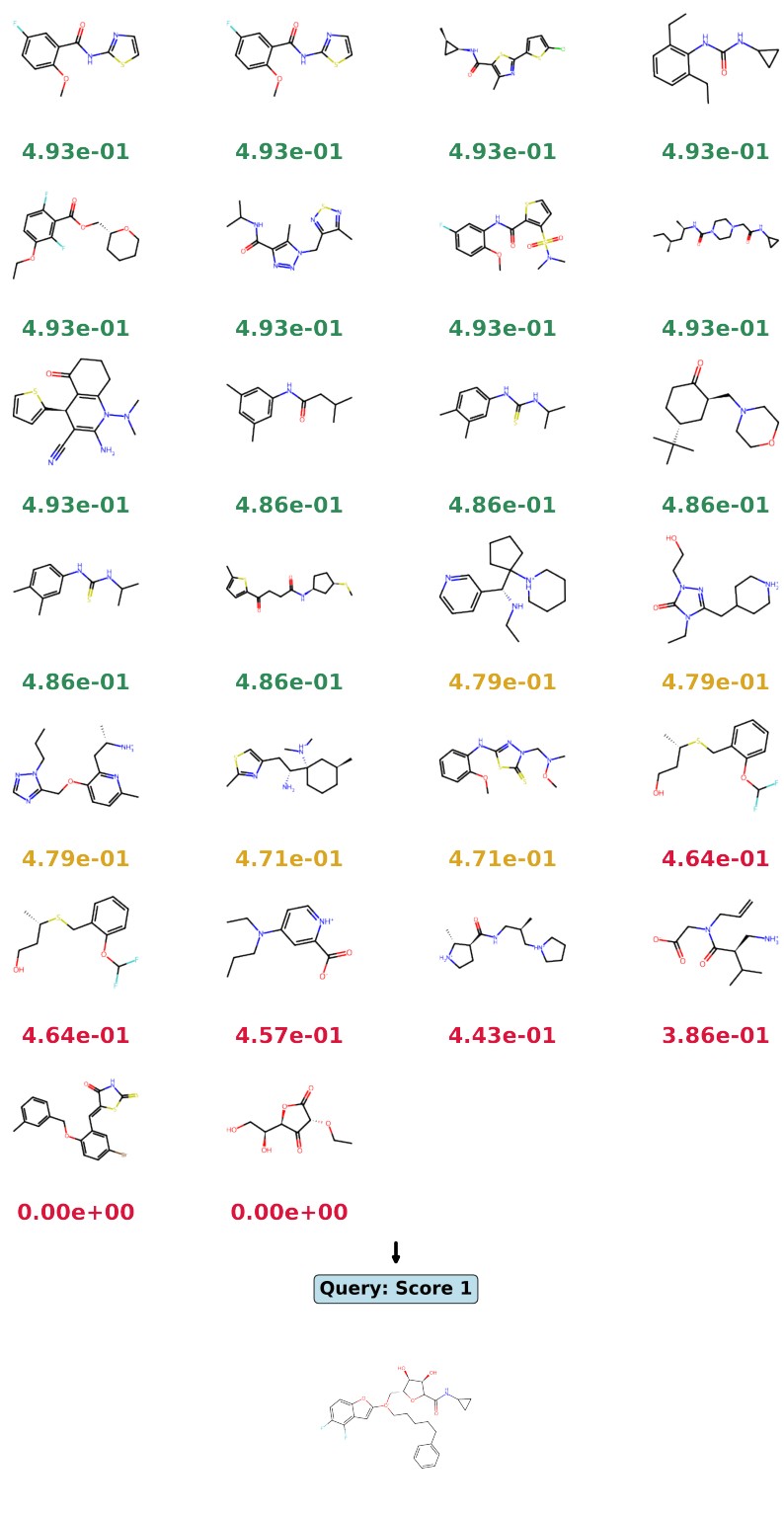

Figure 16: Target-based design for PARP1 with all negative demonstrations.

