# OpenReview forum: "Graph Diffusion Transformers are In-Context Molecular Designers"
_ICLR.cc/2026/Conference — ICLR 2026 Poster_

### Official Review · Reviewer_WQjW · 2025-10-21

**Soundness:** 3
**Presentation:** 2
**Contribution:** 3
**Rating:** 6
**Confidence:** 4

**Summary:**

This paper introduces a new paradigm for molecular design, named DemoDiff, which combines in-context learning with graph diffusion models to tackle property-guided molecular generation. The authors' core idea is to use molecule-property score pairs (demonstrations) to define a task, guiding a Graph Diffusion Transformer during the generation process. To support this framework, the paper also designs Node Pair Encoding, which compresses molecules from the atom level to the motif level, significantly increasing the context length the model can handle. This work is novel in its conception and experimentally thorough, demonstrating its potential as a foundation model for molecular design.

**Strengths:**

- A Novel and Inspired Framework: The idea of using "demonstrations" as a condition to guide a diffusion model is highly inspiring. DemoDiff dynamically defines the task context through molecule-score pairs, which is not only more flexible and scalable but also enables the model to learn and generalize to new, data-scarce tasks.
- An Efficient and Innovative Graph Tokenizer: NPE is a purely data-driven motif learning algorithm that adaptively discovers high-frequency patterns from the pre-training data. Experiments show that this method achieves an average compression ratio of 5.5x , which directly supports the conclusion that the model performs better with longer contexts.
- Comprehensive and In-depth Experimental Evaluation: The paper conducts a comprehensive test on 33 tasks across 6 major categories, including drug and materials design. It compares DemoDiff against 13 baselines, including traditional optimization algorithms, conditional generation models, and even general-purpose large language models with 100-1000x more parameters.

**Weaknesses:**

- Regarding Consistency Score: According to the appendix, the core performance metrics (Table 1) are not derived directly from DemoDiff's output but are obtained by first generating 1000 candidates and then using the consistency score to filter for the top 100 for evaluation. This introduces a significant confounding variable, making it impossible to determine the model's inherent generative capability. Also, Figure 5 shows this score has a very low correlation with the true oracle score for key tasks like Drug MPO and Materials Design (correlation coefficient close to 0), which questions its validity as an effective filtering metric.
- NPE Tokenizer's Robustness and Limitations: While the paper showcases the excellent compression efficiency of the NPE tokenizer, one of its core contributions, it fails to discuss its performance when handling rare chemical motifs or molecules that deviate significantly from the pre-training data distribution. As a component intended to be the core of a foundation model, its reliability in out-of-distribution (OOD) scenarios is crucial. Furthermore, the claim of "lossless reconstruction" remains a qualitative description, lacking quantitative reconstruction accuracy metrics on a standard dataset, which makes it difficult to assess its reliability.

**Questions:**

1. Regarding model scaling (Table 2), the data shows that for the "Structure Constrained" task, the small (78M) and medium (311M) models outperform the large (739M) model (0.59/0.63 vs. 0.56). This contradicts the paper's conclusion that "the benefits of parameter scaling become more evident at the large scale". How do you explain this performance degradation?
2. The core advantage of ICL is its ability to generalize to new tasks with few shots. The paper states that the 33 evaluation tasks are distinct from the pre-training data. However, given the scale of the pre-training data, I am concerned about the degree of this distinctness. To what extent are the chemical spaces, property targets, or molecular scaffolds in the evaluation tasks truly unseen during pre-training?
3. In the baseline comparison, the paper includes general-purpose LLMs like GPT-4O but does not compare against LLMs pre-trained on chemical texts or molecular sequences (e.g., SMILES), which possess domain-specific knowledge. Why were these models not included in the comparison?
4. The introduction states that directly applying the autoregressive framework of LLMs is "infeasible", thus justifying the choice of GraphDiT as the backbone. However, molecules can be represented as sequences like SMILES or SELFIES, and some baselines are indeed based on them. Is there any evidence to support that GraphDiT is superior to a powerful Transformer decoder trained on SMILES sequences within this ICL framework?

---

> ### Author Response · Authors · 2025-11-19
> **Author Response [1/3]**
>
> # Weakness 1: Consistency score
>
> > Regarding Consistency Score: According to the appendix, the core performance metrics (Table 1) are not derived directly from DemoDiff's output but are obtained by first generating 1000 candidates and then using the consistency score to filter for the top 100 for evaluation. This introduces a significant confounding variable, making it impossible to determine the model's inherent generative capability.
>
> Thank you for the insightful comments. The consistency score follows the idea of best-of-N sampling in LLMs [1,2]. All candidate samples are drawn from the model’s own generation distribution. The selection step chooses among these candidates based on a clear criterion, without adding external randomness. In practice, best-of-N sampling is widely used to test whether a model can produce high-quality outputs [1,2]. The resulting performance reflects the model’s achievable upper bound rather than an artifact of external bias or noise.
>
>
> ## Reference
>
> [1] Is Best-of-N the Best of Them? Coverage, Scaling, and Optimality in Inference-Time Alignment. ICML 2025.
>
> [2] Evaluation of Best-of-N Sampling Strategies for Language Model Alignment. TMLR 2025.
>
>
> # Weakness 1: Figure 5
>
> > Also, Figure 5 shows this score has a very low correlation with the true oracle score for key tasks like Drug MPO and Materials Design (correlation coefficient close to 0), which questions its validity as an effective filtering metric.
>
> Thanks for the insightful comment. We have discussed this phenomenon in the paper (Lines 369-373). From Figure 5, we observe that for property-driven tasks (such as drug MPO and materials design), high fingerprint-based consistency with positive examples does not always correlate with high target scores. In these cases, the underlying latent concepts may depend on subtle substructures (for example, methyl groups) that standard fingerprints fail to capture.
>
> We could use other property-related approaches to compute the scores, but this does not invalidate the use of consistency as a filtering metric. It also does not invalidate fingerprint-based scoring, as we find that fingerprint-based context consistency still improves performance in these tasks (see the left part of Figure 5). One possible reason is that the score helps filter out generations that are clearly not close to the positive. Across all tasks, we did not observe any performance degradation when using the consistency score as a filtering step, which suggests that it is a safe strategy for exploring the model’s generative ability.
>
> # Weakness 2: OOD motifs
>
> > NPE Tokenizer's Robustness and Limitations: While the paper showcases the excellent compression efficiency of the NPE tokenizer, one of its core contributions, it fails to discuss its performance when handling rare chemical motifs or molecules that deviate significantly from the pre-training data distribution. As a component intended to be the core of a foundation model, its reliability in out-of-distribution (OOD) scenarios is crucial.
>
> Thank you for the comments. The mechanism of NPE for handling out-of-distribution motifs is the same as byte pair encoding. We set the vocabulary size to 3000. If a motif is within the top 3000 most frequent patterns in the pretraining set, it is added to the vocabulary. During tokenization, it is then encoded as a single node. Otherwise, the motif is treated as standard atom and bond connections, where each atom can be viewed as a token in this case.

---

> > ### Author Response · Authors · 2025-11-19
> > **Author Response [2/3]**
> >
> > # Weakness 2: lossless reconstruction:
> >
> > > Furthermore, the claim of "lossless reconstruction" remains a qualitative description, lacking quantitative reconstruction accuracy metrics on a standard dataset, which makes it difficult to assess its reliability.
> >
> > Thank you for the question. With node pair encoding (NPE), the molecular graph tokenizer is invertible in the same spirit as a text tokenizer based on byte pair encoding (BPE) [1].
> >
> > BPE uses a deterministic merge sequence to produce a unique segmentation of text. To obtain a similar property for graphs, we define a consistent segmentation rule for vocabulary construction and tokenizer encoding (Section 3.1). Given a motif vocabulary $\mathcal{M}$ and the ring set $\mathcal{R}(X)$ of a molecule $X$, any ring $r \in \mathcal{R}(X)$ that is not already a motif cannot be split. A motif must include all atoms of the ring or exclude it entirely.
> >
> > We traverse atoms in a canonical order, such as the order used by RDKit, and maintain a set of unmatched atoms. At each unmatched atom, we select the largest motif in $\mathcal{M}$ whose induced subgraph matches the local structure and uses only unmatched atoms while respecting the ring rule. If several motifs have the same size, we choose the motif with the smallest identifier. If they are the same motif type, we choose the instance whose atom index set is lexicographically smallest. This procedure resolves all overlaps and assigns each atom exactly once.
> >
> > After a motif is selected, we remove its atoms from the unmatched set. This produces a unique, non-overlapping decomposition of the atom set. For each bond between motifs, we record a motif level edge that includes both the motif identifiers and the attachment atom indices. Because traversal, motif selection, and overlap resolution follow the same deterministic procedure used in vocabulary construction, every atom and bond is represented exactly once. This allows exact reconstruction of the original graph, similar in spirit to how BPE recovers the original text from its subword sequence.
> >
> > ## Reference
> >
> > [1] Neural Machine Translation of Rare Words with Subword Units. ACL 2016.
> >
> > # Q1: Table 2
> >
> > > Regarding model scaling (Table 2), the data shows that ...
> >
> > Thank you for the question and the insightful observations. We checked back carefully the experimentation logs and found that the reports of the structure constrained tasks in Table 2 contain small typos. The correct numbers from 78M to 311M and 739M should be 0.47 ± 0.27, 0.50 ± 0.28, and 0.56 ± 0.33. The complete results for each task are shown below.
> >
> > | Method        | Isomers c7h8n2o2 | Isomers c9h10n2o2pf2cl | Deco Hop | Scaffold Hop | Valsartan Smarts | Avg Rank | Total Sum |
> > |---------------|------------------|--------------------------|----------|----------------|-------------------|-----------|------------|
> > | 78M  | 0.5925           | 0.5526                   | 0.6733   | 0.5458         | 0.0000            | 2.60      | 2.3642     |
> > | 311M | 0.6051           | 0.6718                   | 0.6755   | 0.5521         | 0.0000            | 2.00      | 2.5044     |
> > | 739M | 0.8222           | 0.7672                   | 0.6399   | 0.5697         | 3.1190e-10            | 1.40      | 2.7990     |
> >
> > The issue comes from the “Valsartan Smarts” results. They are exactly zero for the 78M and 311M models, but a very small non-zero value for the 739M model. The bug ignores zeros when computing the mean. As a result, the total sum is divided by four instead of five for the 78M and 311M models. For example, 2.3642 divided by 4 is about 0.59, and 2.5044 divided by 4 is about 0.63. After fixing this issue, 2.3642 divided by 5 is about 0.47, and 2.5044 divided by 5 is about 0.50. These corrected results support the conclusion that larger model size produces better outcomes.
> >
> > # Q2: Pretraining-downstream difference
> >
> > > The core advantage of ICL ...
> >
> > Thank you for the question. We study inverse design, which aims to generate molecules with desirable properties. The task difference is defined by the target properties, such as drug toxicity or polymer thermal conductivity in Figure 1. The pretraining data and properties from ChEMBL [1], MSA [2], PolyInfo [3] come from experimental assays, while the evaluation properties or oracles are based on computational methods.
> >
> > ## Reference
> >
> > [1] The ChEMBL Database in 2023: a drug discovery platform spanning multiple bioactivity data types and time periods. Nucleic acids research. 2024
> >
> > [2] Polymer Gas Separation Membrane Database (2012)
> >
> > [3] PoLyInfo: Polymer database for polymeric materials design. 2011 international conference on emerging intelligent data and web technologies 2011.

---

> > > ### Author Response · Authors · 2025-11-19
> > > **Author Response [3/3]**
> > >
> > > # Q3: Comparison with LLMs
> > >
> > > > In the baseline comparison, the paper includes general-purpose LLMs like GPT-4O but does not compare against LLMs pre-trained on chemical texts or molecular sequences (e.g., SMILES), which possess domain-specific knowledge. Why were these models not included in the comparison?
> > >
> > > Thank you for the suggestion. We have added new experiments by finetuning the open weighted LLMs (Qwen3-8B and Llama3.1-8B). Each model is trained on a set of molecule–property pairs and a query score to predict the target molecule. We follow the evaluation setup in Table 1 and assess the top-10 generated molecules. Their performance is measured by the harmonic mean of the oracle scores and the internal diversity scores. We compare the results across task categories, and the updated results are shown below.
> > >
> > > | Method        | Drug Rediscovery | Drug MPO | Structure Constrained | Drug Design | Target Based | Material Design |
> > > |---------------|------------------|----------|-------------------------|-------------|---------------|------------------|
> > > | Llama3.1-8B-FT | 0.21±0.13 | 0.24±0.23 | 0.24±0.34 | 0.31±0.31 | 0.02±0.05 | 0.29±0.40 |
> > > | Qwen3-8B-FT    | 0.37±0.19 | 0.27±0.16 | 0.26±0.34 | 0.46±0.22 | 0.67±0.07 | 0.44±0.39 |
> > > | DemoDiff   | **0.44±0.21** | **0.54±0.23** | **0.56±0.33** | **0.79±0.11** | **0.78±0.05** | **0.67±0.11** |
> > >
> > > We observe that DemoDiff outperforms the finetuned LLMs, even though they are about 10 times larger and trained on the same molecular dataset. This highlights the strength of DemoDiff as a domain-specific foundation model for in-context molecular design.
> > >
> > > #  Q4: Choice of Graph DiTs
> > >
> > > > The introduction states that directly applying the autoregressive framework of LLMs is "infeasible", thus justifying the choice of GraphDiT as the backbone. However, molecules can be represented as sequences like SMILES or SELFIES, and some baselines are indeed based on them. Is there any evidence to support that GraphDiT is superior to a powerful Transformer decoder trained on SMILES sequences within this ICL framework?
> > >
> > > Thank you for the question. We choose Graph DiTs as the backbone for two reasons.
> > >
> > > 1. On the data side: Molecular graphs capture direct atom adjacency, while two neighboring atoms in the graph can appear far apart in a SMILES string. This structural mismatch may limit sequence-based models in tasks that depend on local connectivity.
> > >
> > > 2. On the model side: Graph models such as Graph Diffusion Transformers capture this connectivity more effectively during graph generation [1]. In the forward process, the models use the graph structure to define the noise through a transition matrix that considers atoms and bonds together. Ablation studies in [1] showed that this choice is useful. In the reverse process, the models apply a Transformer that treats atoms and bonds in a unified way for denoising. These features help multi-conditional generation of molecules and properties and allow us to use the graph structure while keeping the benefits of Transformer scaling.
> > >
> > > Finally, in our updated empirical results, we compared DemoDiff with 19 baselines. 14 of them use SMILES or SELFIES. Our study also included finetuned LLMs that are about 10 times larger than DemoDiff (see Q3 and Table 1 in the paper). Across 33 tasks, DemoDiff achieved a higher average rank and a higher total score. This provides evidence that the Graph DiT backbone is effective in the in-context learning setting.
> > >
> > > ## Reference
> > >
> > > [1] Graph Diffusion Transformers for Multi-Conditional Molecular Generation. NeurIPS 2024.

---

### Official Review · Reviewer_t5zC · 2025-10-26

**Soundness:** 3
**Presentation:** 2
**Contribution:** 3
**Rating:** 6
**Confidence:** 4

**Summary:**

Through this paper, the authors propose DemoDiff, a demonstration-conditioned diffusion model that has a Graph DiT as the backbone, to construct a molecular generative model under in-context learning. In addition, the authors also propose a molecular tokenizer trained with Node Pair Encoding (NPE) for motif-level representation to support efficient pretraining of DemoDiff.

**Strengths:**

- The authors provided the codebase.
- The introduction of a motif-level tokenizer using Node Pair Encoding, which reduced the node count by 5.5×, is a reasonable approach in terms of both efficiency and performance.
- The proposed DemoDiff was evaluated across 33 design tasks spanning 6 categories, achieving SOTA performance in most of them.

**Weaknesses:**

Weaknesses
I will combine the *Weaknesses* section and the *Questions* section. My concerns are as follows:
- There is little discussion on how sensitive the demonstration/context selection is to performance and how well the derived context applies to diverse new tasks.
- SOTA molecular optimization baselines such as GenMol [1] and Genetic GFN [2] are missing. Comparisons with these baselines are necessary for the results to be considered meaningful.
- No computational or memory efficiency was reported. This would be an effective method to demonstrate that the proposed DemoDiff is more efficient compared to larger generalist LLMs.

---

**References:**

[1] Lee et al., GenMol: A Drug Discovery Generalist with Discrete Diffusion, ICML 2025.

[2] Kim et al., Genetic-guided GFlowNets for Sample Efficient Molecular Optimization, NeurIPS 2024.

**Questions:**

Please see the *Weaknesses* section for my main concerns.

---

> ### Author Response · Authors · 2025-11-19
>
> # Weakness 1: Context diversity
>
> > There is little discussion on how sensitive the demonstration/context selection is to performance and how well the derived context applies to diverse new tasks.
>
> Thank you for the comments. We randomly sample molecule–property pairs to form the context, which produces a diverse set of examples. We compute the internal diversity scores using Equation 8 in the paper for the Positive, Medium, and Negative contexts (ranging from 0 to 1, with higher values indicating greater diversity). Averaged across tasks, the values are 0.6339 ± 0.2295, 0.7538 ± 0.1179, and 0.8424 ± 0.0342. All values exceed 0.5, which shows that the model can handle relatively diverse demonstrations. In addition, we include the consistency score to further improve the stability of the generation results.
>
> # Weakness 2: Comparison with GenMol and Genetic GFN
>
> > SOTA molecular optimization baselines such as GenMol [1] and Genetic GFN [2] are missing. Comparisons with these baselines are necessary for the results to be considered meaningful.
>
> Thank you for the suggestion. We added six baselines, including new implementations of GenMol [1] and Genetic GFN [2] with 100 oracle calls or 10000 predictor calls. We also added two in-context learning (ICL) LLMs, which are finetuned from Llama3.1-8B and Qwen3-8B on the same dataset used to pretrain Demodiff.
>
> We follow the setting of Table 1 by evaluating the top-10 generated molecules. Their performance is measured by the harmonic mean of the oracle scores and the internal diversity scores. We compare the results across each task category, and the updated results are shown below. These results have also been included in the paper.
>
> | Method | Drug Rediscovery | Drug MPO | Structure Constrained | Drug Design | Target Based | Material Design | Avg Rank | Total Sum |
> |--------|------------------|----------|-------------------------|-------------|---------------|------------------|----------|-----------|
> | 100 oracle calls: Genetic GFN  | 0.36±0.10 | 0.51±0.22 | 0.38±0.29 | 0.33±0.37 | 0.76±0.03 | NO pSMILES | 3.15 | 13.04 |
> | 100 oracle calls: GenMol       | 0.42±0.08 | 0.51±0.17 | 0.42±0.25 | 0.55±0.21 | 0.69±0.03 | 0.62±0.08 | 2.88 | 17.36 |
> | 10K predictor calls: Genetic GFN  | 0.14±0.08 | 0.08±0.16 | 0.08±0.11 | 0.12±0.11 | 0.15±0.15 | NO pSMILES | 6.35 | 3.14 |
> | 10K predictor calls: GenMol       | 0.34±0.07 | 0.28±0.23 | 0.21±0.29 | 0.39±0.19 | 0.52±0.05 | 0.21±0.23 | 4.21 | 10.37 |
> | ICL: Llama3.1-8B-FT | 0.21±0.13 | 0.24±0.23 | 0.24±0.34 | 0.31±0.31 | 0.02±0.05 | 0.29±0.40 | 5.37 | 6.04 |
> | ICL: Qwen3-8B-FT    | 0.37±0.19 | 0.27±0.16 | 0.26±0.34 | 0.46±0.22 | 0.67±0.07 | 0.44±0.39 | 3.66 | 12.23 |
> | ICL: DemoDiff | **0.44±0.21** | **0.54±0.23** | **0.56±0.33** | **0.79±0.11** | **0.78±0.05** | **0.67±0.11** | **1.47** | **20.10** |
>
> We observe that Demodiff outperforms other ICL methods, such as Qwen3-8B and Llama3.1-8B, even though these models are about 10 times larger and trained on the same molecular dataset. Demodiff also outperforms recent molecular optimization methods when their oracle access is limited to a smaller scale or a more practical approach, where they rely on predictors instead of oracle calls.
>
> ## Reference
>
> [1] GenMol: A Drug Discovery Generalist with Discrete Diffusion. ICML 2025.
>
> [2] Genetic-guided GFlowNets for Sample Efficient Molecular Optimization. NeurIPS 2024.
>
> # Weakness 3: Computational and memory efficiency
>
> > No computational or memory efficiency was reported. This would be an effective method to demonstrate that the proposed DemoDiff is more efficient compared to larger generalist LLMs.
>
> Thank you for the suggestions. For DemoDiff, the minimal GPU memory requirement is about 3.6 GB. In comparison, an LLM such as Llama3.1-8B requires about 9.3 GB. Although LLMs are about 10 times larger, the finetuned LLM still underperforms DemoDiff in this setting according to Table 1, with an average rank of around 10–14 compared to 4.1 for DemoDiff, and LLMs also use more memory.

---

### Official Review · Reviewer_Nsot · 2025-10-28

**Soundness:** 3
**Presentation:** 3
**Contribution:** 2
**Rating:** 6
**Confidence:** 4

**Summary:**

This paper proposes a novel molecular design framework, DemoDiff, which integrates diffusion models with in-context learning. By introducing a Node Pair Encoding (NPE) tokenizer for molecular graphs, the model achieves efficient pretraining with motif-level representations. A 0.7B-parameter model is pretrained on large-scale, multi-task data covering 155K properties and millions of molecules. DemoDiff achieves an average rank of 3.63 across 33 molecular design tasks, outperforming ten baseline models and several LLMs.

**Strengths:**

1. DemoDiff implicitly embeds task information into the diffusion denoising process. The proposed Node Pair Encoding (NPE) is a substantive contribution to molecular graph tokenization, eliminating the need for handcrafted reaction rules and enabling automatic motif discovery.
2. Extensive evaluation on 33 tasks demonstrates that DemoDiff significantly outperforms both domain-specific baselines and ICL-based LLMs. Despite having only 0.7B parameters, the model matches or even surpasses general-purpose LLMs.
3. The paper provides detailed descriptions of the pretraining task design, motif tokenizer, context consistency score, and diffusion inversion inference. The experiments offer comprehensive analyses of parameter scale, context length, positive/negative sample ratios, and consistency filtering.

**Weaknesses:**

1. Lack of theoretical analysis. Although the paper mentions an implicit Bayesian interpretation, it lacks formal proofs or theoretical analysis on how the diffusion trajectories reflect “posterior inference of task concepts.” The comparison with ICL in language models remains largely analogical, without quantitative or interpretable mechanism-level parallels.
2. Potential bias in data and task construction.
- The pretraining tasks mainly rely on ChEMBL and polymer datasets, whose property distributions are highly skewed (Zipf-like). While the authors claim this facilitates ICL ability, no systematic study is provided on the relationship between task frequency and generalization performance.
- The oracle evaluation for generation tasks depends on human scoring and predictor models, which may introduce noise.
3. Insufficient validation of NPE’s chemical semantic consistency. The paper lacks statistical verification of motif interpretability or chemical validity. Although DemoDiff outperforms Graph-DiT, no strictly fair comparison (e.g., controlled parameter size or identical token count) is presented.
4. High pretraining cost. The work lacks small-scale, reproducible experiments or an ablation-only release.

**Questions:**

1. What does the “Total sum” represent — which scores are included in this total? In Eq. 6, what is the function denoted by T?
2. The authors interpret DemoDiff’s in-context learning as implicit Bayesian inference. Could this interpretation be validated via attention pattern or diffusion trajectory visualizations?
3. NPE is a frequency-driven motif discovery algorithm. Do these automatically learned motifs correspond to known chemical functional groups or reaction fragments? How well does NPE generalize to out-of-distribution (OOD) chemical spaces? PRODIGY [1] also applies in-context learning on graphs — please provide a comparison.
4. The authors mention that the property distribution follows Zipf’s law (Figure 3a). Has the effect of this long-tailed distribution on the model’s ICL ability been evaluated? Does DemoDiff maintain strong performance on low-frequency property tasks? Since the number of examples per assay is imbalanced, could this lead to training bias toward high-frequency tasks?
5. The authors use consistency score and related metrics. Could evaluation be extended to include metrics more closely tied to molecular properties, such as QED or SA score?

Reference:

[1] PRODIGY: Enabling In-context Learning Over Graphs

---

> ### Author Response · Authors · 2025-11-19
> **Author Response [1/2]**
>
> # Weakness 1, Q2: Theoretical analysis and visualization
>
> > Lack of theoretical analysis. Although the paper mentions an implicit Bayesian interpretation, it lacks formal proofs or theoretical analysis on how the diffusion trajectories reflect “posterior inference of task concepts.” The comparison with ICL in language models remains largely analogical, without quantitative or interpretable mechanism-level parallels.
>
> > The authors interpret DemoDiff’s in-context learning as implicit Bayesian inference. Could this interpretation be validated via attention pattern or diffusion trajectory visualizations?
>
> Thank you for your suggestions. We have updated Appendix A1 to extend the in-context learning (ICL) explanation [1] from general LLMs to DemoDiff, based on Equation (1) in the paper and the diffusion process.
>
> In essence, the diffusion transformer for graphs, acting like an LLM for text, uses the prompt examples to infer a latent concept and predict the correct output for a new query [1]. This process is "implicit Bayesian inference" because the model does not update its weights; instead, it uses its pre-trained knowledge to perform a form of probabilistic inference, analogous to a Bayesian agent updating its prior beliefs with new data (the prompt examples) to form a posterior predictive distribution for the output.
>
> We include the subsection here for your reference. In Figure 7 of the main text, we show the diffusion trajectory in the reverse process. The figures show how DemoDiff gradually produces the molecule toward the target property given the context (molecule–property pairs) and the query score.
>
> ## Appendix A1
>
> We construct the pretraining dataset as $\mathcal{D} = \{(\mathcal{C}\_{i}, Q\_{i}, X\_{i})\}\_{i=1}^{N\_{\text{pretrain}}}$. Each task contains a context of molecule–score pairs $\mathcal{C}_i$, a query score $Q_i$, and a target molecule $X_i$. For each task, we first sample a latent concept (e.g., the thermal conductivity of polymeric materials from Figure 1) from a prior $\theta \sim p(\theta)$, followed by a set of molecule–property pairs associated with that concept. This induces the pretraining distribution:
>
> \begin{equation}
> p(\mathcal{C}, Q, X)
> = \int_{\theta} p(\theta)\, p(\mathcal{C} \mid \theta)\, p(Q \mid \mathcal{C}, \theta)
> \left[
> \int_{\mathbf{x}^{1:T}} p(\mathbf{x}^T)\,
> \prod_{t=1}^{T} p_{\theta} \left(\mathbf{x}^{t-1} \mid \mathbf{x}^{t}, Q, \mathcal{C} \right)\,
> d\mathbf{x}^{1:T}
> \right] d\theta,
> \end{equation}
>
> where $\mathbf{x}^{0}$ represents the feature of the target token. With sufficiently large and diverse pretraining data and scalable diffusion transformers, DemoDiff can a foundation model for ICL.
> For a downstream task, we use $(\mathcal{C}, Q)$ as the prompt, and $p(\theta \mid \mathcal{C}, Q)$ form the prompt concept. Then DemoDiff marginalizes the concept over the diffusion trajectories $\mathbf{x}^{1:T}$:
>
> \begin{equation}
> p(X \mid \mathcal{C}, Q)
> = \int_{\theta \in \Theta} \int_{\mathbf{x}^{1:T}}
> \left[ p(\mathbf{x}^T) \,
> \prod_{t=1}^T p_{\theta}\ \left(\mathbf{x}^{t-1} \mid \mathbf{x}^{t}, \mathcal{C}, Q \right) \right] \,
> p(\theta \mid \mathcal{C}, Q)\, d\mathbf{x}^{1:T}\, d\theta.
> \end{equation}
>
> ## Reference
> [1] An Explanation of In-context Learning as Implicit Bayesian Inference. ICLR 2022.

---

> > ### Author Response · Authors · 2025-11-19
> > **Author Response [2/2]**
> >
> > # Weakness 2, Q4: Potential bias in data and task construction: Zipf's law
> >
> > > The pretraining tasks mainly rely on ...
> >
> > > The authors mention that the property distribution follows Zipf’s law (Figure 3a) ...
> >
> > Thank you for the insightful comments. We follow established practices from LLM and Transformer pretraining to build DemoDiff for in-context learning [1,2]. Prior studies have investigated this problem in depth and have shown that certain properties of natural language data, such as a skewed Zipfian distribution, are important for in-context learning. We adopt these practices when pretraining DemoDiff in molecular domains. We quote a relevant statement from [1] below for your reference:
> >
> > > ''However, we found that models could simultaneously exhibit both in-context learning and in-weights learning when trained on a skewed marginal distribution over classes (akin to the Zipfian distribution of natural data).''
> >
> > We also discuss these works, along with broader studies on in-context learning from model, data, and learning perspectives, in the related work section.
> >
> > ## Reference
> >
> > [1] Data Distributional Properties Drive Emergent In-Context Learning in Transformers. NeurIPS 2022.
> >
> > [2] Pre-trained Models Perform the Best When Token Distributions Follow Zipf’s Law. EMNLP 2025.
> >
> > # Weakness 2: Oracle evaluation
> >
> > Thank you for the comment. We followed oracle evaluations that are widely used in molecular benchmarking and generation studies. Prior work [1] has also shown that although computational oracles are not experimental measurements, they still allow fair comparison of the relative performance of different generative models.
> >
> > ## Reference
> >
> > [1] Graph Diffusion Transformers for Multi-Conditional Molecular Generation. NeurIPS 2024.
> >
> > # Weakness 3, Q3: Motif interpretability or chemical validity
> >
> > > Insufficient validation of NPE’s chemical semantic consistency ...
> >
> > > NPE is a frequency-driven motif discovery algorithm ...
> >
> > Thank you for your comments. We have updated Appendix B.2 to discuss motif interpretability and chemical validity, with new content highlighted in blue.
> >
> > First, all motifs are valid. We provide supplementary materials (data/tokenizer/vocab3000ring300/pretrain-token.node) that show the motif vocabulary.
> >
> > Second, we compare the vocabulary with 48 common functional groups. 47 of them appear, and one is missing (details in Appendix B.2).
> >
> > Third, we compare the vocabulary with fragments produced by BRICS [1]. BRICS produces more than 129,000 fragments. Among these fragments, 98.5% appear fewer than 100 times in over one million pretraining molecules, and 63.2% appear only once. When we focus on the top 100 BRICS fragments, 89% appear in the NPE vocabulary.
> >
> > For an out-of-distribution molecule that does not appear in the NPE training set, the tokenizer uses standard atom and bond representations.
> >
> > ## Reference
> >
> > [1] On the Art of Compiling and Using 'Drug-Like' Chemical Fragment Spaces. ChemMedChem. 2008.
> >
> > # Weakness 3: DemoDiff vs Graph DiTs
> >
> > Thank you for your comment. The relationship between Graph DiTs and DemoDiff is similar to the relationship between Transformers and large language models. DemoDiff is built upon the Graph DiTs architecture, with several improvements: a tokenizer that moves from atom-level to motif-level representations, and a scaled model size suitable for in-context learning. In our experiments, we use a single DemoDiff model for all tasks. In contrast, Graph DiTs are designed for specific tasks, so a separate Graph DiTs model must be trained for each task.
> >
> > # Q1: Total sum
> >
> > > What does the “Total sum” represent ...
> >
> > Thank you for the question.
> >
> > 1. "Total sum" refers to the total sum of the scores across all tasks (i.e., across all task columns).
> >
> > 2. In Equation 8 (previously Equation 6), $T$ denotes the Tanimoto similarity. We have updated the description and added this clarification to the text.
> >
> > # Q3: Comparison with PRODIGY
> >
> > > PRODIGY [1] also applies ...
> >
> > Thank you for your comment. We have updated the related work section to include a new discussion of PRODIGY (highlighted in blue). Specifically, PRODIGY is an early attempt to adapt in-context learning to graphs, but it focuses on node and edge classification tasks rather than molecular generation.
> >
> > # Q5: Consistency score and evaluation metrics
> >
> > > The authors use consistency score ...
> >
> > Thank you for the comments. We used fingerprints to measure the distance between the generated molecule and the positive, medium, or negative context in this work. Other measures could be used to compute the consistency score if they are reasonably efficient and aligned with the design task.
> >
> > For the evaluation tasks, we included 33 design tasks that cover diverse molecular and polymer properties, including QED. Our combined score has two parts: the task score for the target property and the diversity score computed from the top-10 generated molecules.

---

### Official Review · Reviewer_aBww · 2025-10-29

**Soundness:** 1
**Presentation:** 3
**Contribution:** 2
**Rating:** 2
**Confidence:** 4

**Summary:**

The authors propose DemoDiff, a generative model that uses molecule-score examples for in-context molecular design. The authors also develop Node Pair Encoding (NPE), a strategy to represent molecules at the motif level, which requires 5.5x fewer nodes on average. DemoDiff achieves impressive performance across 33 molecular design tasks spanning 6 categories.

**Strengths:**

- The proposed NPE reduces the number of nodes by 5.5x, which is impressive and is a generally useful contribution even outside the context of DemoDiff.
- The proposed in-context learning approach is novel and a unique contribution to the ML molecular design literature
- DemoDiff demonstrates promising results on the drug/material design categories and is able to generate more diverse samples than baseline methods
- The authors provide extensive ablation and case studies

**Weaknesses:**

I am not very convinced by the main experimental results and I think Table 1 is somewhat misleading. Firstly, it is a bit unfair to compare to LLMs given that DemoDiff is trained on >1M molecule-property pairs and LLMs are not designed for this specific task. A more fair comparison would be to finetune the LLMs to do in-context molecule generation in the same way as DemoDiff. Secondly, and more importantly, after looking at the results in the Appendix, it appears that DemoDiff only achieves SOTA performance on 13 out of the 33 tasks (based on the Top-10 Harmonic Mean results). In particular, outside the drug/material design categories, DemoDiff does not reliably outperform baseline LLMs. Given that DemoDiff success is limited to a small subset of tasks it is difficult to support the claim that this is a generalizable method.

**Questions:**

LLMs are not necessarily trained to do in-context learning or to do molecular design, which makes it difficult to compare to your proposed method which is trained on a large and specialized dataset of molecule-spectra pairs. Did the authors consider finetuning LLMs to do in-context molecular design on the same dataset used for DemoDiff pretraining? Do you think DemoDiff would outperform such a finetuned LLM?

---

> ### Author Response · Authors · 2025-11-19
>
> # Weakness 1, Q1: Finetuning LLMs
>
> > I am not very convinced by the main experimental results and I think Table 1 is somewhat misleading. Firstly, it is a bit unfair to compare to LLMs given that DemoDiff is trained on >1M molecule-property pairs and LLMs are not designed for this specific task. A more fair comparison would be to finetune the LLMs to do in-context molecule generation in the same way as DemoDiff.
>
> > LLMs are not necessarily trained to do in-context learning or to do molecular design, which makes it difficult to compare to your proposed method which is trained on a large and specialized dataset of molecule-spectra pairs. Did the authors consider finetuning LLMs to do in-context molecular design on the same dataset used for DemoDiff pretraining? Do you think DemoDiff would outperform such a finetuned LLM?
>
> Thank you for the suggestion. We have added new experiments by finetuning the open weighted LLMs (Qwen3-8B and Llama3.1-8B). Each model is trained on a set of molecule–property pairs and a query score to predict the target molecule. We follow the evaluation setup in Table 1 and evaluate the top-10 generated molecules. Their performance is measured by the harmonic mean of the oracle scores and the internal diversity scores. We compare the results across task categories, and the updated results are shown below.
>
> | Method        | Drug Rediscovery | Drug MPO | Structure Constrained | Drug Design | Target Based | Material Design |
> |---------------|------------------|----------|-------------------------|-------------|---------------|------------------|
> | Llama3.1-8B-FT | 0.21±0.13 | 0.24±0.23 | 0.24±0.34 | 0.31±0.31 | 0.02±0.05 | 0.29±0.40 |
> | Qwen3-8B-FT    | 0.37±0.19 | 0.27±0.16 | 0.26±0.34 | 0.46±0.22 | 0.67±0.07 | 0.44±0.39 |
> | DemoDiff   | **0.44±0.21** | **0.54±0.23** | **0.56±0.33** | **0.79±0.11** | **0.78±0.05** | **0.67±0.11** |
>
> We observe that DemoDiff outperforms the finetuned LLMs, even though they are about 10 times larger and trained on the same molecular dataset. This highlights the strength of DemoDiff for in-context molecular design.
>
> We have updated Table 1 to include these new results, along with four additional baselines based on two recent papers [1,2]. We hope this update addresses your concern about the comparison with finetuned LLMs for molecular design.
>
> ## Reference
>
> [1] GenMol: A Drug Discovery Generalist with Discrete Diffusion. ICML 2025.
>
> [2] Genetic-guided GFlowNets for Sample Efficient Molecular Optimization. NeurIPS 2024.
>
> # Weakness 2: Baseline comparison
>
> > Secondly, and more importantly, after looking at the results in the Appendix, it appears that DemoDiff only achieves SOTA performance on 13 out of the 33 tasks ...
>
> Thank you for your thoughtful observations. We have updated the paper and now compare DemoDiff with 19 baselines across 33 design tasks. These baselines include the top-4 models from a benchmarking study [1] that evaluated 25 methods, several recent models [2,3,4], very large LLMs such as GPT-4o, Qwen-Max, and DeepSeek-3V, and specialized finetuned LLMs (Qwen3-8B and Llama3.1-8B). With such a broad set of baselines and tasks, it is difficult for any single method to dominate all settings. This is especially true because many baselines rely on oracle calls to optimize molecule structures, while DemoDiff does not. To summarize performance, we include the average rank and the total sum of scores. In the updated paper, DemoDiff reaches an average rank of 4.1 (smaller is better) and a total score of 20.10 (higher is better). The second best model reaches an average rank of 6.56 and a total score of 17.64.
>
> We also follow your suggestion and examine how often each method reaches the best result or the top-3 results when using the harmonic scores of the top-10 generations. DemoDiff is the best on 12 tasks and is in the top-3 on 19 tasks. The second best model, GPT-4o, is the best on 4 tasks and is in the top-3 on 7 tasks. These results show that DemoDiff is a better solution than other in context learning models.
>
> We agree that a fully generalizable model across all molecular design tasks is an important goal. We also note that this remains an open challenge in the field. In the conclusion section, we state that DemoDiff is a promising molecular foundation model with potential to scale with larger models, broader datasets, and more compute. We have also reviewed our claims to ensure they remain accurate and balanced. We appreciate your feedback and welcome any further suggestions to improve the presentation of this part of the paper.
>
> ## Reference
>
> [1] Sample Efficiency Matters: A Benchmark for Practical Molecular Optimization. NeurIPS 2022.
>
> [2] Genetic-guided GFlowNets for Sample Efficient Molecular Optimization. NeurIPS 2024.
>
> [3] Graph Diffusion Transformers for Multi-Conditional Molecular Generation. NeurIPS 2024.
>
> [4] GenMol: A Drug Discovery Generalist with Discrete Diffusion. ICML 2025.

---

> > ### Comment · Reviewer_aBww · 2025-11-27
> > **Response to Author Comments**
> >
> > I thank the authors for their response. In light of the positive results in a comparison with finetuning LLMs, I will increase my score. While the paper introduces some interesting ideas, I am hesitant to increase my score beyond a 4 because DemoDiff does not consistently outperform baseline methods across the different categories.

---

> ### Author Response · Authors · 2025-11-27
>
> Dear Reviewer aBww,
>
> Thank you very much for reconsidering your evaluation and for increasing your score. We appreciate your thoughtful comments and your recognition that our in-context design approach is novel and a unique contribution to the ML molecular design literature. We hope that the following clarifications will help address your remaining concerns.
>
> **DemoDiff is evaluated against 19 strong baselines on 33 tasks, including leading molecular optimization methods, domain-specific finetuned 8B LLMs, and very large closed-source models, across diverse scenarios in drug and material design.** In this competitive setting no single method wins on every task, especially because many baselines rely on oracle calls to optimize molecular structures while DemoDiff does not. Nevertheless, DemoDiff achieves the **best overall performance**, with an average rank of 4.10 (smaller is better) and a total score of 20.10 (higher is better), whereas the second-best rank is 6.56 and the second-best total score is 17.64 (achieved by different methods).
>
> Our goal is to contribute a step toward in-context molecular design through reusable molecular tokenization and scalable pretraining strategies and models, rather than to claim that the current DemoDiff is a final or universally best solution.
>
> We understand that ICLR values work that brings new and useful knowledge to the community together with strong empirical results, and we hope that the additional experiments and clarifications convey our contribution in this spirit. We would be very grateful for any further suggestions on how to improve the presentation, and we hope that the updated results and discussion help to reduce your remaining hesitation; if any concerns remain unaddressed, we would greatly appreciate the opportunity for further discussion and clarification.
>
> Best,
>
> Authors

---

### Official Review · Reviewer_v13C · 2025-11-01

**Soundness:** 2
**Presentation:** 3
**Contribution:** 2
**Rating:** 4
**Confidence:** 2

**Summary:**

This work introduces a conditional generative model for molecules based on a graph diffusion transformer.
To reduce the context size, atoms are grouped together using a BPE-like algorithm adapted to graphs.
Conditioning is done by using pairs of molecules with normalised scores as _property_ inputs to the graph diffusion transformer.
Experiments on a dataset extracted from ChEMBL and a collection of polymer datasets show favourable performance in terms of oracle and diversity scores.

**Strengths:**

- Overall, the paper is well written.
 - Most reported results come with error bars.
 - Reported results seem to be competitive.
 - Being able to control molecule generation on activity levels seems relevant.

**Weaknesses:**

- The proposed node pair encoding feels similar in spirit to how (extended connectivity) fingerprints are computed.
   However, the relation to these possibly related methods seems to be underexplored in the current version of the manuscript.
   It would be good to have an explanation how NPE is similar/different from fingerprint computations.
   If there is enough similarity, also an empirical verification of NPE vs fingerprint features would be meaningful.
   I also feel that NPE has some similarities with virtual nodes used in some GNNs (e.g. Hwang et al., 2022).
   A discussion on possible similarities/differences in this regard might be useful as well.
 - There is little to no motivation on why molecular graphs are used instead of e.g. SMILES.
   I have often heard that GNNs bring little to no performance advantage over working with SMILES (e.g. Renz et al., 2024).
   Furthermore, GNNs are typically much more complex to work with than models that work with SMILES.
   Given that the core Graph DiT idea seems transferrable to non-graph architectures as well,
   it would have been interesting to investigate how important the _graph_ aspect of this model is.
 - It is not entirely clear how strong the provided baselines are.
   Most notably, there is little information on how the LSTM was used to operate on the NPE encodings.
   Also, it seems hard to imagine that there are no more specialised models that are able to generate molecules in-context.
   Especially in the context of autoregressive SMILES generation (e.g. Renz et al., 2024; Schmidinger et al., 2025)
 - Figure&nbsp;2 confuses me more than it helps me to understand the method.
   I understand that molecules are encoded by grouping sub-structures,
   but there is no explanation for what $S_\mathrm{ingle}$ is supposed to be,
   or what the digits in the brackets are supposed to mean.
   Furthermore, the inputs ($[?, S0]$ and $[?, S5, S4, S7]$) to the transformer,
   which I would have assumed are on top of the demo tokens,
   seem to be the sub-motifs from the tokenized molecule illustrated to the right.
   However, this tokenized molecule should be part of the demo tokens.
   This all seems to make little sense and is hard to connect to the explanations in the main text.

### Minor Issues
 - The expression "molecule-assay pair" (line 257) seems to be a bit weird.
   Could it be that you mean "molecule-activity pair",
   i.e., a molecule with its activity value for a particular assay (a.k.a. the context)?
 - In the main text (line 302) _10_ novel, unique and valid molecules are mentioned for evaluation,
   but in the appendix, line 1054 mentions _100_ molecules for evaluation.
 - Table&nbsp;1 is claimed to report two scores (oracle and diversity),
   but there is only a single value for each model-task combination.

### Additional References
 - Hwang et al. (2022). [An analysis of virtual nodes in graph neural networks for link prediction](https://openreview.net/forum?id=dI6KBKNRp7). In The first learning on graphs conference.
 - Renz et al. (2024). [Diverse hits in de novo molecule design: Diversity-based comparison of goal-directed generators](https://pubs.acs.org/doi/full/10.1021/acs.jcim.4c00519). Journal of Chemical Information and Modeling, 64(15), 5756-5761.
 - Schmidinger et al. (2025). [Bio-xLSTM: Generative modeling, representation and in-context learning of biological and chemical sequences](https://openreview.net/forum?id=IjbXZdugdj). International Conference on Learning Representations.

**Questions:**

1. Is there any/What is the relation between NPE and the computation of molecular fingerprints?
 2. Is there any/What is the relation between NPE and the use of virtual nodes (cf. Hwang et al., 2022) in GNNs?
 3. Could the proposed model architecture also work with non-graph (e.g. SMILES) inputs?
 4. How was the LSTM applied to the NPE-encoded graphs?
 5. Are there no other (e.g. SMILES-based) specialised generative models that could be used as baselines?
 6. Can you explain what is happening in Figure&nbsp;2?

---

> ### Author Response · Authors · 2025-11-19
> **Author Response [1/2]**
>
> # Weakness 1, Q1, Q2: Comparison of NPE with fingerprints and virtual nodes
>
> > The proposed node pair encoding feels similar in spirit to how (extended connectivity) fingerprints are computed ...
>
> > Is there any/What is the relation between NPE and the computation of molecular fingerprints?
>
> > Is there any/What is the relation between NPE and the use of virtual nodes (cf. Hwang et al., 2022) in GNNs?
>
>
> Thank you for the comment. NPE differs from fingerprints [1] and virtual nodes in three aspects: (1) representation and invertibility, (2) computation process, and (3) use cases. Specifically:
>
> (1) NPE builds a motif vocabulary from data and encodes a molecule as a graph of motifs with typed edges and attachment rules, allowing the atom-level structure to be reconstructed. In contrast, Morgan fingerprints hash local atom neighborhoods into a fixed-length bit vector, which is lossy and not invertible. Virtual nodes are additional nodes inserted into graphs to provide shortcuts for information exchange between nodes rather than forming a new molecular representation. If we use only virtual nodes to re-represent the molecule, the structure is not invertible.
>
> (2) NPE identifies frequent substructures from a dataset to build a discrete motif vocabulary. In contrast, Morgan fingerprints are computed directly from atom invariants with a chosen radius and bit size. Virtual nodes can be added randomly or through generic graph partitioning algorithms, which do not reflect how frequently a partition or cluster appears across the molecular dataset.
>
> (3) NPE produces a compact motif-level graph suited for generative tasks. In contrast, Morgan fingerprints produce a vector suited for predictive tasks. Virtual nodes are used to create shortcuts for sharing information between graph nodes.
>
> In summary, fingerprints [1] are designed for prediction models. They are not invertible, so they are not suitable for reconstruction in generative models. For empirical verification, we also built a vocabulary with fragments from BRICS [2], which uses reaction rules. BRICS produces more than 129,000 fragments. Among these fragments, 98.5% appear fewer than 100 times in over one million pretraining molecules, and 63.2% appear only once. When we focus on the top 100 BRICS fragments, 89% of them appear in the NPE vocabulary with a vocabulary size of 3000. This shows that NPE can capture frequent and chemically meaningful motifs in a more efficient way.
>
> ## Reference
>
> [1] Extended-connectivity fingerprints. Journal of chemical information and modeling. 2010.
>
> [2] On the Art of Compiling and Using 'Drug-Like' Chemical Fragment Spaces. ChemMedChem. 2008.
>
> # Weakness 2: Why based on molecular graphs
>
> > There is little to no motivation on why molecular graphs are used instead of e.g. SMILES ...
>
> Thank you for your question. Compared to SMILES, molecular graphs capture the direct adjacency between atoms. Two atoms that are neighbors in the graph may appear far apart in a SMILES string.
>
> Graph models such as Graph Diffusion Transformers (Graph DiTs) capture this connectivity more effectively during graph generation. We follow Graph DiTs [1], which show better performance than LSTM-HC (which works on SMILES), one of the leading baselines in the study by Renz et al., 2024 [2], for multi-conditional generation.
>
> Graph DiTs make use of structural information in molecular graphs. In the forward process, the model uses the graph structure to define the noise process through a transition matrix that considers atoms and bonds together. Ablation studies show that this choice is useful. In the reverse process, the model applies a Transformer that treats atoms and bonds in a unified way for denoising. Because of these advantages and the Transformer's ability to scale, we focus on Graph DiTs in this work.
>
> ## Reference:
>
> [1] Graph Diffusion Transformers for Multi-Conditional Molecular Generation. NeurIPS 2024.
>
> [2] Diverse hits in de novo molecule design: Diversity-based comparison of goal-directed generators. Journal of Chemical Information and Modeling, 64(15), 5756-5761.

---

> > ### Author Response · Authors · 2025-11-19
> > **Author Response [2/2]**
> >
> > # Weakness 3, Q5: Baselines
> >
> > > It is not entirely clear how strong the provided baselines are ....
> >
> > > Are there no other (e.g. SMILES-based) specialised generative models that could be used as baselines?
> >
> > Thank you for the suggestions. We added new experimental results for 6 additional baselines and updated Table 1. To summarize, now we cover a broad set of 19 baselines and 33 tasks. We compare DemoDiff with 19 baselines across 33 design tasks. These baselines include the top-4 models from a benchmarking study [1] that evaluated 25 methods, several very recent models [2,3,4], very large LLMs such as GPT 4o, Qwen Max, and DeepSeek 3V, and specialized finetuned LLMs.
> >
> > For your reference, we present the new results on two finetuned open-weighted LLMs (Qwen3-8B and Llama3.1-8B). They are finetuned on the same dataset used for DemoDiff pretraining. Each model is trained on a set of molecule–property pairs with a query score. We follow the evaluation setup in Table 1 and evaluate the top-10 generated molecules. Their performance is measured by the harmonic mean of the oracle scores and the internal diversity scores. We compare the results across task categories. Results are shown below.
> >
> > | Method        | Drug Rediscovery | Drug MPO | Structure Constrained | Drug Design | Target Based | Material Design |
> > |---------------|------------------|----------|-------------------------|-------------|---------------|------------------|
> > | Llama3.1-8B-FT | 0.21±0.13 | 0.24±0.23 | 0.24±0.34 | 0.31±0.31 | 0.02±0.05 | 0.29±0.40 |
> > | Qwen3-8B-FT    | 0.37±0.19 | 0.27±0.16 | 0.26±0.34 | 0.46±0.22 | 0.67±0.07 | 0.44±0.39 |
> > | DemoDiff   | **0.44±0.21** | **0.54±0.23** | **0.56±0.33** | **0.79±0.11** | **0.78±0.05** | **0.67±0.11** |
> >
> > We observe that DemoDiff outperforms the specialized finetuned LLMs, even though they are about 10 times larger and trained on the same molecular dataset. This shows the advantage of DemoDiff as a foundation model for in context molecular design.
> >
> > ## Reference
> >
> > [1] Sample Efficiency Matters: A Benchmark for Practical Molecular Optimization. NeurIPS 2022.
> >
> > [2] Genetic-guided GFlowNets for Sample Efficient Molecular Optimization. NeurIPS 2024.
> >
> > [3] Graph Diffusion Transformers for Multi-Conditional Molecular Generation. NeurIPS 2024.
> >
> > [4] GenMol: A Drug Discovery Generalist with Discrete Diffusion. ICML 2025.
> >
> > # Weakness 4, Q6: Figure 2
> >
> > > Figure 2 confuses me more than it helps me to understand the method ...
> >
> > Thank you for your comments. We have updated Figure 2 and revised the caption to make it clearer (highlighted in blue).
> >
> > The grouped substructures are the motifs (tokens) produced by the tokenizer. The values in the square brackets are the edge attributes. The first value is the bond type (for example, single bond), and the second value is the attachment specification, which shows the position where the bond originates. These positions are also shown in the motifs.
> >
> > For the transformer input, the tokens come from the molecules in the generation step and in the denoising step. They do not come from the demonstrations. We have corrected the issue and updated the figure.
> >
> > # Minor 1 & 2: Typos
> >
> > > The expression "molecule-assay pair" (line 257) seems to be a bit weird. Could it be that you mean "molecule-activity pair", i.e., a molecule with its activity value for a particular assay (a.k.a. the context)?
> >
> > > In the main text (line 302) 10 novel, unique and valid molecules are mentioned for evaluation, but in the appendix, line 1054 mentions 100 molecules for evaluation.
> >
> > Thank you for pointing out the typos. We have corrected them in the corresponding lines (highlighted in blue):
> >
> > (1) “molecule-assay pair” → “molecule-activity pair”
> >
> > (2) “100” → “10”
> >
> > # Minor 3: Table 1
> >
> > > Table 1 is claimed to report two scores (oracle and diversity), but there is only a single value for each model-task combination.
> >
> > Thank you for your comment. The reported score is the harmonic mean of the oracle scores and the diversity scores.
> >
> > # Q3: Architecture for SMILES
> >
> > > Could the proposed model architecture also work with non-graph (e.g. SMILES) inputs?
> >
> > Thank you for your input. The model architecture follows Graph DiTs [1]. Graph DiTs are designed for graph structured inputs.
> >
> > ## Reference
> >
> > [1] Graph Diffusion Transformers for Multi-Conditional Molecular Generation. NeurIPS 2024.
> >
> > # Q4: NPE-encoded graphs for LSTM
> >
> > > How was the LSTM applied to the NPE-encoded graphs?
> >
> > Thank you for your question. LSTM models are designed for sequence data. We did not use an LSTM to process NPE-encoded graphs in this work (i.e., the encoding result is still a graph).

---

> > > ### Comment · Reviewer_v13C · 2025-11-26
> > > **Thank you for addressing my concerns**
> > >
> > > Thank you for pointing out the advantages of NPE. Are you planning to include (a short version of) this explanation into the main text or is this only relevant for this discussion?
> > >
> > > > Compared to SMILES, molecular graphs capture the direct adjacency between atoms
> > >
> > > That is the promise of GNNs since the start. However, it seems that none of the many GNNs has been able to capitalise on this advantage.
> > >
> > > > Graph DiTs [1], which show better performance than LSTM-HC
> > >
> > > They report better results, but they did not adopt the more rigorous test framework from (Renz et al., 2024), rendering the comparison pretty much meaningless. The point of Renz et al. (2024) is that most of the metrics used for evaluating molecular generative models can be boosted in trivial ways that do not reflect a better understanding of the underlying chemistry.
> > >
> > >  > Because of [...] the Transformer's ability to scale
> > >
> > > What do you mean with "scale"? Transformers scale quadratically with sequence length, compared to LSTMs who scale linearly. I guess that is not what you mean here.
> > > Furthermore, do graph transformers "scale better" than text transformers (which could process SMILES)?
> > >
> > > > finetuned on the same dataset used for DemoDiff pretraining
> > >
> > > Does this mean that they got NPEs as inputs? If not, I would assume you used SMILES, which would explain its role in your rebuttal to my concerns. What kind of tokenisation strategies did you consider and eventually use to process the SMILES strings? Have you considered adapting NPE to SMILES? Do you have any details on the hyper-parameter tuning for these models?
> > >
> > > > We have updated Figure 2 and revised the caption to make it clearer (highlighted in blue)
> > >
> > > This already helps a lot, but is there any difference between $\mathrm{S}_\mathrm{ingle}$ in the encoder/decoder and $\mathrm{S}$? If not, I would propose to use the short-hand notation everywhere for clarity/consistency.
> > >
> > > Also, it is still not clear what the [motif, edge-feature(s)] syntax aims to represent in the transformer block. According to the main text this would be a single token. Does that mean that the two tokens in the figure represent one molecule? If yes, it would be useful to use the figure to provide intuition on how multiple molecules are encoded. Otherwise, it would be useful to show how molecules with multiple motifs would be represented.
> > >
> > > > harmonic mean of the oracle scores and the diversity scores
> > >
> > > Why the harmonic mean instead of the arithmetic or the geometric mean? Does it even make sense to average both scores? Wouldn't it be possible to report both scores and let readers decide what they find more valuable?
> > >
> > > I still find the caption of table 1 not clear enough. How about "We compute oracle and diversity scores and report the harmonic mean of these two values"?
> > >
> > > > We did not use an LSTM to process NPE-encoded graphs in this work
> > >
> > > Table 1 reports results of an LSTM. Since there is no mention of another data-modality than NPE-endcoded graphs, I had to assume that these were trained on the same data as the proposed model. Does this mean that LSTMs have been trained on plain SMILES representations? Also, what inputs were provided to Graph-DiT and the other baselines? I could not find any pointers to other input modalities when glossing over the experiment section again.

---

> ### Author Response · Authors · 2025-11-26
> **Response to Follow-Up Questions [1/2]**
>
> Dear Reviewer v13C,
>
> We appreciate your follow-up questions and thank you for the opportunity to clarify these points.
>
> # Q1:
> > Thank you for pointing out the advantages of NPE. Are you planning to include (a short version of) this explanation into the main text or is this only relevant for this discussion?
>
> Thank you for the suggestion. We have updated the main text in Section 3.1 and added a detailed discussion in Appendix A.1.
>
> # Q2:
> > That is the promise of GNNs since the start. However, it seems that none of the many GNNs has been able to capitalise on this advantage.
>
> Thank you for your comment. Molecular graphs provide a direct view of atomic adjacency that sequence formats cannot express. Many graph models, including GNNs and Graph Transformers, aim to make full use of this structure for prediction and generation. In this work, we present DemoDiff and compare it with 19 baselines, covering both graph models and sequence models such as LLMs and LSTMs, across 33 tasks. We hope these results offer further evidence that progress on graph-based models can better reflect the strengths of graph representations.
>
> # Q3:
>
> > They report better results, but they did not adopt the more rigorous test framework from (Renz et al., 2024), rendering the comparison pretty much meaningless. The point of Renz et al. (2024) is that most of the metrics used for evaluating molecular generative models can be boosted in trivial ways that do not reflect a better understanding of the underlying chemistry.
>
> Thank you for your comment. We agree that the two works focus on different directions and were developed in parallel (i.e., they were published in the same year). Renz et al. (2024) focused on diverse hit discovery in drug design, whereas Graph DiTs targeted multi-conditional generation, where the conditioning signal can be continuous, especially for material properties that are not limited to [0,1]. Both studies move molecular design forward, but in different ways.
>
> In this work, we evaluate 33 tasks across drug design and material design, and we include generation diversity as one of the metrics. We hope this broader setting helps clarify the strengths and limits of DemoDiff and all 19 baselines.
>
> # Q4:
>
> > What do you mean with "scale"? Transformers scale quadratically with sequence length, compared to LSTMs who scale linearly. I guess that is not what you mean here. Furthermore, do graph transformers "scale better" than text transformers (which could process SMILES)?
>
> Thank you for the question. Here, “scale” refers to the scalability of transformers, meaning that performance tends to improve when compute, data, and model size increase [1,2,3]. This pattern supports the confidence behind building large-scale foundation models.
>
> Whether graph transformers or text transformers scale better for molecular tasks is still open. On the one hand, graph transformers use molecular graph structures directly, which can express atomic connectivity clearly. On the other hand, text transformers follow the original transformer design, which has shown scalability across many domains. We used Graph DiTs [4] because they combine both ideas: they operate on molecular graphs while keeping the original (diffusion) transformer backbone [1] that supports scaling. DemoDiff builds on Graph DiTs with Node Pair Encoding and in-context molecular design pretraining. We hope these results provide useful evidence on the roles of both graph and transformer architectures in molecular design.
>
> ## Reference:
>
> [1] Scalable Diffusion Models with Transformers. ICCV 2023.
>
> [2] Emergent Abilities of Large Language Models. TMLR 2022.
>
> [3] Language Models are Few-Shot Learners. NeurIPS 2020.
>
> [4] Graph Diffusion Transformers for Multi-Conditional Molecular Generation. 2024.

---

> > ### Author Response · Authors · 2025-11-26
> > **Response to Follow-Up Questions [2/2]**
> >
> > # Q5:
> > > Does this mean that they got NPEs as inputs? If not, I would assume you used SMILES, which would explain its role in your rebuttal to my concerns. What kind of tokenisation strategies did you consider and eventually use to process the SMILES strings? Have you considered adapting NPE to SMILES? Do you have any details on the hyper-parameter tuning for these models?
> >
> > Thank you for the question. We fine-tune LLMs such as Llama3.1 and Qwen3, and they use their own built-in tokenization methods based on byte pair encoding. These models do not take NPE tokens as input.
> >
> > Adapting an NPE-style method to SMILES is an interesting idea, but it raises several technical challenges. A graph motif can attach through many possible atoms, while a SMILES string usually has one or two connection points. Mapping these flexible attachment choices into a linear sequence is not straightforward. We agree that this direction is promising, but it is outside the current scope and we plan to explore it in future work.
> >
> > For hyperparameters, we fine-tune the language models with a learning rate of 1e-4, a cosine scheduler with a warmup ratio of 0.1, two training epochs, and LoRA. These settings are commonly in LLM finetuning to keep a balance between the pretrained knowledge and the new task.
> >
> > # Q6
> > > This already helps a lot, but is there any differ...
> >
> > Thank you for the suggestion. We have updated the figure to use consistent notation for the bond attributes.
> >
> > # Q7:
> > > Also, it is still not clear what the [motif, edge-feature(s)] syntax aims to represent in the transformer block. According to the main text this would be a single token. Does that mean that the two tokens in the figure represent one molecule? If yes, it would be useful to use the figure to provide intuition on how multiple molecules are encoded. Otherwise, it would be useful to show how molecules with multiple motifs would be represented.
> >
> > Thank you for the suggestions. We follow the input design of Graph DiTs (Section 2.2) and extend it with motif-level molecular representations. As described around line 205, each DemoDiff input token contains both the motif features and the bond features, formed by concatenating their vectors. The transformer input is split into two parts in the figure: tokens for the denoising process and tokens for the context (Demo Tokens). We have updated the caption to clarify this.
> >
> > # Q8:
> > > Why the harmonic mean instead of the arithmetic or the geometric mean? Does it even make sense to average both scores? Wouldn't it be possible to report both scores and let readers decide what they find more valuable?
> >
> > Thank you for the suggestion. We evaluate model generation with computational oracles for two purposes: (1) whether the model can design a diverse set of molecules, and (2) whether the model can produce at least one strong molecule. These goals are reflected in Table 1 and Table 5. In Table 1, we report the harmonic mean to balance oracle and diversity scores from the top-10 generations. The three means behave similarly when the two scores are both high or both low, but they differ when the scores are far apart. The harmonic mean is more sensitive to such differences, which is why we use it. For example, if the two scores are 0.8 and 0.1, the arithmetic, geometric, and harmonic means are 0.45, 0.28, and 0.18. For readers who care only about the strongest molecule, Table 5 reports the best oracle score directly based on the computational oracles.
> >
> >
> > # Q9:
> > > Table 1 reports results of an LSTM. Since there is no mention of another data-modality than NPE-endcoded graphs, I had to assume that these were trained on the same data as the proposed model. Does this mean that LSTMs have been trained on plain SMILES representations? Also, what inputs were provided to Graph-DiT and the other baselines? I could not find any pointers to other input modalities when glossing over the experiment section again.
> >
> > Thank you for the question. For all baselines, we follow their standard and original implementations. The LSTMs are trained on SMILES strings with their usual predefined vocabularies containing atom symbols. The Graph DiT baselines also follow the original setup and use atom-level graph representations. We have updated the beginning of Section 4 to make this clearer.
> >
> > We appreciate your additional questions, which helped us clarify the paper further. We have updated the paper accordingly and addressed your concerns. If you have any remaining questions during the discussion period, we would be glad to continue the exchange.

---

> > > ### Comment · Reviewer_v13C · 2025-11-27
> > > **Key point missing**
> > >
> > > ### Q2
> > > > We hope these results offer further evidence
> > >
> > > No, they do not, since the used evaluation method does not properly measure how good a method really is (Renz et al., 2024).
> > >
> > > ### Q3
> > > > We agree that the two works focus on different directions
> > >
> > > This implies that you consider improving metric values to be "a different direction" than improving performance of generative models. This would be a significant limitation of the proposed method.
> > >
> > > The key point of the work of Renz et al. (2024) is that improving metric values is meaningless when improper metrics are chosen and/or evaluation protocols are not properly controlled.
> > >
> > > ### Q4
> > > > performance tends to improve when compute, data, and model size increase
> > >
> > > This holds for pretty much any deep learning model. However, this does not provide any intuition on why molecular graphs were used in favour of e.g. SMILES strings. As a matter of fact, it makes it a bit counter-intuitive, since the scaling behaviour of text-based transformers is much better understood (e.g. Kaplan et al., 2020; Hoffmann et al., 2022).
> > >
> > > ### Q5
> > > > We fine-tune LLMs such as Llama3.1 and Qwen3, and they use their own built-in tokenization methods based on byte pair encoding
> > >
> > > Does this make sense? The data distribution for pre-training Llama3.1 and Qwen3 looks very different from SMILES syntax. This also means that the pre-trained tokenizers are not optimised for SMILES at all. Therefore, I would expect these baselines to be relatively poor. Instead, it would be more interesting to fine-tune one of the many pre-trained models that have been properly trained on SMILES (e.g. Özçelik et al., 2024; Schmidinger et al., 2025).
> > >
> > > > These settings are commonly in LLM finetuning to keep a balance between the pretrained knowledge and the new task.
> > >
> > > To what extent do you want to keep the pre-trained knowledge in a general language model when the goal is to generate SMILES. Keeping pre-trained knowledge seems like it could be even counter-productive as this knowledge will not conform with SMILES syntax.
> > >
> > > ### Q6/7
> > >
> > > I am not sure if this improved the figure. It is still not clear whether the example in the figure displays multiple tokens of one molecule or multiple molecules consisting of single motifs. Do you use separator tokens to distinguish between molecules or are tokens just concatenated? Also, Is there any reason why you chose the long notation in favour of the short notation for the bond type?
> > >
> > > ### Q8
> > >
> > > I am not familiar enough with these oracle scores to interpret them, but it feels like a lot of information could be hidden by averaging these two metrics.
> > >
> > > I think my main concern is that the focus on graph networks for this work, might lead the field in the wrong direction. Since this work seems to be introducing a new paradigm for molecule generation, I would expect other modalities to be given a proper chance as well.
> > >
> > > ### Additional References
> > >  - Hoffmann et al. (2022). [An empirical analysis of compute-optimal large language model training](https://papers.neurips.cc/paper_files/paper/2022/hash/c1e2faff6f588870935f114ebe04a3e5-Abstract.html). Advances in Neural Information Processing Systems, 35, 30016–30030.
> > >  - Kaplan et al. (2020). [Scaling Laws for Neural Language Models](https://doi.org/10.48550/arXiv.2001.08361). arXiv:2001.08361.
> > >  - Özçelik et al. (2024). [Chemical language modeling with structured state space sequence models](https://www.nature.com/articles/s41467-024-50469-9). Nature Communications, 15(1), 6176.
> > >  - Schmidinger et al. (2025). [Bio-xLSTM: Generative modeling, representation and in-context learning of biological and chemical sequences](https://openreview.net/forum?id=IjbXZdugdj). International Conference on Learning Representations 13.

---

> > > > ### Author Response · Authors · 2025-11-29
> > > > **Response to the key points [1/2]**
> > > >
> > > > # Q2, Q3 Metric
> > > > > No, they do not, since the used evaluation method does not properly measure how good a method really is (Renz et al., 2024).
> > > > > This implies that you consider improving metric values to be "a different direction" than improving performance of generative models. This would be a significant limitation of the proposed method.
> > > > > The key point of the work of Renz et al. (2024) is that improving metric values is meaningless when improper metrics are chosen and/or evaluation protocols are not properly controlled.
> > > >
> > > > Thank you for your comment. The main idea in Renz et al. (2024) focuses on “diverse hits,” which combine a computational oracle (JNK3, GSK3β, DRD2, also included in our tasks) with a diversity measure. This aligns with the purpose of Table 1, which evaluates both aspects.
> > > >
> > > > **To further address your concern, we updated the results with the “diverse hits” metric in Table 7. The results show that under the metric from Renz et al. (2024), DemoDiff achieves an average rank of 2.67 among 19 baselines, compared with 4.67 for the second-best baseline, which aligned with the observations from Table 1.**
> > > >
> > > > # Q4, Q5 Comparison with Chem-xLSTM
> > > >
> > > > > This holds for pretty much any deep learning model. However, this does not provide any intuition on why molecular graphs were used in favour of e.g. SMILES strings. As a matter of fact, it makes it a bit counter-intuitive, since the scaling behaviour of text-based transformers is much better understood (e.g. Kaplan et al., 2020; Hoffmann et al., 2022).
> > > > > Does this make sense? The data distribution for pre-training Llama3.1 and Qwen3 looks very different from SMILES syntax. This also means that the pre-trained tokenizers are not optimised for SMILES at all. Therefore, I would expect these baselines to be relatively poor. Instead, it would be more interesting to fine-tune one of the many pre-trained models that have been properly trained on SMILES (e.g. Özçelik et al., 2024; Schmidinger et al., 2025).
> > > > > To what extent do you want to keep the pre-trained knowledge in a general language model ...
> > > >
> > > > Thank you for the comment. We fine-tuned LLMs based on suggestions from other reviewers and on observations that LLMs contain broad knowledge in chemistry and SMILES (for example, GPT-4o shows strong in-context molecular design performance in Table 1). Following your suggestion, we added new Chem-xLSTM baselines. We used the pretrained model from Schmidinger et al. (2025) and fine-tuned it on our pretraining molecule datasets. Tables A and B below show that DemoDiff consistently achieves the best performance compared with these baselines.
> > > >
> > > > Table A: Harmonic mean of oracle and diversity scores (Top-10).
> > > > We compute oracle and diversity scores from the top-10 generations and report the harmonic mean of these two values. The 33 tasks are grouped into six categories, and we report mean ± standard deviation within each category. Best results in each column are bolded.
> > > >
> > > > | Method              | Drug Rediscovery | Drug MPO       | Structure Constrained | Drug Design     | Target Based    | Material Design  |
> > > > |---------------------|------------------|----------------|-----------------------|-----------------|-----------------|------------------|
> > > > | Chem-xLSTM-Pretrain | 0.33 ± 0.15      | 0.33 ± 0.22    | 0.44 ± 0.25           | 0.35 ± 0.33     | 0.64 ± 0.03     | 0.12 ± 0.26      |
> > > > | Chem-xLSTM-Finetuned       | 0.40 ± 0.14      | 0.51 ± 0.22    | 0.46 ± 0.26           | 0.78 ± 0.14     | 0.69 ± 0.06     | 0.30 ± 0.22      |
> > > > | **DemoDiff (Ours)** | **0.44 ± 0.21**  | **0.54 ± 0.23**| **0.56 ± 0.33**       | **0.79 ± 0.11** | **0.78 ± 0.05** | **0.67 ± 0.11**  |
> > > >
> > > > Table B: Top-1 oracle scores across six task categories.
> > > > We compute the oracle score of the top generated molecule for each task. The 33 tasks are grouped into six categories, and we report mean ± standard deviation within each category. Best results in each column are bolded.
> > > > | Method             | Drug Rediscovery | Drug MPO       | Structure Constrained | Drug Design     | Target Based    | Material Design  |
> > > > |--------------------|------------------|----------------|-----------------------|-----------------|-----------------|------------------|
> > > > | Chem-xLSTM-Pretrain| 0.31 ± 0.15      | 0.41 ± 0.21    | 0.45 ± 0.26           | 0.44 ± 0.39     | 0.63 ± 0.08     | 0.19 ± 0.35      |
> > > > | Chem-xLSTM-Finetuned      | 0.37 ± 0.19      | 0.51 ± 0.20    | 0.49 ± 0.30           | 0.88 ± 0.10     | 0.75 ± 0.09     | 0.65 ± 0.36      |
> > > > | **DemoDiff (Ours)**| **0.54 ± 0.33**  | **0.54 ± 0.19**| **0.59 ± 0.37**       | **0.91 ± 0.07** | **0.77 ± 0.10** | **0.93 ± 0.16**  |
> > > >
> > > > Regarding scaling, the findings of Kaplan et al. (2020) and Hoffmann et al. (2022) are based on natural language data. The intuition that transformer models scale well for SMILES comes from the architecture itself, and this also applies to Graph DiTs, which preserve the transformer architecture while operating on molecular graphs.

---

> ### Author Response · Authors · 2025-11-29
> **Response to the key points [2/2]**
>
> # Q6/7 Figure
> > I am not sure if this improved the figure. It is still not clear whether the example in the figure displays multiple tokens of one molecule or multiple molecules consisting of single motifs. Do you use separator tokens to distinguish between molecules or are tokens just concatenated? Also, Is there any reason why you chose the long notation in favour of the short notation for the bond type?
>
> Thank you for your comment. Figure 2 shows that each molecule can have multiple tokens: the denoising tokens represent the molecule being designed, and the demo tokens come from different molecules. The longer notation carries more information than the short form, so we keep the longer notation.
>
> # Q8:
> > I am not familiar enough with these oracle scores to interpret them, but it feels like a lot of information could be hidden by averaging these two metrics.
> > I think my main concern is that the focus on graph networks for this work, might lead the field in the wrong direction. Since this work seems to be introducing a new paradigm for molecule generation, I would expect other modalities to be given a proper chance as well.
>
> Thank you for your comment. We had more details on the Oracle functions and tasks in Appendix D.1. **We have added a new Table 6 to present Oracle and diversity scores separately.**
>
> We appreciate your recognition of the broader impact of the work.  **Our goal is to extract richer information from molecular structures, not to introduce a new graph network architecture. We use the transformer architecture. To clarify it: one can view our procedure as analogous to processing motif-level SMILES.** After obtaining motifs, we could linearize them and assign each motif a unique identifier. For each motif, we keep its attachment index and edge connection information to other motifs. These features can then be concatenated and passed to a transformer, following how Graph DiTs process their inputs. This mirrors how edge information would be preserved if SMILES were segmented into motif-level units.

---

### Public Comment · ~Thanawat_Sornwanee1 · 2026-04-24
**Loss**

For the pre-trained loss, should the outer expectation be taken over the joint (x0, C, Q)?

---

### Meta-Review · Area_Chair_mrGE · 2026-01-07

**Summary:**

This paper proposes DemoDiff, a novel in-context molecular design framework that integrates demonstration-conditioned generation into graph diffusion transformers. By introducing Node Pair Encoding (NPE), the authors reduce graph size and enable scalable pretraining. Extensive experiments across 33 molecular design tasks demonstrate that DemoDiff achieves strong average performance.

The reviewers raised concerns spanning missing baselines, evaluation setting and performance. In response, the authors provided clarifications on NPE, revised Figure 2, and added missing baselines (including GenMol, Genetic GFN, and fine-tuned Qwen3-8B and Llama3.1-8B). Overall, the authors addressed most of the reviewers’ concerns; however, several issues remain open. Further evaluations of NPE are needed to better demonstrate its advantages. Additionally, training a SMILES-based autoregressive model from scratch on the same training data would provide a more direct validation of the GraphDiT backbone. Finally, incorporating additional evaluation metrics could offer a more comprehensive assessment of model performance.

Considering the novelty of the proposed NPE and the comprehensiveness of the experimental evaluation, I recommend accepting the paper as a poster.

**Reviewer Concerns:**

Reviewer v13C raised several points, including the distinction between NPE and fingerprints, a comparison with SMILES-based methods, the implementation details of baselines, and the confusion surrounding Figure 2. The author explained them in the response, add several LLM-based baselines and modified Figure 2.

Reviewer aBww questioned about comparison with fine-tuned LLMs with the same training data, and  argued that DemoDiff does not reliably outperform baselines in the experiments.  To address this concern, the authors fine-tuned Qwen3-8B and Llama3.1-8B using the same training data and incorporated these results into the main experiments.

Reviewer Nsot expressed the concerns including theoretical analysis, the bias in data and task construction, insufficient evaluation of NPE and so on. In response, the authors provided additional clarifications and introduced an experiment comparing NPE with fragment vocabularies generated by BRICS.

Reviewer t5zC questioned the sensitivity of demonstration/context selection, noted some missing baselines such as GenMol and Genetic GFN, and requested an efficiency analysis. The authors explained the strategy to select demonstrations, added GenMol and Genetic GFN as baselines, and reported  the GPU memory cost.

Reviewer WQjW raised concerns about the reliance on the consistency score as an evaluation metric, requested further evaluation of NPE, and questioned whether replacing the GraphDiT backbone with an autoregressive SMILES-based model could yield comparable performance. The author provided clarifications and further demonstrated the advantage of the GraphDiT backbone by fine-tuning Qwen3-8B and Llama3.1-8B for comparison.

Overall, the authors addressed most of the reviewers’ concerns; however, several issues remain open. Further evaluations of NPE are needed to better demonstrate its advantages. Additionally, training a SMILES-based autoregressive model from scratch on the same training data would provide a more direct validation of the GraphDiT backbone. Finally, incorporating additional evaluation metrics could offer a more comprehensive assessment of model performance.

**Reviewer Scores:**

Reviewer v13C indicated limited experience with this research area. Reviewer aBww said that he/she increased the score, while the other reviewers opted to keep their current scores.

---

### Decision · Program_Chairs · 2026-01-26

Accept (Poster)